# Mature tertiary lymphoid structures evoke intra-tumoral T and B cell responses via progenitor exhausted CD4$^+$ T cells in head and neck cancer

Hao Li [1,2,3], Meng-Jie Zhang[1,3], Boxin Zhang[1], Wen-Ping Lin [1], Shu-Jin Li[1], Dian Xiong[1], Qing Wang[1], Wen-Da Wang[1], Qi-Chao Yang[1], Cong-Fa Huang[1], Wei-Wei Deng [1,2] ✉ & Zhi-Jun Sun [1,2] ✉

Tumor tertiary lymphoid structures (TLS), especially mature TLS (mTLS), have been associated with better prognosis and improved responses to immune checkpoint blockade (ICB), but the underlying mechanisms remain incompletely understood. Here, by performing single-cell RNA, antigen receptor sequencing and spatial transcriptomics on tumor tissue from head and neck squamous cell carcinoma (HNSCC) patients with different statuses of TLS, we observe that mTLS are enriched with stem-like T cells, and B cells at various maturation stages. Notably, progenitor exhausted CD4$^+$ T cells, with features resembling follicular helper T cells, support these responses, by activating B cells to produce plasma cells in the germinal center, and interacting with DC-LAMP$^+$ dendritic cells to support CD8$^+$ T cell activation. Conversely, non-mTLS tumors do not promote local anti-tumor immunity which is abundant of immunosuppressive cells or a lack of stem-like B and T cells. Furthermore, patients with mTLS manifest improved overall survival and response to ICB compared to those with non-mTLS. Overall, our study provides insights into mechanisms underlying mTLS-mediated intra-tumoral immunity events against cancer.

Immune checkpoint blockade (ICB) is a paramount form of immunotherapy that has achieved unprecedented success in treating multiple forms of cancer, offering potential for prolonged survival[1]. Following the initial approval of programmed death-1 (PD-1) inhibitors for head and neck squamous cell carcinoma (HNSCC) in 2016, ICB targeting the PD-1/PD-L1 axis, such as pembrolizumab or nivolumab, has been associated with improved survival rates in patients with HNSCC, and has become a primary therapeutic option for recurrent and/or metastatic HNSCC[2]. Unfortunately, only ~20% of patients experience durable clinical advantages from the ICB treatment that targets the PD-1/PD-L1 axis[1,3]. Therefore, the search for predictive biomarkers of ICB response in HNSCC cases, along with efforts to increase response rates to ICB treatments, is becoming the primary focus of research aimed at advancing precision medicine in the field of cancer immunotherapy.

Intra-tumoral tertiary lymphoid structures (TLS) have been proven to be associated with improved outcomes and responses to ICB in cancer patients[4–6]. TLS are ectopic lymphoid formations that develop

[1]State Key Laboratory of Oral & Maxillofacial Reconstruction and Regeneration, Key Laboratory of Oral Biomedicine Ministry of Education, Hubei Key Laboratory of Stomatology, School & Hospital of Stomatology, Frontier Science Center for Immunology and Metabolism, Taikang Center for Life and Medical Sciences, Wuhan University, Wuhan, China. [2]Department of Oral Maxillofacial-Head Neck Oncology, School & Hospital of Stomatology, Wuhan University, Wuhan, China. [3]These authors contributed equally: Hao Li, Meng-Jie Zhang. ✉e-mail: dww@whu.edu.cn; sunzj@whu.edu.cn

in non-lymphoid tissues subjected to chronic inflammation and antigen persistence in autoimmune diseases, chronic infections, graft rejection, and cancers[7]. TLS in tumor microenvironment (TME) usually includes a B cell zone surrounded by a T cell zone composed of a mixture of CD4[+] and CD8[+] T cells and dendritic cells (DCs) as well as natural killer (NK) cells. The presence and density of intra-tumoral TLS correlate with a favorable prognosis in many cancer types[8,9]. Research has identified two primary categories of TLS within tumors according to their maturity[10]. Immature TLS (imTLS) consist of aggregates of T and B cells, with a few present DCs, and are typically linked with T cell exhaustion and an immunosuppressive TME[11]. In contrast, mature TLS (mTLS) is characterized by the presence of a germinal center (GC), which contains T follicular helper (Tfh) cells, follicular dendritic cells (FDCs), and high endothelial venules. Existing findings suggest the proposition that GC-containing mTLS, rather than imTLS without a GC, are associated with clinical benefits in lung cancer[12], hepatocellular cancer[13], colorectal cancer[14], and pancreatic cancer[15]. In the correlation between TLS and therapeutic response, several clinical trials have also examined the positive correlation between the presence of TLS and/or TLS-associated gene signatures and the therapeutic responses to chemotherapy and ICB therapy[8]. Furthermore, the status of TLS is also associated with therapeutic responses, and recent evidence supports the claim that the presence of mTLS is associated with improved response rates of ICB in multiple types of cancer, including HNSCC[16].

However, the mechanism causing mTLS to result in a positive prognosis for cancer patients and a better response to immunotherapy is still unclear. While a recent study has shown that intra-tumoral B cells in mTLS can generate and propagate plasma cells producing anti-tumor antibodies[17], it remains unclear whether the induction of an anti-tumor CD8[+] T cell response, which is most conducive to ICB treatment, can occur in the local TME, particularly at the TLS site. Furthermore, it remains ambiguous which immune cell subsets, such as macrophages, DCs, and T cells, participate and regulate immunity in mTLS induction.

In this study, we explore the function of mTLS as a center for the initiation of local immunity in the TME. We observe that mTLS is enriched with stem-like CD4[+] and CD8[+] T cells and the presence of B cells at different maturation stages, supporting the generation of B and T cell responses. Additionally, CD4[+] progenitor exhausted T cells (Tex[prog]), which share features with Tfh such as CXCL13, PD-1, IL6ST, and TCF1, can activate the B cells to generate plasma cells in the GC of the B cell zone, and support the activation of stem-like CD8[+] cells in collaboration with mature DCs enriched in immunoregulatory molecules (mregDCs) in the T cell zone. Our study suggests that mTLS facilitates an anti-tumor immunity cycle within local TME and may support the induction of mTLS as a strategy to improve therapeutic outcomes in cancer treatment.

## Results

### Study design and overview of the study cohort

To analyze the impact of TLS on the local TME, we collected primary tumor samples mainly from oral site in HNSCC, including fresh samples ($n = 14$) and formalin-fixed paraffin-embedded (FFPE) samples ($n = 422$) from HNSCC patients (Fig. 1a, Supplementary Fig. 1 and Supplementary Data 1 and 2). For each case, the TLS status was detected and assessed through hematoxylin and eosin (H&E) staining, immunohistochemistry (IHC) staining, and multiplex IHC (mIHC), followed by annotation by two certified pathologists who were blinded to the clinical data. TLS status was determined by the aggregation of CD20[+] and CD3[+] cells, while maturity was assessed based on the presence of CD23[+] cells within TLS (Supplementary Fig. 2). In order to further verify TLS maturity, the status was subsequently reconfirmed by mIHC staining of CD20, BCL6, CD23, CD4, CD8, and TCF1. mIHC further verified the TLS status that mTLS is characterized by GC formation, a dynamic region with a network of CD23[+] FDCs and CD20[+]BCL6[+] B cells (Supplementary Figs. 3–5)[6,10]. Among the fresh samples collected from primary tumors of HNSCC

patients, three TLS statuses were determined as TME without TLS (nTLS, $n = 4$), imTLS ($n = 4$), and mTLS ($n = 6$) based on the appearance and maturity of TLS. Each tumor sample was divided into three parts, and each part was subjected to different treatments and analyses, including: (1) scRNA-seq for whole tumor cells; (2) magnetic-activated cell sorting (MACS) targeting CD45[+] cells in tumor, followed by paired scRNA-seq and scTCR/BCR-seq; and (3) spatial transcriptomic sequencing for frozen tumor samples (Fig. 1a).

We obtained single-cell data of 248,336 cells after initial quality control and constructed a cell-type atlas with 11 broad cellular lineages spanning endothelial cells (ECs), lymphatic ECs, stromal cells (pericytes, fibroblasts), lymphoid cells (T/NK cells, B cells, plasma cells), myeloid cells, neutrophils and cancer cells (Fig. 1b, Supplementary Figs. 6a–f and 7, and Supplementary Data 3). All of the single cells were separated from a set of 14 tumor biopsies, from different treatments, using unsorted tumor tissue and sorted CD45[+] cells (Fig. 1c). The cells included unsorted tumor cells and sorted CD45[+] cells, and were classified into three different types based on TLS status (Fig. 1d, e).

Clonotype analysis by scTCR/BCR-seq revealed heterogeneity among immune cell subsets with diverse TCR ($n = 41,806$) and BCR ($n = 10,208$) after excluding ambiguous ($n = 2,178$) and multichain ($n = 122$) clonotypes (Fig. 1f). The immune cell composition varied across TLS status, with higher frequency of T/NK cells and B cells in mTLS compared to non-mTLS (Fig. 1g and Supplementary Fig. 6g–l), consistent with prior reports[17]. Interestingly, we observed that mast cells were scarce in nTLS but highly enriched in mTLS (Fig. 1g). While the relationship between mast cells and TLS has not been previously reported, recent research suggests that mast cells may recruit cytotoxic T cells through the CCL2–CCR2 axis[18]. This finding indicates that mast cells enriched in mTLS may possess similar pro-inflammatory and chemotactic properties. A higher density of stromal area and expression of several immune cell markers were enriched in mTLS compared to non-mTLS, including CD20, CD8, CD4, CD23, and TCF1, which were identified via H&E and IHC staining (Fig. 1h). Additionally, we performed a gene enrichment analysis of spatial transcriptomics data to confirm the TLS status using the gene sets of both the classical 12 chemokine signature[19] and the newly reported TLS imprint[17]. The results show that both gene sets are able to determine the presence of TLS, but when identifying mTLS, the TLS imprint signature performed better (Fig. 1i). These findings also support our previous assessment of TLS status in the TME.

### Accumulation of stem-like CD8[+] and CD4[+] T cells in mTLS

As a central component of the TLS structure, the cell state and function of T cells play a pivotal role in TLS induction and the mounting of TLS-mediated anti-tumor immunity. To analyze the distinct states and functional heterogeneity of T cells within TLS, we identified 5 major T and NK clusters with a total of 28 subclusters (Fig. 2a), broadly defining CD8[+] T cells (clusters 1–5), cycling T/NK cells (clusters 6–8), conventional CD4[+] T cells (Tconv, clusters 9–14), regulatory CD4[+] T cells (Tregs, clusters 15–21), NK cells (clusters 22–26), and other minor clusters including γδT cells (cluster 27) and innate lymphoid cells (ILCs, cluster 28). Clusters were annotated based on known marker genes and cross-referenced with other published annotations[20–22] (Supplementary Fig. 8 and Supplementary Data 3). The differentiation of CD4[+] exhausted T cells (Tex) into two distinct subclusters, namely progenitor exhausted (Tex[prog]/Tfh) and terminally exhausted (Tex[term]) types, was observed (Fig. 2a, b). Conversely, CD8[+] Tex cells have been identified as comprising a single exhausted subcluster (Fig. 2a, b).

T and NK cell clusters followed a gradient across uniform manifold approximation and projection (UMAP) space (Fig. 2b), highlighting TLS-specific phenotypic differences that were quantified by fitting a generalized linear model (GLM) of cluster composition (Fig. 2c). Comparisons of TLS statuses within tumor samples showed that naive/central memory and effector memory phenotypes of CD8[+] T

cells (clusters 1 and 2), which highly expressed memory/stem-like markers including *TCF7*, *CCR7* and *IL7R*, were enriched in mTLS samples and depleted in imTLS and nTLS samples. In contrast, CD8[+] Tex

cells (cluster 4) were more closely associated with imTLS and nTLS (Fig. 2c). Among the CD4[+] T subclusters, almost all phenotypes (clusters 9–13) were found to be more enriched in mTLS, moderately

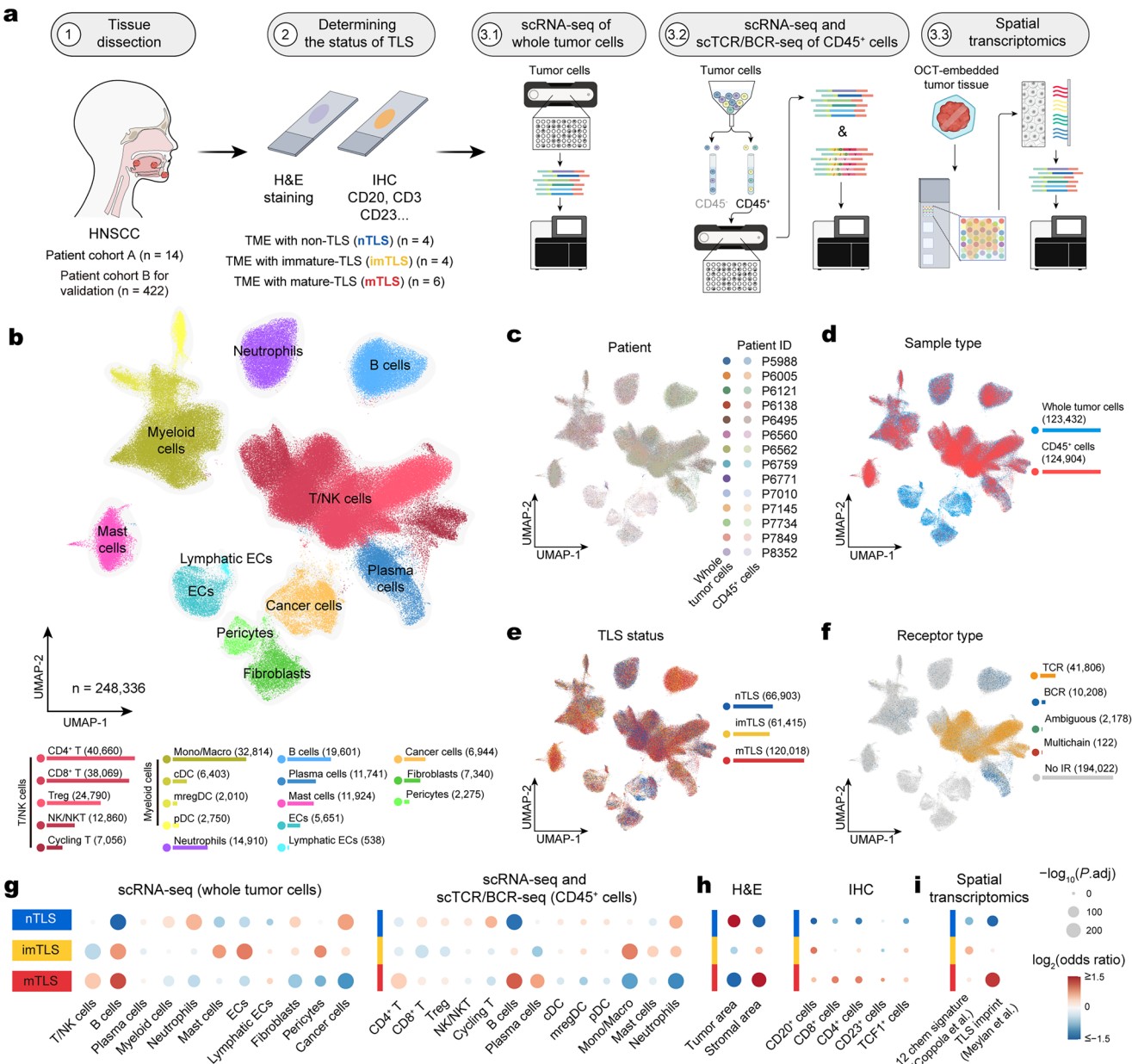

**Fig. 1 | TLS-associated immune landscape in HNSCC. a** Workflow shows the collection and processing of fresh samples of primary oral cavity HNSCC tumors for scRNA-seq, scTCR/BCR-seq, and spatial transcriptomics repertoire analysis. The collection of fresh tumor samples from HNSCC patients ($n = 14$ independent samples) was used to identify the status of TLS by IHC staining and classified into nTLS ($n = 4$ independent samples), imTLS ($n = 4$ independent samples), and mTLS ($n = 6$ independent samples). Subsequently, the fresh tumor sample was divided into three parts and subjected to different processes, including: (1) scRNA-seq of whole tumor cells; (2) MACS targeting CD45[+] cells in tumor, followed by paired scRNA-seq and scTCR/BCR-seq; and (3) spatial transcriptomic sequencing for the tumor sample. **b** UMAP plot of 248,336 cells profiled by all scRNA-seq, colored by cell types. The 11 broad cellular lineages contain ECs, lymphatic ECs, stromal cells (pericytes, fibroblasts), lymphoid cells (T/NK cells, B cells, plasma cells), myeloid cells, neutrophils, and cancer cells. Below the UMAP plot, the number of identified cells for each cell type is included. **c** UMAP plot colored by patients ($n = 14$ independent samples) from whole tumor cells or sorted CD45[+] cells profiled by scRNA-seq. **d** UMAP plot colored by origin of the cells and the number of cells,

either from whole tumor cells ($n = 123,432$) or sorted CD45[+] cells ($n = 124,904$). **e** UMAP plot colored by nTLS (66,903), imTLS (61,415), and mTLS (120,018)– three different TLS statuses profiled by scRNA-seq. **f** UMAP plot colored by TCR/BCR information profiled by scTCR/BCR-seq. The cell types for scRNA-seq of whole tumor cells and the composition difference of CD45[+] cells (**g**), H&E and IHC of tumor slides (**h**), and spatial transcriptomics of tumor tissue (**i**) in nTLS, imTLS, and mTLS statuses. The analysis was conducted using a generalized linear model (GLM) with a binomial distribution and a logit link function. Estimated marginal means and contrasts were computed with *P* values indicating the statistical significance of the observed differences. The *P* values were adjusted using the Bonferroni correction method. A color gradient, transitioning from red (indicating enrichment) to blue (signifying depletion), encodes the log$_2$-transformed odds ratios, while the sizes of the depicted points are governed by the Bonferroni-adjusted $-\log_{10}$(*P* values), accentuating the statistical significance of observed variations. TLS tertiary lymphoid structures, OCT optimal cutting temperature, H&E hematoxylin and eosin, IHC immunohistochemistry, ECs endothelial cells.

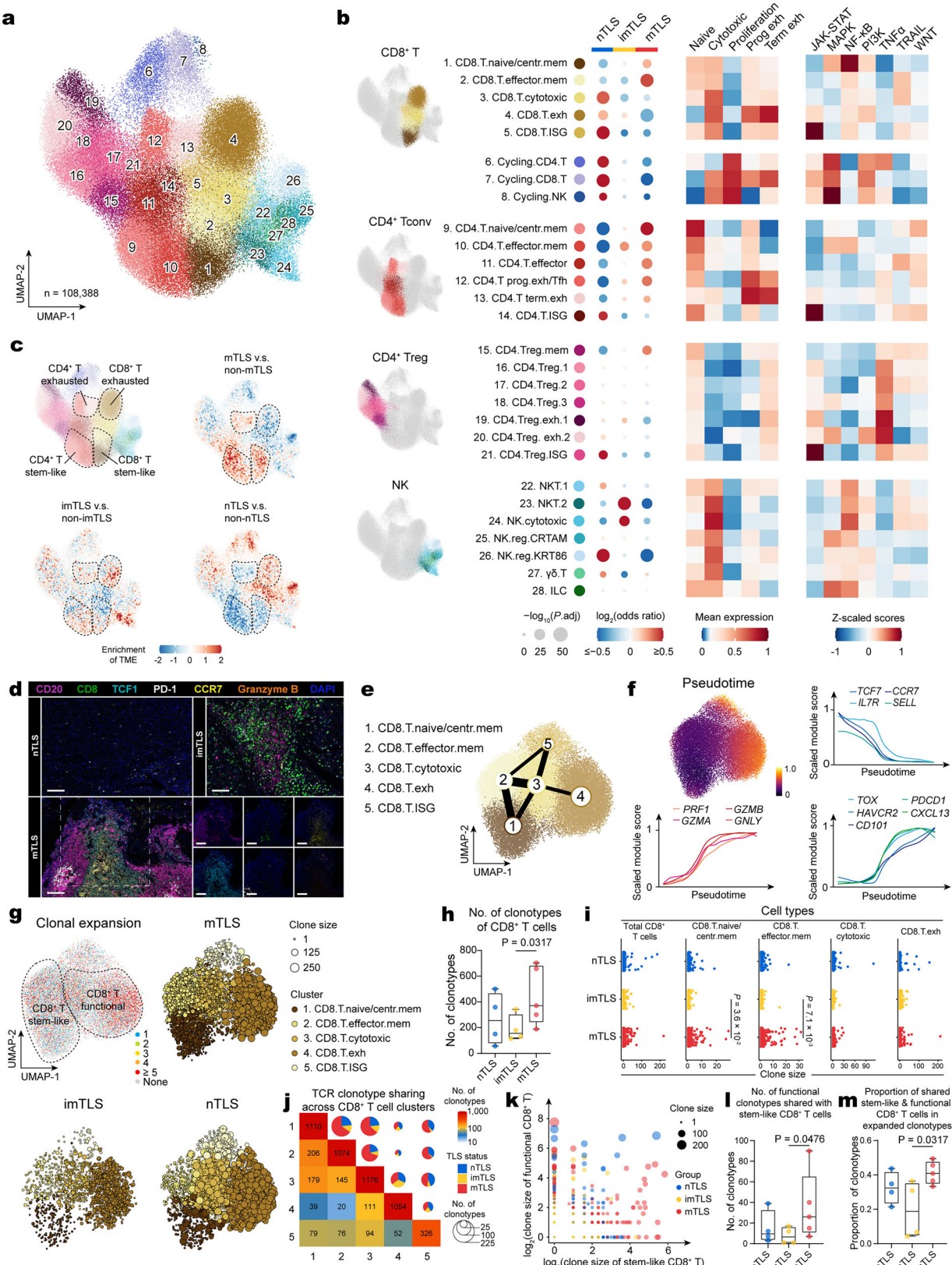

enriched in imTLS, and less abundant in nTLS, except for cluster 14, referring to interferon-stimulated gene (ISG) T cell phenotype (Fig. 2b, c). On the other hand, Treg clusters (clusters 15–21) were found to be reduced in mTLS compared to non-mTLS (nTLS/imTLS), except for the memory Treg (Fig. 2b, c). NK cells (clusters 22–26) were more abundant in nTLS and imTLS, with varying levels of activity

among the different NK cell phenotypes (Fig. 2b, c). mIHC-stained images confirm that memory (CD8+CCR7+) and stem-like (CD8+TCF1+) CD8+ T cell phenotypes are enriched in mTLS (Fig. 2d and Supplementary Figs. 9 and 10). The enrichment analysis revealed that clusters with naive or memory characteristics (clusters 1, 2, 9–10, 15) were enriched in the naive T cell signature, while those with effector features

**Fig. 2 | TLS-associated heterogeneity of T and NK cell states in HNSCC. a** UMAP plot of T/NK cell subclusters identified from scRNA-seq. Subclusters are numbered and colored by identity: CD8+ T cells (clusters 1–5), cycling T/NK cells (clusters 6–8), CD4+ T cells (clusters 9–14), Treg cells (clusters 15–21), NK cells (clusters 22–26), γδT cells (cluster 27), and ILCs (cluster 28). **b** Left, GLM-based dot plot showing TLS status-specific enrichment of T/NK cell subclusters. The analysis was conducted using a GLM with a binomial distribution and a logit link function. Estimated marginal means and contrasts were computed with *P* values indicating the statistical significance of the observed differences. The *P* values were adjusted using the Bonferroni correction method. A color gradient, transitioning from red (representing enrichment) to blue (representing depletion), encodes the log$_2$-transformed odds ratios, while the sizes of the depicted points are governed by the Bonferroni-adjusted −log$_{10}$(*P* values), highlighting the statistical significance of observed variations. Middle, heatmap of average T cell state scores across T/NK cell subclusters. Right, heatmap of signaling pathway activity scores across T/NK cell subclusters. **c** Pairwise comparisons of kernel density estimates in UMAP space. The color gradient from red to blue indicates decreasing enrichment of T/NK cells and plasma cells in different TLS statuses. **d** Images of a mIHC-stained different subclusters of CD8+ T cells in HNSCC tumor with different TLS status. Regions with a high density of memory (CCR7+CD8+), stem-like (TCF1+CD8+), cytotoxic (Granzyme B+CD8+), and exhausted (PD-1+CD8+) CD8+ T cells. nTLS and imTLS were repeated four times independently with similar results, mTLS were repeated six times independently with similar results. Scale bars = 100 μm. **e** PAGA analysis of CD8+ T cells. Each color represents a subcluster of CD8+ T cell. **f** UMAP plot colored by pseudotime across subclusters of CD8+ T cells. Scaled module scores within these subclusters of CD8+ T cells with respect to pseudotime and stem-like, functional, and exhausted markers. **g** UMAP plot embedding for CD8+ T cell subclusters, colored by clone size (upper left) and UMAP embedding for CD8+ T cell clusters in different TLS status, colored by cluster, indicating the different clone size by dot size. **h** Boxplot showed the number of clonotypes of CD8+ T cells in different statuses of TLS (*n* = 4 independent samples nTLS, *n* = 4 independent samples imTLS, *n* = 5 independent samples mTLS, one-tailed Mann–Whitney *U*-test; for box plots: box center line, median; box limits, upper and lower quartiles; box whiskers, maximum and minimum values). **i** Clone size in each CD8+ T cell subclusters separated by TLS status (two-tailed Mann–Whitney *U*-test). **j** Heatmap and pie chart showing the number of clonotypes shared between neighboring functional clusters of CD8+ T cells in different statuses of TLS. For the heatmap, the color represents the number of shared clonotypes. For the pie chart, the color represents the status of TLS, and the size represents the number of shared clonotypes. **k** Dot plot showing the overlaps of shared clonotypes between stem-like and functional CD8+ T cells, sized by number of clonotypes and colored by TLS status. **l** Boxplot showing the number of functional clonotypes shared with stem-like CD8+ T cells in different status of TLS (*n* = 4 independent samples nTLS, *n* = 4 independent samples imTLS, *n* = 5 independent samples mTLS, one-tailed Mann–Whitney *U*-test; for box plots: box center line, median; box limits, upper and lower quartiles; box whiskers, maximum and minimum values). **m** Boxplot showing the proportion of shared stem-like and functional features in expanded clonotypes across different statuses of TLS (*n* = 4 independent samples nTLS, *n* = 4 independent samples imTLS, *n* = 5 independent samples mTLS, one-tailed Mann–Whitney *U*-test; for box plots: box center line, median; box limits, upper and lower quartiles; box whiskers, maximum and minimum values). Tconv conventional T cell, NK natural killer, mIHC multiplex IHC, ISG interferon-stimulated gene, ILC innate lymphoid cell, Tfh T follicular helper cell, Treg regulatory T cell.

(clusters 3, 22–28) were enriched in the cytotoxic T cell signature category (Fig. 2b). Clusters annotated as cycling (clusters 6–8) were enriched in the proliferation T cell signature category, and exhausted clusters exhibiting varying degrees of exhaustion-related gene expression (clusters 4, 12, 13, 16–20) were enriched in different exhausted categories (Fig. 2b). The cell pathway activity scores demonstrated differences in signaling pathway activities across clusters. T cell subsets with high expression of ISG (clusters 5, 14, 21) showed enhanced activity in the JAK-STAT pathway, while cycling T/NK cells (clusters 6–8) exhibited increased activity in the MAPK pathway (Fig. 2b), consistent with previous reports[23]. Treg subsets (clusters 16–21) displayed enhanced activity in the TNFα pathway (Fig. 2b), aligning with earlier findings[24].

**Generation of effector-molecule-expressing cytotoxic CD8+ T cells from stem-like CD8+ T cells within mTLS**
To evaluate the infiltration of CD8+ T cells, we first performed differentiation trajectories, which identified subclusters of CD8+ T cells at different developmental stages. These subclusters demonstrated a progression from memory/stem-like phenotypes towards an exhausted phenotype, as evidenced by follow-up partition-based graph abstraction (PAGA) analysis (Fig. 2e). This transition was marked by a decrease in naive/stem-like T cell markers (*TCF7*, *CCR7* and *SELL*) and an increase in cytotoxicity-associated markers (*PRF1*, *GZMB*, *GZMA*, and *GNLY*). Additionally, exhaustion-associated markers (*TOX*, *PDCD1*, *HAVCR2*, *CXCL13*, and *CD101*) were found to increase during the transition, as shown by pseudotime analysis (Fig. 2f). This aligns with previous studies that have reported that exhaustion occurs after T cell activation[20,25].

Next, we utilized paired scRNA-seq and scTCR/BCR-seq for clonotype analysis of CD8+ T cells. We defined TCR clonotypes based on paired α and β chains (*TRA* and *TRB*) to track clonal cell fates. Differential clonal diversity, measured by Shannon entropy and the D50 index, was revealed among CD8+ T cell clusters, and CD8+ Tex cells were observed to have the least clonal diversity (Supplementary Fig. 11a, b). The TCR clone information was mapped to the UMAP to facilitate the visualization of clone size and its distribution across different subclusters and TLS statuses (Fig. 2g). mTLS status harbored a larger number of expanded clonotypes of CD8+ T cells (Fig. 2g, h and Supplementary Fig. 11c–f). In concordance with the result of enrichment analysis, the naive/central memory and effector memory phenotypes of CD8+ T cells showed enrichment for larger clonotype sizes in mTLS (Fig. 2i). We also used a heatmap to show the sharing of clonotypes among clusters of CD8+ T cells (Fig. 2j) and noted a significant sharing of clonotypes among neighboring functional clusters of CD8+ T cells (clusters 1–4), consistent with the trajectory established by PAGA analysis (Fig. 2e).

To investigate whether cytotoxic CD8+ T cells arise within the local TME, we categorized clusters 1 and 2, characterized by elevated expression of memory/stem-like molecules (*TCF7*, *CCR7*, and *IL7R*), as possessing a "stem-like" phenotype[26,27]. Subsequently, we characterized clusters 3 and 4, noted for their increased expression of cytotoxic molecules (*PRF1*, *GZMA*, *GZMB*, and *GNLY*), as embodying a "functional" phenotype. Notably, we observed a high degree of overlap of shared clonotypes between the stem-like and functional CD8+ T cell phenotypes within mTLS, indicating the existence of common pathways from stem-like to functional CD8+ T cell phenotypes within mTLS (Fig. 2k, l and Supplementary Fig. 11g, h). Consistent with these findings, an increased frequency of shared clonotypes between stem-like and functional phenotypes was identified amongst the expanded clonotypes of CD8+ T cells within mTLS (Fig. 2m). Gene enrichment analysis showed that stem-like clusters of CD8+ T cells enriched in mTLS upregulated DC chemotaxis (Supplementary Fig. 11i). This suggested the ability of these stem-like CD8+ T cells to attract and interact with DCs, potentially activating a CD8+ T cell response[28]. In summary, mTLS was associated with more enrichment of stem-like CD8+ T cell phenotypes, and mature CD8+ T cells toward functional CD8+ T cells occur, suggesting mTLS can generate an intra-tumoral CD8+ T cell response.

**Predominance of CD4+ Tex^prog/Tfh cells over Tregs within mTLS**
Next, we sought to elucidate the association between TLS statuses and CD4+ T cells in the TME. The differentiation trajectories projected CD4+ T cells along two axes of Tex (clusters 9–14) and Treg phenotypes (clusters 15–21) (Fig. 3a), as defined by the transcription factors *TOX* and *FOXP3*[21] (Fig. 3b). Pseudotime analysis showed that both differentiation trajectories showed a decrease in stem-like markers (*CCR7*,

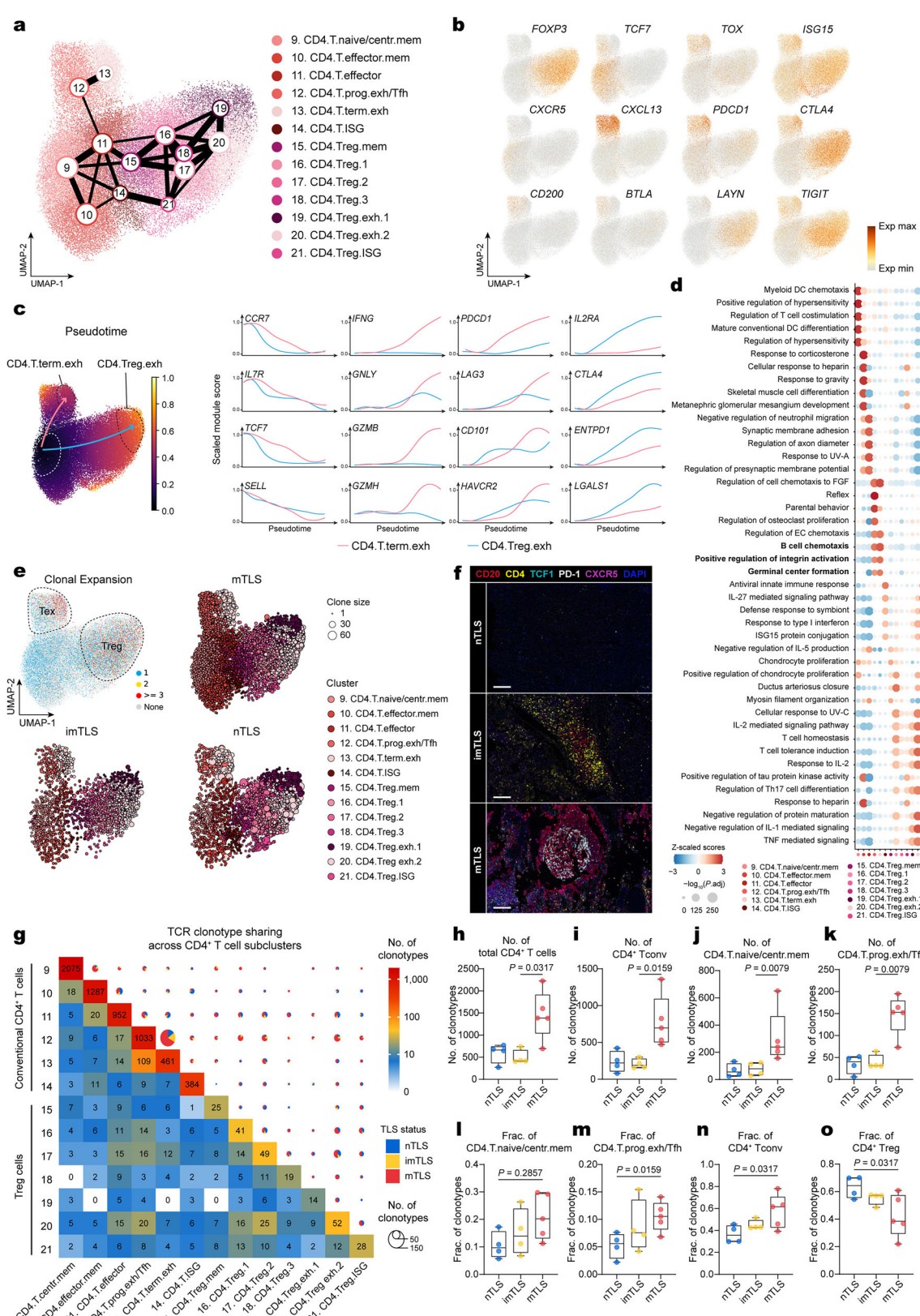

*IL7R, TCF7,* and *SELL*) and an increase in exhaustion-associated markers (*PDCD1, LAG3, CD101,* and *HAVCR2*) (Fig. 3c). The exhaustion pathway was accompanied by higher expression of cytotoxicity (*IFNG, GNLY, GZMB,* and *GZMH*), while the Treg pathway demonstrated a reduction in cytotoxic markers alongside an increase in the expression of immune inhibitory and immunosuppressive molecules (*IL2RA, CTLA4,*

*ENTPD1,* and *LGALS1*) which are consistent with the immunosuppressive function of Treg clusters (Fig. 3c, d).

Subsequently, we also used TCR clonotype analysis to characterize the phenotype and TCR clonality across all CD4+ T cell clusters and TLS statuses (Supplementary Fig. 12a–d). The phenotypes of Tex and Treg occupied a larger clonal expansion (Fig. 3e). Differential clonal

**Fig. 3 | TLS-associated heterogeneity of CD4⁺ T cell states in HNSCC. a** PAGA analysis of CD4⁺ T cells. Each color represents a subcluster of CD4⁺ T cells. **b** UMAP plots showing normalized expression profiles of cell-type-specific markers in CD4⁺ T cell subclusters. **c** Left, UMAP plot showing pseudotime trajectories of CD4⁺ T cell clusters, revealing two potential differentiation trajectories: exhaustion and Treg. Right, scaled module scores within these subclusters of CD4⁺ T cells with respect to two pseudotime trajectories and stem-like, functional, exhausted, and immunosuppressive markers. **d** Dot plot showing the analysis of enrichment for the four most significant Gene Ontology (GO) Biological Process terms across CD4⁺ T cell subclusters. A color gradient, transitioning from red (representing enrichment) to blue (representing depletion), encodes the $Z$-score normalized enrichment score, while the sizes of the depicted points are governed by the Benjamini–Hochberg-adjusted $-\log_{10}(P$ values), highlighting the statistical significance of observed variations. Benjamini–Hochberg-adjusted $P$ values were obtained by a two-tailed Wilcoxon rank-sum test. **e** UMAP plot embedding for CD4⁺ T cell subclusters, colored by clone size (upper left) and UMAP embedding for CD4⁺ T cell subclusters in different TLS status, colored by subcluster, indicating the different clone size by dot size. **f** Images of a mIHC-stained different subcluster of CD4⁺ T cells in HNSCC tumor with TLS status. Regions with a high density of central memory (CXCR5⁺CD4⁺) CD4⁺ T cells and

progenitor exhausted (TCF1⁺PD-1⁺CD4⁺) CD4⁺ T cells. nTLS and imTLS were repeated four times independently with similar results, mTLS were repeated six times independently with similar results. Scale bars = 100 μm. **g** Heatmap and pie chart showing the number of clonotypes shared between neighboring functional clusters of CD4⁺ T cells in different statuses of TLS. For the heatmap, the color represents the number of shared clonotypes. For the pie chart, the color represents the status of TLS, and the size represents the number of shared clonotypes. **h–k** Box plots showing the number of clonotypes of total CD4⁺ T cell, CD4⁺ Tconv cell, naive/central memory CD4⁺ T cell, and progenitor exhausted CD4⁺ T cell in different TLS status ($n = 4$ independent samples nTLS, $n = 4$ independent samples imTLS, $n = 5$ independent samples mTLS, one-tailed Mann–Whitney $U$-test; for box plots: box center line, median; box limits, upper and lower quartiles; box whiskers, maximum and minimum values). **l–o** Boxplot showing fraction clonotypes of naive/central memory CD4⁺ T cell, progenitor exhausted CD4⁺ T cell, CD4⁺ Tconv cell (clusters 9–13) and Treg (clusters 15–20) in total CD4⁺ T cell clonotypes across different TLS status ($n = 4$ independent samples nTLS, $n = 4$ independent samples imTLS, $n = 5$ independent samples mTLS, one-tailed Mann–Whitney $U$-test; for box plots: box center line, median; box limits, upper and lower quartiles; box whiskers, maximum and minimum values). Tconv conventional T cell, Treg regulatory T cell, Tfh T follicular helper cell.

diversity was revealed among CD4⁺ T cell subclusters, and CD4⁺ Tex^prog/Tfh were observed to have the least clonal diversity (Supplementary Fig. 12a). Consistent with the analysis of differentiation trajectories, significant clonotypes sharing between neighboring functional clusters of CD4⁺ T cells were observed (clusters 9–21) (Fig. 3g). Additionally, we noted a higher number of clonotype sharing overlap between progenitor and terminal Tex compared to other clusters, and mTLS accounted for the majority of these shared clonotypes and showed a slight increase in the clusters of CD4⁺ Tex (Fig. 3g, Supplementary Fig. 12c). Interestingly, CD4⁺ Tex^term phenotype exhibited the highest expression of cytotoxic markers (*PRF1*, *GZMA*, *GZMB*, *IFNG*, and *GNLY*) (Supplementary Fig. 12e), aligning with recent discoveries indicating that cytotoxic CD4⁺ T cells within tumors can directly kill cancer cells[29,30]. Moreover, recent studies have also shown that CD4⁺ T cells have the ability to promote stem-like TCF1⁺CD8⁺ T cell responses through IFNγ production[27]. This evidence is further supported by a high level of overlap in clonotype sharing between CD4⁺ Tex^term and CD4⁺ Tex^prog/Tfh (Fig. 3g), further supporting the role of CD4⁺ T cells in promoting CD8⁺ T cell immune responses. mIHC-stained images also confirmed that CD4⁺ Tex^prog/Tfh (CD4⁺TCF1⁺PD-1⁺) were enriched in mTLS, particularly in the GC of the mTLS (Fig. 3f).

Meanwhile, mTLS accumulated the clonotypes of total CD4⁺ T cells and subclusters of CD4⁺ Tconv cells along the exhaustion trajectory, including naive/central memory and progenitor exhausted clusters (Fig. 3h–n). On the contrary, nTLS had a larger clone size and made up the major proportion of the clusters of Treg (Fig. 3e–o), while imTLS developed along both pathways of Tex and Treg and was in between mTLS and nTLS (Fig. 3l–o). Gene enrichment analysis revealed that the CD4⁺ Tex^prog/Tfh subset was significantly enriched with features of Tfh-like functions, including B cell chemotaxis and GC formation (Fig. 3d), which is consistent with previously reported studies[22,31]. In conclusion, mTLS demonstrated greater enrichment of CD4⁺ T cells predisposed to differentiate towards exhaustion phenotypes. This includes the CD4⁺ Tex^prog/Tfh cell subtype, which is characterized by Tfh-like functions crucial for B cell chemotaxis, immune cell maturation, and GC formation.

### Promotion of B cell maturation and plasma cell generation by mTLS

We then subclustered 27,622 cells that were manually annotated as B cells and plasma cells, subsequently obtaining 9 distinct subclusters based on RNA expression profiles of established markers (Fig. 4a, Supplementary Fig. 13 and Supplementary Data 3). These subclusters were further cross-referenced with annotations from other published studies[32,33] (Supplementary Fig. 14). We identified all major stages of B

cell maturation in HNSCC, including naive B cells (cluster 1), memory B cells (clusters 2 and 3), tissue-resident Fc receptor like 4 positive (FCRL4⁺) B cells (cluster 4), GC B cells (cluster 5), ISG B cells (cluster 6), and plasmablasts (cluster 7), and plasma cells (clusters 8 and 9) (Fig. 4b). Naive B cells were characterized by a high expression of *IGHM*, *IGHD* and *FCER2* (CD23), *CD72* and *CD200* (Fig. 4b, d). The two memory B cell clusters showed high expression of the classical *CD27* gene for human memory B cells, but differed in that one had a higher expression of the activation marker of *CD69* (Fig. 4d and Supplementary Fig. 13d). Additionally, GC B cells were annotated with higher levels of *BCL6*, *LMO2*, *RGS13* and *HMCES* that may increase their ability to reside in the GC or differentiate to plasma cells[33] (Fig. 4d and Supplementary Fig. 13d). We also identified tissue-resident FCRL4⁺ cells in HNSCC (Fig. 4d), which are rare in the marginal zone of B cell follicles in the spleen and lymph nodes[33,34]. The plasmablasts showed the highest expression of *MKI67* as well as other cell cycle-related genes, underscoring their proliferative capacity (Fig. 4d and Supplementary Data 3).

We next explored the differences in B cell subclusters among various TLS statuses. Although mTLS and imTLS enriched B cells as expected (Fig. 4b, c), the subclusters of these B cells were different. We observed that all major stages of B cell maturation, including naive B cells, memory B cells, and GC B cells, were significantly enriched in mTLS (Fig. 4b, c). Additionally, imTLS were found to be abundant in many B cells, including ISG B cells and FCRL4⁺ B cells, but not many plasma cells (Fig. 4b, c). This suggests that the B cells in imTLS do not possess the ability to produce plasma cells or that this process is hindered by an obstacle. Interestingly, nTLS showed minimal numbers of B cells, except for an enrichment of plasma cells, suggesting that these plasma cells may have originated from outside the TLS (Fig. 4b, c). Differential expression analysis revealed increased gene expression related to activation/effector molecules, B cell-mediated immunity, antigen presentation, and cell migration in B cells between mTLS and non-mTLS (nTLS/imTLS) (Fig. 4h). In line with this, gene enrichment analysis showed that the mTLS-enriched GC B cells exhibit immunoglobulin (Ig) production and antigen presentation through major histocompatibility complex class II (Fig. 4e).

Subsequently, we conducted a trajectory inference to better understand the differentiation trajectories of B cells and plasma cells. Our results showed a pseudotemporal ordering of naive B cells, memory B cells, GC B cells, plasmablasts, and plasma cells (Fig. 4f), which is consistent with previous reports on the generation of B cell immunity within lymphoid follicles[33]. Additionally, the differentiation trajectory of B cells involves deletional recombination of the IgM and IgD constant domain genes (*IGHM* and *IGHD*) and the expression of

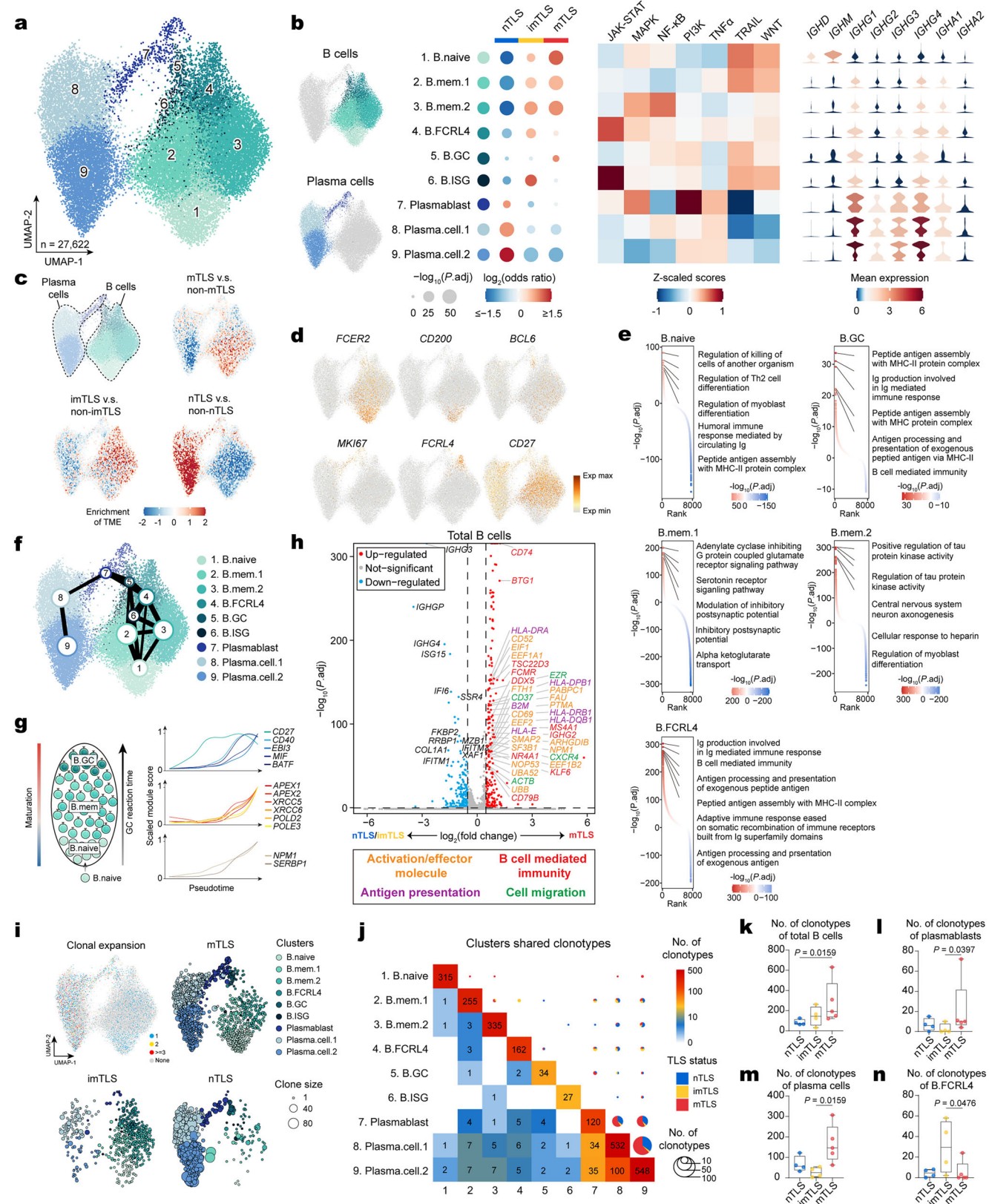

different downstream constant domain genes (*IGHG1*, *IGHG2*, *IGHG3*, *IGHG4*, *IGHA1*, and *IGHA2*) (Fig. 4b)[35]. The pseudotime analysis of B cell clusters showed increased expression of genes associated with activation (*CD27*, *CD40*, *EBI3*, *MIF*, and *BATF*), as well as genes associated with the class switch recombination machinery (*APEX1*, *APEX2*, *XRCC5*, *XRCC6*, *POLD2*, and *POLE3*) and class switch recombination interactors

(*NPM1*, *SERBP1*) (Fig. 4g). These findings are consistent with previously reported roles of these genes in B-cell activation[33].

Next, we performed BCR clonotype analysis of B cells and plasma cells, defining BCR clonotypes through the sequencing of *IGH*, *IGK*, and *IGL* (Supplementary Fig. 13a–c). The UMAP incorporates the BCR clone data to facilitate visualization of clone size and distribution across

**Fig. 4 | TLS-associated heterogeneity of B cell states in HNSCC. a** UMAP plot of B cell subclusters identified from scRNA-seq. Subclusters are numbered and colored by identity: B cells (clusters 1–6) and plasma cells (clusters 7–9). **b** Left, GLM-based dot plot showing TLS status-specific enrichment of B cell subclusters. The analysis was conducted using a GLM with a binomial distribution and a logit link function. Estimated marginal means and contrasts were computed with $P$ values indicating the statistical significance of the observed differences. The $P$ values were adjusted using the Bonferroni correction method. A color gradient, transitioning from red (representing enrichment) to blue (representing depletion), encodes the $\log_2$-transformed odds ratios, while the sizes of the depicted points are governed by the Bonferroni-adjusted $-\log_{10}(P$ values), highlighting the statistical significance of observed variations. Middle, heatmap of signaling pathway activity scores across B cell and plasma cell subclusters. Right, heatmap of normalized expression of genes associated with immunoglobulin heavy chain constant region, which correlate with types of Ig across B cell subclusters. **c** Pairwise comparisons of kernel density estimates in UMAP space. The color gradient from red to blue indicates decreasing enrichment of B cells and plasma cells in different TLS statuses. **d** UMAP plots showing normalized expression profiles of cell-type-specific markers in B cell subclusters. **e** Ranking of GO Biological Process terms in subclusters of B cells. A color gradient, transitioning from red (representing enrichment) to blue (representing depletion), encodes the Benjamini–Hochberg-adjusted $-\log_{10}(P$ values), highlighting the statistical significance of observed variations. Benjamini–Hochberg-adjusted $P$ values were obtained by a two-tailed Wilcoxon rank-sum test. **f** PAGA analysis of B cells and plasma cells. Each color represents a subcluster of B cells or plasma cells. **g** Left, Schematic illustrates that as B cells differentiate from naive B cells to GC B cells, the maturation progressively increased, and the GC reaction time was extended. Right, scaled module scores

within these subclusters of B cells with respect to two pseudotime trajectories and genes associated with activation, class switch recombination machinery and class switch recombination interactors. **h** Volcano plot showing differentially expressed genes (DEGs) of total B cells between non-mTLS (imTLS/nTLS) and mTLS statuses (red dots: Benjamini–Hochberg-adjusted $P < 0.05$ and $\log_2$(fold change) $> 10$, gray dots: adjusted $P > 0.05$, blue dots: adjusted $P < 0.05$ and $\log_2$(fold change) $< -10$). Benjamini–Hochberg-adjusted $P$ values were obtained by a two-tailed Wilcoxon rank-sum test. **i** UMAP plot embedding for subclusters of B cells and plasma cells, colored by clone size (upper left) and UMAP embedding for subclusters of B cells and plasma cells in different TLS status, colored by subcluster, indicating the different clone size by dot size. A color gradient, transitioning from red (representing enrichment) to blue (representing depletion), encodes the $Z$-score normalized enrichment score, while the sizes of the depicted points are governed by the Benjamini–Hochberg-adjusted $-\log_{10}(P$ values), highlighting the statistical significance of observed variations. Benjamini–Hochberg-adjusted $P$ values were obtained by a two-tailed Wilcoxon rank-sum test. **j** Heatmap and pie chart showing the number of clonotypes shared between neighboring functional subclusters of B cells and plasma cells in different statuses of TLS. For the heatmap, the color represents the number of shared clonotypes. For the pie chart, the color represents the status of TLS, and the size represents the number of shared clonotypes. **k–n** Boxplot showing number of clonotypes of total B cells, plasmablasts, plasma cells, and FCRL4$^+$ B cells in different TLS status ($n = 4$ independent samples nTLS, $n = 4$ independent samples imTLS, $n = 5$ independent samples mTLS, one-tailed Mann–Whitney $U$-test; for box plots: box center line, median; box limits, upper and lower quartiles; box whiskers, maximum and minimum values). Ig immunoglobulin, MHC major histocompatibility complex, Th2 T helper 2 cell, GC germinal center.

---

various B cell phenotypes and TLS statuses (Fig. 4i). The analysis indicated a correlation between mTLS status and an enrichment of clones from both B cells and plasma cells (Fig. 4i), and more expanded clonotypes of B cells, plasmablasts, and plasma cells (Fig. 4k–m). The analysis of clonotype sharing showed that plasma cells predominantly originated from memory B cells and plasmablasts (Fig. 4j). This finding, in conjunction with clonotype analysis, suggests that mTLS in HNSCC has the ability to produce plasma cells. Conversely, imTLS was associated with an enrichment of B cells, but rare plasma cells and more expanded clonotypes of FCRL4$^+$ B cells (Fig. 4i, n). This suggests that the B cells with imTLS do not possess the ability to produce plasma cells. Additionally, nTLS was associated with an enrichment of plasma cells, but few B cells (Fig. 4i), which was similar to the enrichment seen in the clusters of CD8$^+$ T cells (CD8$^+$ Tex cells) within nTLS, suggesting that plasma cells in nTLS may have originated from outside the TME.

### Accumulation of DCs in mTLS

Within TLS, while B and T lymphocytes constitute the predominant fraction of immune cells, these structures are also significantly populated by diverse myeloid cells, notably DCs[7]. We further focused on the myeloid cell compartment and re-clustered 70,093 cells into four main clusters of myeloid cells with 16 subclusters, including DCs, macrophages, neutrophils, and mast cells (Fig. 5a and Supplementary Fig. 15)[20]. The DC compartment is comprised of conventional DCs (cDC1s, cDC2s) (clusters 1 and 2), mregDCs (clusters 3), and plasmacytoid DCs (pDCs) (clusters 4) (Fig. 5b). mregDCs were enriched in maturation and immunoregulatory molecules (*CD274*, *CCR7*, *CCL22*, *BIRC3*, *IDO1*, *IL4I1*, and notably *LAMP3*, a highly expressed DC lysosomal-associated membrane glycoprotein (DC-LAMP)) (Fig. 5d and Supplementary Data 3). In addition, seven clusters of classical and alternatively activated macrophages were identified (clusters 5–11), as well as phagocytic macrophages (cluster 12) and cycling macrophages (cluster 13). Neutrophils (clusters 14 and 15) and mast cells (cluster 16) were also identified in the TME of HNSCC. Clusters of myeloid cells differed by expression of MHC-encoding genes (Fig. 5b).

We next explored the differences in myeloid cell subclusters among various TLS statuses. We observed that DC, specifically cDC2, and pDC were significantly enriched in mTLS (Fig. 5b, c). In contrast,

imTLS were found to be abundant in M2-like macrophage subclusters (M2.Macro, M2.MMP9, and M2.MMP12), but contained few DCs (Fig. 5b, c). nTLS were predominantly composed of neutrophils that have previously been demonstrated to have an immunosuppressive function and are associated with poor clinical outcomes[36] (Fig. 5b, c). Interestingly, there appeared to be a correlation between the content of mast cells and the status of TLS, with an increase in mast cell content observed in mTLS (Fig. 5b, c).

The gene enrichment analysis demonstrated that the cDC2 subset, prominently present in mTLS, showed significant enrichment for features associated with the processing and presentation of exogenous antigens, as well as Ig production (Fig. 5e). These findings suggest that cDC2s play a crucial role in regulating antibody responses through the induction of GC responses and plasma cell formation[37].

### Spatial transcriptomics reveals the co-occurrence of cells involved in intra-tumoral T and B cell immunity within the mTLS

To analyze the influence of TLS status on the TME architecture, we used spatial transcriptomics technology from the Visium platform to identify TLS status (Fig. 1i), complementing findings from paired scRNA-seq and scTCR/BCR-seq. Despite the limitations of the Visium v1 platform in providing single-cell resolution data, the Cell2location[38] tool was utilized to spatially map cell types annotated from scRNA-seq datasets within the TME. This method facilitated the identification of a cell cluster associated with TLS, which included B cells, T cells, DCs, and myeloid cells (Fig. 6a–c).

In mTLS, we delineated the progression from the naive B cell to the GC B cell and subsequently to the plasma cell, establishing a continuum from the inner GC of the mTLS to the adjacent tissue (Fig. 6c). And we also discovered a high level of enrichment in GC B cells, plasma cells, exhausted CD4$^+$ T cells, and CD8$^+$ T cells, as well as DCs that were enriched in mTLS (Supplementary Fig. 16). This sequence adheres to the previously outlined developmental pathway of B cells to plasma cells (Fig. 4f). Subclusters of CD4$^+$ Tconv, including naive/central memory CD4$^+$ T cells and Tex$^{prog}$/Tfh cells were observed to be notably enriched within the interior of mTLS (Fig. 6c). Consistent with the above results, CD8$^+$ T cell subclusters were seen to follow a previously established developmental trajectory, presenting a gradient from the

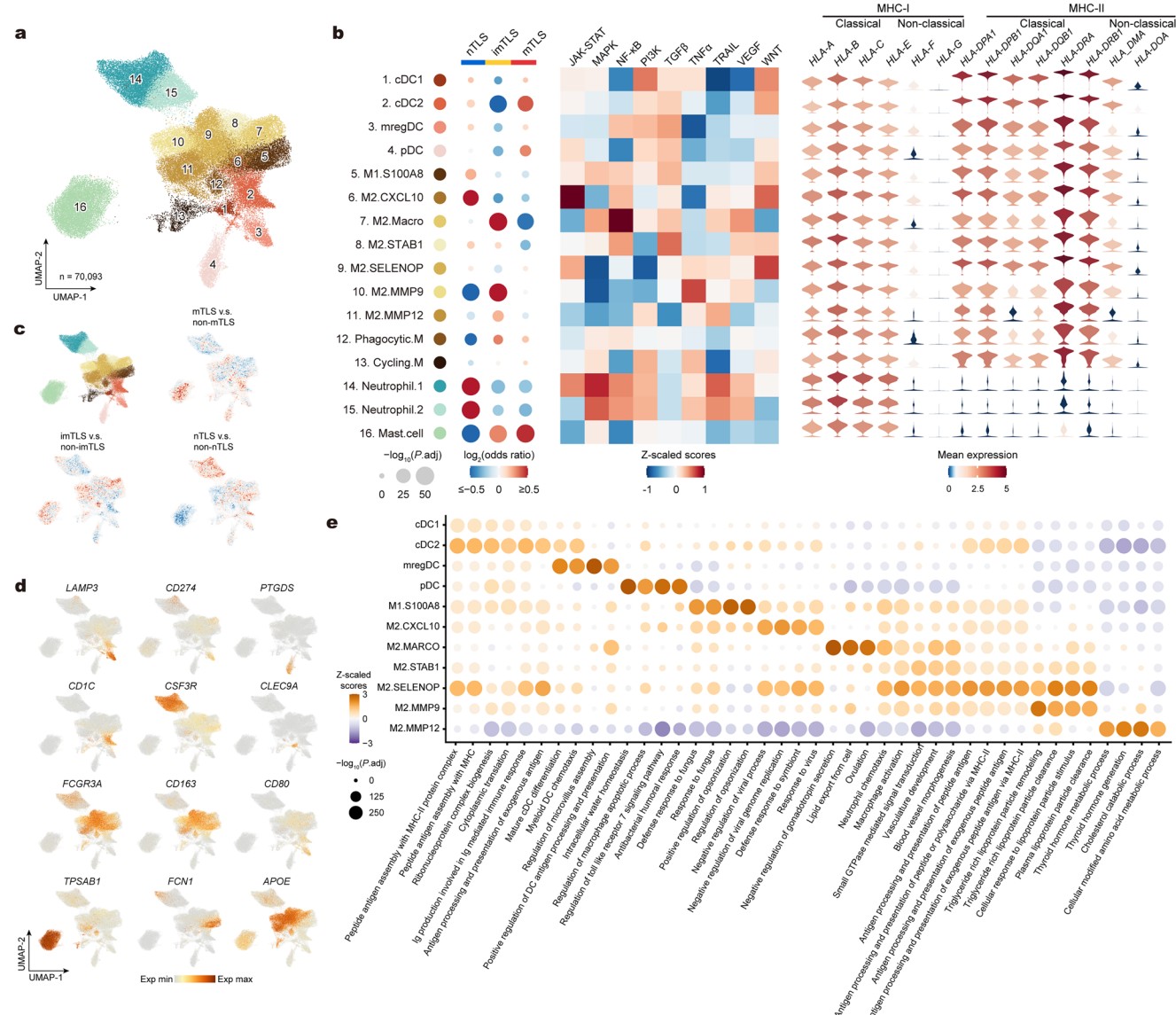

**Fig. 5 | TLS-associated heterogeneity of myeloid cell states in HNSCC. a** UMAP plot of myeloid cell subclusters identified from scRNA-seq. Subclusters are numbered and colored by identity: DCs (clusters 1–4), macrophages (clusters 5–13) and neutrophils (clusters 14 and 15), and mast cells (cluster 16). **b** Left, GLM-based dot plot showing TLS status-specific enrichment of myeloid cell subclusters. The analysis was conducted using a GLM with a binomial distribution and a logit link function. Estimated marginal means and contrasts were computed with *P* values indicating the statistical significance of the observed differences. The *P* values were adjusted using the Bonferroni correction method. A color gradient, transitioning from red (representing enrichment) to blue (representing depletion), encodes the log$_2$-transformed odds ratios, while the sizes of the depicted points are governed by the Bonferroni-adjusted −log$_{10}$(*P* values), highlighting the statistical significance of observed variations. Middle, heatmap of signaling pathway activity scores across myeloid cell subclusters. Right, heatmap of normalized expression of genes

associated with MHC across myeloid cell subclusters. **c** Pairwise comparisons of kernel density estimates in UMAP space. The color gradient from red to blue indicates decreasing enrichment of myeloid cells in different TLS statuses. **d** UMAP plots showing normalized expression profiles of cell-type-specific markers in myeloid cell subclusters. **e** Dot plot showing the analysis of enrichment for the four most significant GO Biological Process terms across myeloid cell subclusters. A color gradient, transitioning from orange (representing enrichment) to purple (representing depletion), encodes the *Z*-score normalized enrichment score, while the sizes of the depicted points are governed by the Benjamini–Hochberg-adjusted −log$_{10}$(*P* values), highlighting the statistical significance of observed variations. Benjamini–Hochberg-adjusted *P* values were obtained by a two-tailed Wilcoxon rank-sum test. Ig immunoglobulin, MHC major histocompatibility complex, DC dendritic cell, cDC conventional DC, mregDC mature DC enriched in immunoregulatory molecules, pDC plasmacytoid DC.

inner GC of mTLS extending towards the adjacent tumor tissue (Fig. 6c), suggesting that mTLS can locally generate anti-tumor T cell immunity. DCs including notably cDC1, cDC2 and mregDC were localized within mTLS (Fig. 6c). Furthermore, co-occurrence analysis supported the close spatial association among B cells, CD4$^+$ T cells, CD8$^+$ T cells and DCs in mTLS (Fig. 6f). While imTLS also contain these cellular types, they do not demonstrate a closely associated spatial distribution (Fig. 6e). Moreover, Tregs were shown to have a closely associated spatial relationship with both CD8$^+$ and CD4$^+$ T cells in

imTLS (Fig. 6e), suggesting Tregs may impede local anti-tumor immunity in the TME. In nTLS, the spatial relationship between these cells is comparatively diminished (Fig. 6d), indicative of the absence of cellular interactions.

To investigate the mechanisms of cell–cell communication within TLS, the CellPhoneDB database[39] was utilized for TLS-associated cell clusters including B cell, T cell and DC to reveal ligand–receptor interactions among annotated clusters. Among all cell clusters, cDC2 exhibited the greatest number of significant interactions among other

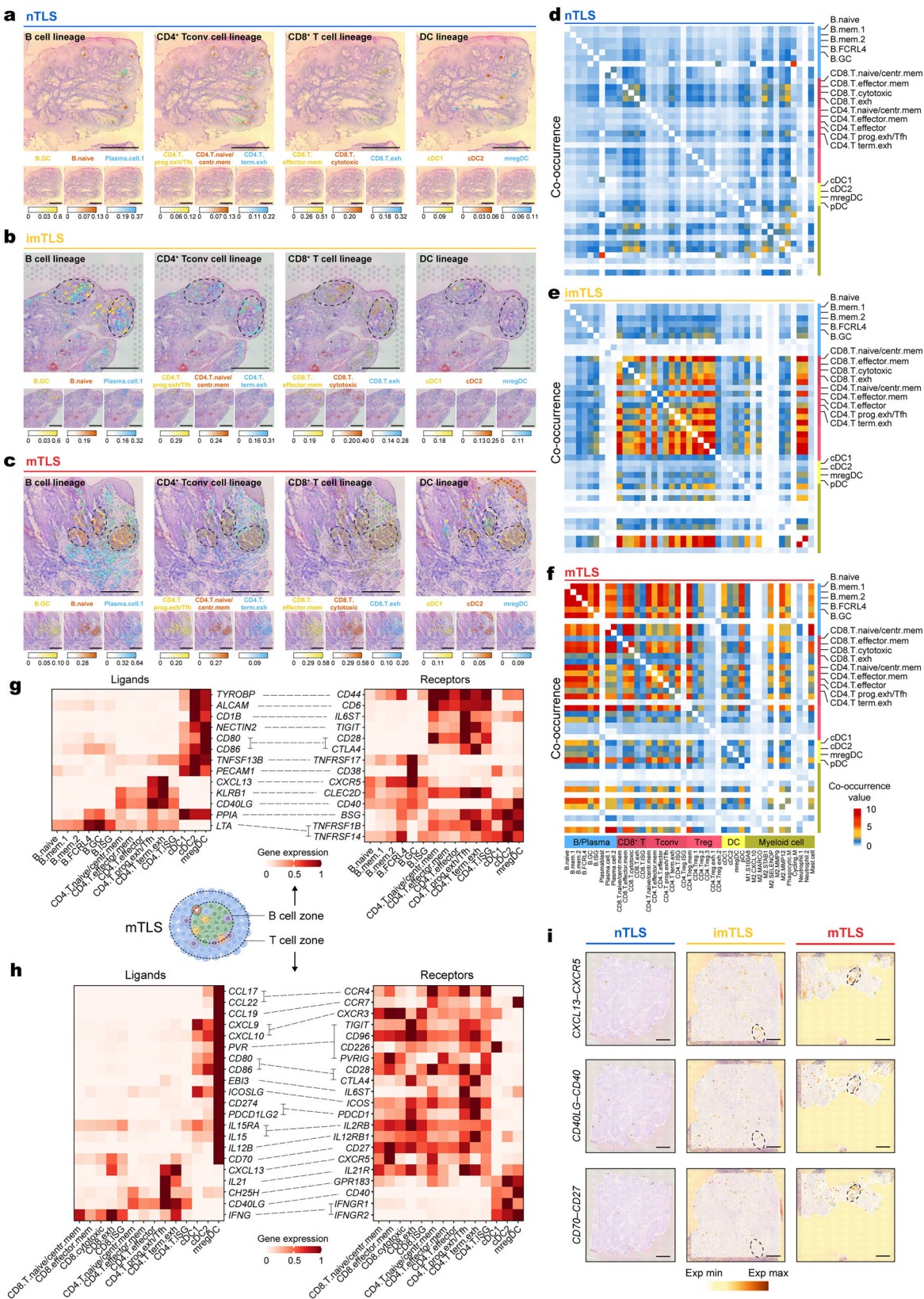

cell types (Supplementary Fig. 18a). cDC2 and GC B cells, which are enriched in mTLS, exhibit a notably enhanced costimulatory signaling interaction involving in B cell activation, such as *TNFSF13B* (BAFF)–*TNFRSF13C* (BAFFR) and *TNFSF13B* (BAFF)–*TNFRSF17* (BCMA) (Fig. 6g, Supplementary Fig. 18b and Supplementary Data 4), consistent with the function analysis of cDC2 (Fig. 5e). Another important

subcluster for B cell immunity is CD4+ Tex[prog], which also shares features of Tfh. We found that CD4+ Tex[prog]/Tfh has the ability to attract and activate GC B cells through *CD40LG–CD40* and *CXCL13–CXCR5* (Fig. 6g). Recent studies demonstrated the association between TLS and T cell immunity via mregDC expressing DC-LAMP (*LAMP3*)[4,28], we therefore revealed that the cell clusters participate in this process.

**Fig. 6 | Spatial distribution of intra-tumoral TLS throughout different TLS status of HNSCC. a–c** Spatial mapping of granulosa cell types from the scRNA-seq to spatial transcriptomics of nTLS, imTLS, and mTLS HNSCC using Cell2location. Spatial mapping of lineage of B cell, CD4⁺ and CD8⁺ T cell, and DC from the human scRNA-seq dataset to a respective spatial transcriptomics slide of HNSCC with nTLS (**a**), imTLS (**b**), and mTLS (**c**) statuses with cell2location. Estimated abundance for cell types (color intensity) to each Visium spot (color) shown over the H&E images. Scale bars = 1 mm. **d–f** heatmap of co-occurrence of B cells, CD4⁺ T cells, CD8⁺ T cells, and DCs across different TLS status. **g** Ligand–receptor pair expression analysis across B cells, CD4⁺ Tconv cells, DC, and B cell subclusters in B cell zone of mTLS, showing average gene expression by scRNA-seq **h** Ligand–receptor pair expression analysis across CD8⁺ T cell, CD4⁺ Tconv cell, and DC subclusters in T cell zone of mTLS, showing average gene expression by scRNA-seq. **i** spatial distributed expression of the CXCL13–CXCR5, CD40L (encoded by *CD40LG*)–CD40, and CD70–CD27 signaling axis in different TLS status. The color gradient indicates the gene expression (red, high expression; white, low expression). Scale bars = 1 mm.

Using ligand–receptor mapping, we showed the expression of candidate molecules that might promote interactions between mregDC, CD4⁺ Tex^prog/Tfh and effector memory CD8⁺ T cells (Fig. 6h). We found that mregDC expressed the highest levels of the CCR4 ligands (*CCL22*, *CCL17*); the naive or central memory T cell chemokine ligand CCL19, costimulatory genes (*CD80*, *CD86*, *PVR*, *ICOSLG* and *CD40*); cytokines that modulate T cells, such as *IL12B*, which is known to promote CD4⁺ T cell differentiation, and *IL15*, known for its role in promoting CD8⁺ T cells survival. In addition, the expression of regulatory genes *CD274* (encoding PD-L1) and *PDCD1LG2* (encoding PD-L2) was also increased in mregDC. Finally, leveraging spatial transcriptomics data, we identified that receptor–ligand pairs, such as *CXCL13–CXCR5*, *CD40LG–CD40*, and *CD70–CD27* were highly expressed in the mTLS site (Fig. 6i, Supplementary Fig. 17a–c). The significance of these receptor–ligand communications was further corroborated by analysis using CellphoneDB (Supplementary Data 4).

### mTLS is associated with better survival and response to ICB therapy in HNSCC

To investigate the prognostic value of mTLS in HNSCC, we used bulk RNA-seq from The Cancer Genome Atlas (TCGA) HNSCC dataset (TCGA-HNSC). This dataset was subjected to deconvolution, utilizing our scRNA-seq data as a reference for characterizing cellular populations, incorporating a deep learning model as described[40]. Subsequently, we conducted further analysis on the data using non-negative matrix factorization clustering, which led to the development of a TLS-based classification for HNSCC (Supplementary Fig. 19a). We assigned the TLS classification upon the expression levels of the TLS imprint signature. It is observed that the "mTLS" category manifests a high expression of this signature. Conversely, the "imTLS" category displays an intermediate expression of the signature, and the "nTLS" class is characterized by a low expression of the same (Fig. 7a). Furthermore, the category denoted as "mTLS" has consistently exhibited augmented signatures of various cell types, with the enrichment of these cell types corroborated by preceding scRNA-seq data (Fig. 7a). This includes B cells at the naive and GC stages, effector memory CD8⁺ T cells, naive/ central memory CD4⁺ T cells and CD4⁺ Tex^prog/Tfh, as well as cDC2 (Fig. 7a).

Upon validating the TLS classification, we advanced to explore the survival outcomes correlated with TLS status in HNSCC. It was observed that patients manifesting mTLS exhibited the longest overall survival in contrast to individuals with either imTLS or nTLS (Fig. 7b). In concordance with these observations, two specific cell types that are abundant in mTLS—GC B cells and effector memory CD8⁺ T cells—demonstrated a notable association with increased overall survival rates (Fig. 7c, d), suggesting a potential contribution of B cells and CD8⁺ T cells towards a better prognosis. Additionally, when clinical factors were integrated into a multivariate model, the status of mTLS was found to be significantly correlated with enhanced overall survival, disease-specific survival, and progression-free interval (Fig. 7e and Supplementary Fig. 19b, c). These results highlight the potential importance of mTLS as a prognostic marker in the clinical management of HNSCC.

To support these findings obtained from the scRNA-seq and spatial transcriptomics, we turned to a separate patient cohort, comprising 422 FFPE tumor specimens collected from individuals diagnosed with HNSCC (Supplementary Data 2). We utilized H&E staining along with mIHC to stain and distinguish the distinctive markers of B cells, T cells, DCs, and chemokines in the tumor tissues, thus allowing us to classify the TLS status of each sample (Fig. 7f, Supplementary Data 5). In accordance with our earlier analyses, mTLS is associated with a high density of CD20⁺ B cells, CD4⁺ T cells, CD8⁺ T cells, and DC-LAMP⁺ mregDC (Fig. 7g–j). Notably, this includes a high concentration of CD8⁺TCF1⁺ T cells and CD4⁺TCF1⁺CXCL13⁺ T cells, which are primarily categorized as stem-like CD8⁺ T cells containing naive/central memory and effector memory phenotypes, and as progenitor exhausted phenotypes within CD4⁺ T cells, respectively, according to scRNA-seq data (Fig. 7k, l). This investigation uncovered that specific cell types, specifically stem-like CD8⁺ T cells and CD4⁺ Tex^prog/Tfh, demonstrated a moderate correlation with B cells (Fig. 7m).

TLS status, as revealed by scRNA-seq and spatial transcriptomics, notably influences cellular interactions through increased receptor–ligand expression and alters the spatial organization of cellular neighborhoods in the TME (Fig. 6d–f). To identify the cell type most closely associated with mTLS in a large dataset, we quantified the spatial distances between every pair of cell types in the TME. Our analyses uncovered that CD4⁺ Tex^prog/Tfh, characterized by CXCL13⁺TCF1⁺CD4⁺ expression and primarily localized within the T cell zone of mTLS, not only resided in close proximity to B cells, but also demonstrated an aggregation near stem-like CD8⁺ T cells and mregDCs (Fig. 7n), forming a cellular niche as previously reported[28,31]. mIHC not only showed that CD4⁺ Tex^prog/Tfh cells (CD4⁺CXCL13⁺TCF1⁺) were enriched in the T cell zone within mTLS, but also provided confirmatory evidence for the existence of cellular triads, comprised of mregDCs, stem-like CD8⁺ T cells, and CD4⁺ Tex^prog/Tfh in T cell zone of mTLS (Fig. 7n, Supplementary Fig. 19d). Although there appears to be no difference in the general cellular neighborhoods among CD20⁺ B cells, CD4⁺ and CD8⁺ T cells across different TLS statuses, our findings indicate that the highest frequencies and shortest nearest-neighbor distances between CD4⁺ Tex^prog/Tfh and B cells, as well as stem-like CD8⁺ T cells occur in mTLS (Fig. 7o, Supplementary Fig. 20). In addition, we used our patient cohort with HNSCC to verify the relationship between CD4⁺ Tex^prog/Tfh cells, cDC2, and GC B cells (Supplementary Fig. 21). We found that both the CD4⁺TCF1⁺PD-1⁺ Tex^prog/Tfh cells and CD20⁺BCL6⁺ GC B cells increased in mTLS (Supplementary Fig. 21d, e). Furthermore, the cell-cell interactions between CD4⁺TCF1⁺PD-1⁺ Tex^prog/Tfh cells and CD1C⁺ cDC2, as well as between CD1C⁺ cDC2 and CD20⁺BCL6⁺ GC B cells, also increased in mTLS (Supplementary Fig. 21f, g).

We then assessed the relationship between mTLS and the patient responses to ICB therapy, which were evaluated by analyzing publicly available datasets from four published clinical trials[25,41–43]. In the scRNA-seq data of pre-treatment tumor samples from 6 patients with HNSCC undergoing ICB therapy[25], we initially migrated the cell annotation from our scRNA-seq dataset to this dataset utilizing the TOSICA model[44] (Supplementary Fig. 22). It was observed that cell types enriched with mTLS were linked to a favorable response to ICB therapy, including naive/central memory and effector memory phenotypes of CD8⁺ T cells, mregDCs, as well as CD4⁺ Tex^prog/Tfh and CD4⁺ Tex^term (Fig. 7p). Although a comprehensive RNA expression dataset from

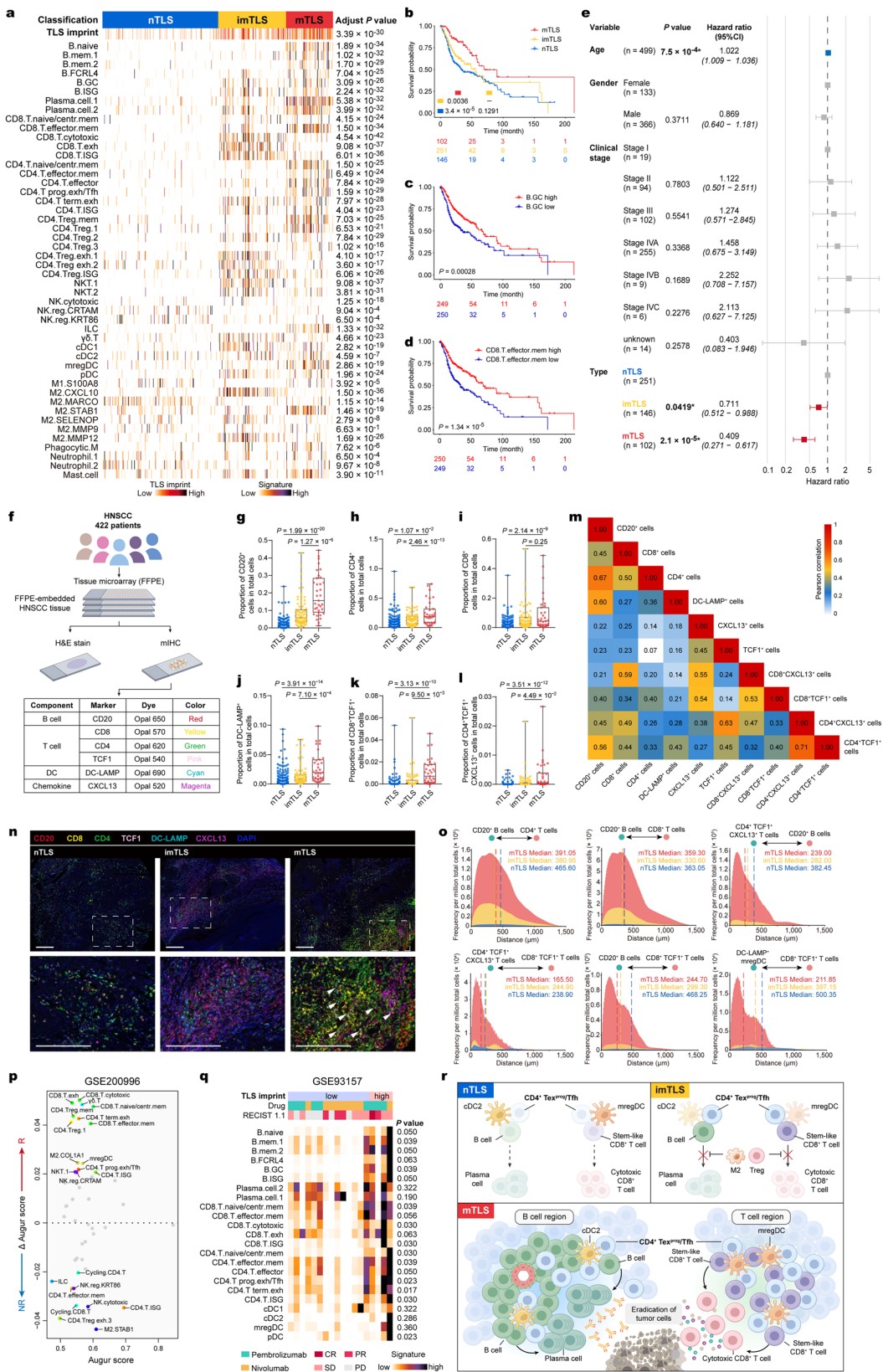

ICB-treated HNSCC patients was not accessible, it was recognized that squamous cell carcinomas (SCCs) arising from diverse anatomical locations frequently exhibit somatic mutations, gene amplifications, and deletions in critical regulators of cell cycle progression and chromatin remodeling[45]. Therefore, we employed SCC datasets as a proxy to infer potential associations between the presence of mTLS

and response to ICB therapy in HNSCC. In a patient cohort combining HNSCC and lung SCC using pembrolizumab and nivolumab for PD-1 blockade[41], it was observed that tumor samples with a high TLS imprint signature demonstrated a more favorable response to ICB therapy. Additionally, a significantly elevated signature of GC B cells and CD4[+] Tex[prog]/Tfh, which was linked to mTLS, was noted (Fig. 7q). Overall,

**Fig. 7 | mTLS is associated with better survival and response to ICB therapy in HNSCC. a** Heatmap plot showing TLS status in HNSCC (TCGA-HNSC) database (*n* = 500 independent samples) identified by the different immune cell types. A color gradient, transitioning from purple (representing enrichment) to white (representing depletion). The Benjamini–Hochberg adjusted *P* values obtained from two-sided Kruskal–Wallis tests. **b** Overall survival of patients with HNSCC by TLS status (*n* = 499 independent samples). Overall survival of patients with HNSCC by GC B cells (**c**) and CD8[+] effector memory T cells (**d**) was analyzed using log-rank test results, which indicated statistical significance. **e** Multivariate Cox proportional regression outcome, with clinical variables. For every variable included in the analysis, the reference level is the first one. A gray bar symbolizes a *P* value > 0.05; and blue and red bars symbolize *P* value < 0.05 positively and negatively, respectively. Error bars represent the 95% confidence interval. **f** Scheme of mIHC for HNSCC tissue. **g**–**l** Proportion of CD20[+], CD4[+], CD8[+], DC-LAMP[+], CD8[+]TCF1[+], and CD4[+]TCF1[+]CXCL13[+] cells in total cells of HNSCC tissue with different TLS status, respectively (*n* = 303 independent samples nTLS, *n* = 83 independent samples imTLS, *n* = 36 independent samples mTLS, two-tailed Mann–Whitney *U*-test, for box plots: box center line, median; box limits, upper and lower quartiles; box whiskers, maximum and minimum values). **m** Heatmap showing the Pearson correlation among cell subcluster abundances. The color gradient indicates the *P* value of Pearson correlation (red, high expression; white, low expression). **n** Images of an mIHC-stained HNSCC tissue with different statuses of TLS showing co-localization of B cell (CD20), T cell (CD4, CD8, and TCF1), mregDC (DC-LAMP), and chemokine (CXCL13) in HNSCC across. nTLS were repeated 303 times

independently with similar results, imTLS were repeated 83 times independently with similar results, mTLS were repeated 36 times independently with similar results. Scale bars = 200 μm. **o** The Distance between each type of cell in each core of the TMAs, with cores grouped by TLS status. Vertical dotted lines indicate the median distance. **p** Relationship between differential prioritization ΔAugur score for parenting between response and no response, and Augur score for HNSCC patients treated with ICB therapy. The scRNA-seq data of pre-treatment tumor samples from six patients with HNSCC undergoing ICB therapy GSE200996[25] were used for Augur analysis. Each dot represents a cell type in the TME of HNSCC. Gray symbolizes a ΔAugur score of <0.02; and the colored dot symbolizes positively and negatively a ΔAugur score's absolute value of >0.02, respectively. **q** Heatmap plot showing the response to anti-PD-1 therapy and abundance of different immune cell types. GSE93157[41] was used for this analysis. The Benjamini–Hochberg adjusted *P* values obtained from a two-tailed Mann–Whitney *U*-test. **r** Sketch map showing the TME of HNSCC with nTLS, imTLS, and mTLS. nTLS are unable to generate local anti-tumor immunity due to a lack of stem-like T cells and B cells. Meanwhile, imTLS, which is also enriched with B cells, cannot produce in situ anti-tumor immunity due to obstruction by immunosuppressive cells, including Tregs and M2 macrophages. mTLS enriched with stem-like CD4[+] and CD8[+] T cells and the presence of B cells at different maturation stages, supporting the generation of T and B cell responses. CD4[+] Tex[prog]/Tfh collaborated with cDC2 and mregDCs play a crucial role in regulating B cell and T cell response within mTLS. CR complete response, PR partial response, SD stable disease, PD progressive disease.

these findings demonstrate that mTLS was associated with improved response to ICB therapy in HNSCC.

## Discussion

Here, we investigated the role of TLS status in HNSCC and explored their respective roles in anti-tumor immunity. Employing an integrative approach that combines single-cell and spatial multi-omics technologies, we aimed to uncover the underlying mechanisms governing T and B cell responses, particularly emphasizing the distinctions observed among distinct TLS statuses (Fig. 7r).

We revealed that the cellular composition within different TLS statuses exhibits variations, with a higher frequency of T/NK cells and B cells in mTLS. Utilizing the scTCR/BCR-seq, we detected a greater number of clonotypes in T and B cells within mTLS, thus implying a higher probability for mTLS to recognize tumor neoantigens. Further, we identified stem-like CD8[+] T cells, specifically effector memory CD8[+] T cells that strongly express *GZMK* and *TCF7*, which are notably enriched in mTLS. These cells demonstrate the capacity to develop into fully functional CD8[+] T cells[31,46,47], suggesting that mTLS fosters an in situ CD8[+] T cell response. This observation aligns with recent findings that stem-like CD8[+] T cells are likely to supply an influx of tumor-specific activated CD8[+] T cells into the tumor[48]. However, it should be noted that non-mTLS also contain CD8[+] T cells, yet the dominant subset of these CD8[+] T cells displays a phenotype characterized by cytotoxicity and exhaustion. These cells express high levels of genes typically associated with exhaustion, such as *TOX*, *PDCD1* (PD-1), and *HAVCR2* (TIM-3), and they lack significant proliferative potential. Our earlier study and multiple other studies have confirmed the association between TLS and TCF1[+] T cells, demonstrating their role in predicting favorable prognoses[9,49–51]. Moreover, contemporary research points out that GZMK[+] effector (memory) CD8[+] T cells with stem-like features are positively correlated with a better response to ICB, whereas CD8[+] Tex[term] cells (PD-1[hi]TIM-3[+]TOX[+]) predominate in non-responders[28,31,52,53]. In subclusters of CD4[+] T cells, based on the single-cell level data, we observed that mTLS exhibit more enrichment of CD4[+] T cells which have tendency to differentiate towards exhaustion phenotypes instead of Treg phenotypes. There is accumulating evidence indicating that CD4[+] Tex cells exhibiting Tfh-like functions are positively associated with a favorable response to ICB therapy[30,31].

Regarding the B cell lineage, we found that mTLS harbored diverse subtypes of B cells and plasma cells, supporting that mTLS

generates anti-tumor B cell immunity in situ, in line with previous research findings[17]. In contrast, imTLS displayed an enrichment of B cells without concurrent plasma cells, hinting at a possible immunosuppressive milieu, possibly due to the higher prevalence of Tregs and M2 macrophages within imTLS, which could suppress the generation of plasma cells. nTLS showed a predominance of plasma cells without corresponding B cells, suggesting that these plasma cells might be produced at alternative sites, such as tumor-draining lymph nodes (tdLNs), as reported previously[30].

To elucidate the cellular contributors to intra-tumoral production of T cell immunity and B cell immunity within mTLS, we employed spatial transcriptomics and mIHC to precisely determine the localization of distinct subclusters of T cells, B cells, and DCs within the TME. The findings indicate that triads comprising CD4[+] T cells, GC B cells, and cDC2 are potentially instrumental in the local differentiation of B cells into plasma cells within mTLS. Concurrently, CD4[+] Tex[prog]/Tfh cells, mregDCs, and effector memory CD8[+] T cells form another cellular triad that is likely to facilitate the local differentiation of CD8[+] Tex[prog] cells into effector CD8[+] T cells within these mTLS. Through ligand–receptor mapping, we further hypothesize the signaling pathways involved in the generation of plasma cells and effector CD8[+] T cells. Current evidence suggests that mregDCs can form immune niches alongside CD4[+] and CD8[+] T cells[28], where CXCL13[+]PD-1[+] CD4[+] T cells interact with TCF1[+]CD8[+] stem-like T cells, leading to the generation of effector CD8[+] T cells responsive to ICB therapy as reported[31]. Moreover, mTLS that contain GCs can efficiently produce plasma cells and are capable of generating in situ antibodies within the TME[10,17]. In our work, we bridge the processes by which mTLS locally generate both T and B cell immunity within the TME and identify key cellular subsets mediating these critical processes, and identify a key cellular subtype that mediates both these critical processes, as CD4[+] Tex[prog]/Tfh cells. These CD4[+] T cells, which exhibit Tfh-like functionality, engage in collaborative interactions with cDC2 cells. They have the potential to promote the activation and subsequent differentiation of GC B cells into antibody-producing plasma cells. Simultaneously, in the context of T cell immunity, these same CD4[+] T cells cooperate with mregDCs to activate CD8[+] T cells, eventually giving rise to cytotoxic CD8[+] T cells.

Our study has several limitations. Firstly, while our trajectory and scTCR/BCR-seq sharing analyses support the intra-tumoral differentiation of CD8[+] and CD4[+] T cells as well as B cells in mTLS, the origin

of the T and B cells, whether already in the site of tumor or from the organ, such as tdLNs or blood, is unclear. Secondly, due to the limitations of technology, we were unable to investigate the TCR/BCR repertoire in a spatial context within the TME. Future studies could benefit from spatial transcriptomics of B and T cell receptors, which may reveal lymphocyte clonal dynamics as previously reported[54]. Thirdly, the localization of different immune cell subtypes was determined using Visium v1 technology with relatively low resolution (55 μm in diameter) and may encompass 5–30 cells per spot; thus, we employed bioinformatic tools to deconvolute the spatial transcriptome without directly observing cell-cell interactions. To enhance this research further, it is recommended to adopt newer methodologies featuring higher resolutions, such as Visium HD technology, multiplexed ion beam imaging by time-of-flight or imaging mass cytometry (CyCIF), to more accurately delineate the precise structure of mTLS and ligand–receptor interactions[55]. In our study utilizing TMA with HNSCC patients, considering the finite size of each tissue core and the inherent complexity of the three-dimensional organization of TLS[56], there exists a possibility of underestimation or uneven representation of TLS maturation stages across different regions, which may introduce a degree of bias into our assessment of TLS status in the samples. Fourthly, our study does not include direct functional assays to confirm the results obtained from single-cell and spatial multi-omics technology data. Our research aimed to study the different statuses of TLS in HNSCC tumors employing an integrative approach that combines single-cell and spatial multi-omics technologies.

Nevertheless, our study revealed that mTLS functionally involves CD4$^+$ Tex$^{prog}$/Tfh cells, along with DCs, acting as drivers of the intratumoral differentiation of cytotoxic CD8$^+$ T cells and antibody-producing plasma cells. These novel insights can enhance our understanding of the mechanistic function of mTLS and potentially inform strategies to induce mTLS formation within the TME, thereby maximizing the prognostic and predictive value of mTLS presence for HNSCC patients. Our study provides insights into the mechanisms underlying mTLS-mediated intra-tumoral immunity cycle against cancer.

## Methods

### Patient samples
This study was approved by the Medical Ethics Committee of the School and Hospital of Stomatology, Wuhan University. Written informed consent was obtained from each patient. HNSCC samples were derived from the Department of Oral and Maxillofacial Surgery, Wuhan University School and Hospital of Stomatology. All patients had histologically proven HNSCC and were suitable candidates for surgical resection. The TNM classification at diagnosis was based on the 8$^{th}$ edition of the TNM Classification of Malignant Tumors (Union for International Cancer Control). The pathological grade was diagnosed according to the 4th World Health Organization classification of head and neck tumors. A total of 14 HNSCC patients were enrolled for scRNA-seq, paired scRNA-seq, and scTCR/BCR-seq, as well as spatial transcriptomics. All 14 patients included were confirmed to be human papillomavirus (HPV) negative through both P16 protein staining and in situ hybridization dual testing. More detailed clinical characteristics of the patients are summarized in Supplementary Fig. 1.

### Sample processing
Fresh samples of HNSCC, mainly from the oral site, were collected from the primary tumor site during surgery. All tissues for sequencing, spatial analysis, and further work were stored in tissue storage solution (Miltenyi) at 2–8 °C until processing. Subsequently, the tissue was divided into three portions: one portion will be used for preparing single-cell suspensions, another portion will be utilized for sorting CD45$^+$ cells, and a third portion will undergo optimal cutting temperature (OCT) embedding for spatial transcriptomics as well as IHC staining to identify the status of TLS.

### IHC and mIHC to determine the status of TLS
For H&E staining, OCT-embedded tissues were cut into 6–8 μm sections, separated from the largest diameter of the tumor. Sections were also taken from 1/4 depth and 3/4 depth of the tumor and stained using a Leica automated slide stainer. The TLS status was then further determined by IHC and mIHC[57–59]. CD20, CD3, and CD23 immunostaining were used to determine the status of TLS as previously described[16]. Briefly, IHC assays were performed using anti-human CD20 (1:200, Cell Signaling Technology), CD3 (1:150, Abcam), and CD23 (1:200, Abcam) antibodies, as previously described[9]. Then, the mIHC staining was conducted using the Opal 6-Plex Manual Detection Kit (Akoya Biosciences) to further detect the TLS status as previously described[9]. Briefly, FFPE tissue slides were first deparaffinized and then incubated sequentially with primary antibodies CD20 (1:1000, Cell Signaling Technology), CD4 (1:1000, Abcam), CD8 (1:800, Cell Signaling Technology), BCL6 (1:600, Cell Signaling Technology), CD23 (1:900, Abcam), TCF1 (1:600, Cell Signaling Technology). This was followed by incubation with secondary antibodies (Akoya Biosciences) and corresponding reactive Opal fluorophores. Nucleic acids were stained with DAPI. Seven-color stained slides were scanned using a Vectra 3 Imaging System (Akoya Biosciences). Scanning was conducted at a magnification of 20×. The channel employed for multispectral imaging included DAPI, FITC, Cy3, Texas Red, and Cy5. The scanned data was subsequently processed by the inForm software (v.2.4, Akoya Biosciences) for analysis of multispectral imaging. Autofluorescence was acquired by assessing the region of the slides that did not contain fluorophores. mTLS refers to GC formation, a dynamic region with a network of CD23$^+$ FDCs and CD20$^+$BCL6$^+$ B cells[6,10]. The evaluation criteria for TLS status are as follows, as a previous study described[57]: if a sample contains only imTLS and not mTLS, it will be classified as imTLS. If a sample contains mTLS or both mTLS and imTLS, it will be defined as mTLS. Dilutions and catalog numbers for each antibody used for IHC and mIHC are provided in Supplementary Data 6.

### Single-cell suspension preparation
Tumor tissue was cut into approximately <2 mm$^3$ in tissue storage solution and then enzymatically digested using RPMI-1640 medium modified with collagenase IV (Biosharp, catalog no. BS165), DNase I (BioFroxx, catalog no.1121MG010), and hyaluronidase (BioFroxx, catalog no.1141MG100) in a C-tube using the gentleMACS system (Miltenyi, catalog no.130-093-237) at 37 °C for 1 h. After digestion, the cell suspension was added to 20 mL of RPMI-1640 medium, then passed through a 40-μm filter and pelleted by centrifugation at $300 \times g$ at 4 °C for 10 min.

### MACS
To isolate CD45$^+$ cells in tumor tissue, we first removed dead cells from the single-cell suspension by using the Dead Cell Removal Kit (Miltenyi, catalog no. 130-090-101) according to the manufacturer's instructions. Then, the CD45$^+$ cells were magnetically labeled with CD45 (TIL) MicroBeads (Miltenyi, catalog no. 130-118-780) and loaded onto a MACS LS-Column (Miltenyi, catalog no. 130-042-401) for positive selection. After centrifugation at $300 \times g$ at 4 °C for 10 min, the selective CD45$^+$ cells were stored in PBS with 0.04% BSA for further processing.

### scRNA-seq library preparation and sequencing
For the scRNA-seq experiments, the single-cell suspension ($1 \times 10^5$ cells/mL) was loaded onto a microwell chip according to the manufacturer's protocol for the GEXSCOPE Single Cell Kit (Singleron Biotechnologies, catalog no. 4180012). The scTCR-seq/scBCR-seq

libraries were constructed according to the protocol of GEXSCOPE Single Cell Immuno-TCR/BCR Kit (Singleron Biotechnologies, catalog no. 41831251). Sequencing libraries suitable for the Illumina sequencing platform were constructed after partial cDNA fragments and splicing. The remaining cDNA was enriched for the immune receptor (TCR/BCR), and the enriched products were amplified by PCR to construct a sequencing library suitable for the Illumina sequencing platform. Finally, each library was sequenced on Illumina HiSeq X, generating 2 × 150 bp PE reads.

### Spatial transcriptomics library preparation and sequencing

The tissues were frozen in isopentane and then embedded in OCT. OCT-embedded tissues were cryosectioned and placed on a Visium Spatial Gene Expression Slide following Visium Spatial Protocols-Tissue Preparation Guide (10x Genomics). Briefly, fresh tissues were coated carefully and thoroughly with room temperature OCT without any bubbles. OCT-coated tissues were then placed on a metal block chilled in dry ice until the OCT turned solidified and white. After assessing RNA quality using the TapeStation system (Agilent) and checking tissue morphology through H&E staining of the OCT-embedded tissues, the blocks were trimmed down to a suitable size fitting the Capture Areas. Sample fixing and imaging have been done in sample preparation, and section permeabilization will be performed as follows. Permeabilization was carried out for a duration determined by tissue optimization. The first strand of cDNA is synthesized via reverse transcription, and the second strand of cDNA was synthesized via PCR. Then the cDNA is denaturation, making the second strand of cDNA dissociated from slide. Free cDNA was then transferred from slides to tubes for further amplification and library construction. The final libraries contain the P5 and P7 primers used in Illumina amplification.

### Downstream analysis of scRNA-seq data

**Overview.** The pipeline was built using the CeleScope pipeline framework (Singleron Biotechnologies). CeleScope software (version 1.13.0) was used to perform read alignment, barcode filtering, and unique molecular identifier (UMI) quantification using the human reference genome (GRCh38-2020-A) and generate a raw counts matrix.

**Doublet detection.** We used Scrublet for cell doublet calling on a per-library basis[60]. Cells with predicted doublets scores greater than 0.2 were excluded from the following analysis.

**scRNA-seq data processing.** CeleScope-filtered matrices were loaded into individual AnnData objects using the Scanpy package (version 1.10.3) with the pipeline following their recommended standard practices, which are mainly based on the scverse ecosystem[61,62]. We excluded genes expressed by fewer than three cells and excluded cells expressing fewer than 200 genes or more than 4000 genes, more than 20% mitochondrial content, or with a total UMI count of more than 25,000. Cell cycle phase was assigned using the "score_genes_cell_cycle" function in Scanpy. After normalizing with the "normalize_total" and "log1p" functions with their default parameters, highly variable genes with mean expressions between 0.0125 and 3, and a minimum of normalized dispersions greater than 0.5, were identified using the "Seurat" method.

We regressed out the effect of the percentage of mitochondrial genes and the total count of UMI and scaled the data. Principal-component analysis (PCA) was calculated using the PCA function, and nearest neighborhood graphs were built using the neighbors function using 50 principal components. Then the harmony (version 0.0.9) was used for batch correction to remove the patient-specific effects[63]. And the community algorithm was applied for clustering using the Leiden function (resolution = 1)[64], and UMAP embeddings of major cell-type supersets (see below) were based on the 50 batch-corrected harmony

components. Differential expression between identified clusters was computed using a two-sided Wilcoxon rank-sum test.

**Clustering and cell subtype identification.** We first annotated the 14 major cells types identified in our dataset on the basis of well-known marker genes, including *PTPRC* (CD45), *CD3D*, *CD3E*, *CD8A*, *CD4*, *FOXP3*, *TRDC*, *NKG7*, *MIK67*, *CD79A*, *MS4A1*, *IGHG1* for lymphoid lineage (CD8⁺ T, CD4⁺ Tconv, Treg, NK/NKT, cycling T cells, B cells and plasma cells); *CD14*, *CD68*, *CD163*, *CD1C*, *LAMP3*, *JCHAIN*, *GZMB*, *TPSAB1* and *CSF3R* for myeloid lineage (monocytes, macrophages, cDC, mDC, pDC, mast cells and neutrophils); *PLVAP*, *VWF*, *PROX1*, *COL1A1*, *LUM*, *FAP*, *PDGFRA*, *ACTA2* for stromal cells (ECs, lymphatic ECs, fibroblasts and pericytes); and *KRT14*, *KRT6A* and *EPCAM* for cancer cells.

For T/NK cell subclustering, to avoid the dominant effect of T cell receptor variable genes in T cell clustering analysis, all *TRAV*, *TRBV*, *TRDV*, and *TRGV* genes related to the TCR were removed from the high variable gene list[25]. Similarly, for B cell subclustering, all *IGLV*, *IGKV*, and *IGHV* genes related to Ig were removed from the highly variable gene list to avoid somatic hypermutation-associated variances[30]. The list of genes related to TCR and Ig was acquired from the international ImMunoGeneTics information system. For each subcluster, the harmony was also used for batch correction to remove the patient-specific effects[63]. Owing to the variable amount and properties of cells in each major cell type, different parameters for clustering were used. For the clustering of T/NK cells, the top 40 batch-corrected harmony components were selected on the basis of 2000 HVGs. For the clustering of B cells and myeloid cells, the top 30 batch-corrected harmony components were selected on the basis of 3500 HVGs.

As a result, we identified 5 CD8⁺ T cells, 6 CD4⁺ Tconv, 7 Treg, 3 cycling T/NK, 2 NKT, 3 NK, 1 γδT, 1 ILC for T/NK lineage, 6 B cell and 3 plasma cell clusters for the B cell lineage, 4 DC, 9 macrophage, 2 neutrophil and 1 mast cell for the myeloid lineage, and 2 EC, 1 fibroblast and 1 pericyte for the stromal components (Fig. 2b and Supplementary Fig. 7). Clusters were annotated on the basis of known marker genes and cross-referenced against other published annotations (Supplementary Figs. 8, 14 and 15)[20–22,32,33]. Clusters or cells manually annotated as doublets or dissociated will be removed in the downstream analysis.

**scTCR/BCR-seq data processing.** Sequencing data for TCR sequences and BCR sequences was aligned and quantified using CeleScope against the GRCh38 human VDJ reference genome. Filtered annotated contigs for TCR sequences and BCR sequences were analyzed using Scirpy (version 0.17.2)[65]. In TCR/BCR quality control, cells expressing both BCR and TCR chains are categorized as "ambiguous," and those in which more than two receptor pairs are detected are labeled as "multichain." Both of "ambiguous" and "multichain" categories of scTCR/BCR-seq data were excluded from the following analysis. For TCR analysis, we selected CD8⁺ T and CD4⁺ T cells that were annotated via scRNA-seq analysis. Each unique *TRA* and *TRB* pair was defined as a TCR clonotype. Clonotypes were defined based on CDR3 nucleotide sequences with receptor_arms = "all." If one clonotype was present in at least two cells, cells harboring this clonotype were considered to be clonal, and the number of cells with such pairs indicated the degree of clonality of the clonotype. For BCR analysis, we selected B cells and plasma cells that were annotated via scRNA-seq analysis. Each unique heavy chain (IGH) and light chain (IGK or IGL) pair was detected, and a relaxed definition of B cell and plasma cell clonotypes by Immcantation Portal (version 4.5.0)[66]. Specifically, the optimal threshold for trimming the hierarchical clustering into B cell clones was firstly calculated by shazam package[66]. We then used the hierarchical clustering method[67] by the SCOPer package[68] to cluster B cell receptor sequences. This clustering was based on the similarity of junction region sequences within partitions that shared the same V gene, J gene, and junction length, while accommodating ambiguous V or J gene annotations.

TCR/BCR sharing was calculated as the number of unique TCRs shared between sample types (exact CDR3 nucleotide match). Using barcode information, T cells with prevalent TCR clonotypes and B cells with prevalent BCR clonotypes were projected on UMAP embedding.

**Single-cell pseudotime analysis.** To characterize the developmental trajectories in T and B cells, we applied the PAGA method[69] in a part of the single-cell analysis package Scanpy in Python to infer the potential differential trajectory. Moreover, we used PAGA to assess the most likely trajectories of cell progression among CD8+ and CD4+ T and B cells in HNSCC. The computations were carried out using default parameters. The edge connectivity between each subpopulation node for all edges is further compared by using an unpaired two-sided Student's t-test. We also applied Palantir to complement the trajectory analysis using default parameters[70].

**GLMs of cluster composition and kernel density estimates.** To estimate the effect of TLS status specificity on the composition of cell clusters, refer to the previous study[20]. We considered a GLM where we included interactions between TLS status and cluster identity for each major cell type defined in the scRNA-seq, H&E, and mIHC data. Using a binomial linear model, one can analyze counts of repeated observations of cell types or cell states as binary choices. We also use a method of kernel density estimation to determine the effect of TLS status specificity on the composition of cell clusters, as referenced in the previous study[20]. First, we performed kernel density estimation on the UMAP of each cell cluster for each TLS status using the kde2d function from the MASS package. Then, we calculated the contrast between the kernel density estimates of each two different groups of TLS status and visualized it in the UMAP space using the ggplot2 package.

**Calculation of gene set activity score.** We applied decoupleR (Python package) to evaluate the activity score of a gene set in each single cell based on transcriptome profiles[71]. For TLS signature scores, we calculated the score using a manually curated list of genes as input to the AUCell method[72]. We calculated the signaling pathway activity scores using the gene list provided by PROGENy[73], a comprehensive repository that contains curated sets of pathways and their associated target genes, as input for the multivariate linear model based on the recommendation. We calculated the functional enrichment scores using a manually curated list or gene list derived from Gene Ontology (GO) Biological Process terms as input for the overrepresentation analysis based on the recommendation. To verify the accuracy of cell annotation results from scRNA-seq, we compared the cell annotation using the AUCell algorithm with previously reported single-cell annotations.

The classical 12 chemokine signature contained[19]: *CCL2*, *CCL3*, *CCL4*, *CCL5*, *CCL8*, *CCL18*, *CCL19*, *CCL21*, *CXCL9*, *CXCL10*, *CXCL11*, *CXCL13*. The TLS imprint signature contained[17]: *IGHA1*, *IGHG1*, *IGHG2*, *IGHG3*, *IGHG4*, *IGHGP*, *IGHM*, *IGKC*, *IGLC1*, *IGLC2*, *IGLC3*, *JCHAIN*, *CD79A*, *FCRL5*, *MZB1*, *SSR4*, *XBP1*, *TRBC2*, *IL7R*, *CXCL12*, *LUM*, *C1QA*, *C7*, *CD52*, *APOE*, *PTLP*, *PTGDS*, *PIM2*, *DERL3*. The naive T cell signature contained: *CCR7*, *IL7R*, and *TCF7*, as derived from a previous publication[20]. The cytotoxicity T cell signature contained: *GZMA*, *GZMB*, *GZMH*, *PRF1*, *NKG7*, and *GNLY*, as derived from a previous publication[25]. The proliferation T cell signature contained: *STMN1*, *TUBB*, and *MKI67*. The Tex^prog^/Tfh cell signature contained: *CXCL13*, *TCF7*, and *PDCD1*. The Tex^term^ cell signature contained: *TOX*, *HAVCR2*, *CD101*, *PDCD1*, *LAG3*, *CXCL13*, *ENTPD1*, and *TIGIT*.

**Cell–cell interaction analysis.** We used the CellphoneDB tool, based on the CellphoneDB database (version 4.1.0), to infer cell–cell interactions of selected ligand–receptor pairs between T cell subclusters, B cell subclusters, and DC subclusters. The potential strength of interaction between two cell subsets was predicted using the "statistical_analysis_method" function, which retrieves interactions where the mean expression of the interacting partners displays significant cell state specificity by employing a random shuffling methodology. We extracted significant ligand–receptor pairs with a *P* value of less than 0.01. The results perform some exploratory analysis of the results obtained from CellphoneDB.

**The cell abundance estimated from bulk gene expression profiles.** To identify the cell-type proportions in the HNSCC, we used a bulk deconvolution tool called Bulk2Space (version 1.0.0)[40] based on deep learning frameworks to generate single-cell expression profiles from bulk transcriptome in several databases. The gene expression from TCGA-HNSC was downloaded from the TCGA data portal (https://portal.gdc.cancer.gov/). Other gene expression from these bulk RNA-seq datasets was downloaded from the GEO database (https://www.ncbi.nlm.nih.gov/geo/, GSE93157).

**Annotation transfer of cell-type annotation from scRNA-seq.** To demonstrate the correlation between mTLS and immunotherapy of HNSCC, we used TOSICA (version 1.0.0)[44], a new AI-based cell-type label transfer tool, to annotate cell types from our scRNA-seq data to a published dataset. Additionally, we verified our transferred annotation using classical markers for cell types.

Based on the annotated scRNA-seq data, we applied Augur (version 1.0.0)[74], a technique designed for prioritizing cell types that exhibit significant responses to biological perturbations within single-cell datasets. This approach facilitated the identification of correlations between various cell types and their responses to ICB. Other gene expression data from these scRNA-seq datasets was downloaded from the GEO database (https://www.ncbi.nlm.nih.gov/geo/, GSE200996).

**Location of cell types in spatial transcriptomic data.** To spatially ascertain the positions of cell states on the Visium transcriptomics slides, we employed Cell2location (version 0.1.3) for the purpose of establishing a spatial mapping of cell types. As a reference framework, we utilized general cell annotations derived from the primary analysis of scRNA-seq data, which comprised the cellular clusters of immune cells. We adhered to the default parameters, with the sole modification being the adjustment of cells_per_spot to 20. Each Visium section underwent independent analysis. The results were thereafter visualized in accordance with the guidelines provided in the Cell2location tutorial.

For co-occurrence analysis, due to the Visium v1 technology, a spot typically contains multiple cells (55 μm in diameter). We used the normalized distribution of cell types from Cell2location to analyze the co-occurrence of cell types. In order to predict the co-occurrence of two cell types in the spatial data, we multiplied the predicted cell abundance of each spot by the two cell types, added all values, and divided by the total number of spots, utilizing the following formula (1):

$$\text{Co} - \text{occurrence score}_{AB} = \frac{\sum_{j=1}^{N}\left(C_{Aj} \times C_{Bj}\right)}{N} \times 1000 \qquad (1)$$

where $C_{ij}$ denotes the predicted abundance of the *i*th cell type in the *j*th spot, $N$ represents the total number of spots, and $A$ and $B$ are the two specific cell types of interest. This calculation was performed by multiplying the relevance factor by 1000.

**mIHC.** For mIHC staining, FFPE tumor tissue blocks were serially sectioned into 4–6 mm sections. mIHC staining was conducted using the Opal 6-Plex Manual Detection Kit (Akoya Biosciences) as previously described[9]. Briefly, FFPE tissue slides were first deparaffinized and then incubated sequentially with primary antibodies CD20 (1:1000, Cell Signaling Technology), CD4 (1:1000, Abcam), CD8 (1:800, Cell Signaling Technology), CXCL13 (1:600, Abcam), DC-LAMP (1:600, Atlas

Antibodies), TCF1 (1:600, Cell Signaling Technology), PD-1 (1:600, Cell Signaling Technology), CCR7 (1:1200, Abcam), CXCR5 (1:1200, Abcam), Granzyme B (1:800, Cell Signaling Technology), BCL6 (1:600, Cell Signaling Technology), CD23 (1:900, Abcam) and CD1C (1:1000, Abcam). This was followed by incubation with secondary antibodies (Akoya Biosciences) and corresponding reactive Opal fluorophores, as shown in Figs. 2f, 3f and 7n, and Supplementary Figs. 3–5, 9, 19d and 21a–c. Following the Series 2 experiments, part of the tissue cores in TMAs were missing, as illustrated in Supplementary Fig. 21a–c. Nuclei acids were stained with DAPI. Seven-color stained slides were scanned using a Vectra 3 Imaging System (Akoya Biosciences). Scanning was conducted at a magnification of 20×. The channel employed for multispectral imaging included DAPI, FITC, Cy3, Texas Red, and Cy5. The scanned data are subsequently processed by the inForm software (v.2.4, Akoya Biosciences) for analysis of multispectral imaging. Autofluorescence was acquired by assessing the region of the slides that did not contain fluorophores. Dilutions and catalog numbers for each antibody used for mIHC are provided in Supplementary Data 6.

For the quantitative analysis, we carried out nuclear segmentation based on DAPI with an expected nucleus area ranging between 5 and $20\,\mu m^2$. Membrane segmentation was based on CD20, CD4, CD8, CXCL13, and DC-LAMP intensity by the inForm software (v.2.4, Akoya Biosciences). We observe ~20–30 positive cells in the field of view and set a threshold for determining positive cells based on their expression levels on the cell membrane or nucleus. Calculations for the distances between each cell were performed using a distance matrix based on the X and Y positions of each cell in the field of view, which can be obtained from inForm software. Following the identification of the distances for each cell, summary statistics regarding these distances were compiled for each status of TLS under examination.

### Reporting summary

Further information on research design is available in the Nature Portfolio Reporting Summary linked to this article.

### Data availability

The data deposited and made public are compliant with the regulations of the Ministry of Science and Technology of China, and have get the accessions in the Ministry of Science and Technology of China. The raw scRNA-seq, scTCR/BCR-seq, and spatial transcriptomic data generated in this study have been deposited in the Genome Sequence Archive at the National Genomics Data Center (China) under the accession code GSA-Human (HRA007402) [https://ngdc.cncb.ac.cn/gsa-human/browse/HRA007402]. The matrix of scRNA-seq, scTCR/BCR-seq, and spatial transcriptomic data has been uploaded to the OMIX database under accession code OMIX009480 with open access [https://ngdc.cncb.ac.cn/omix/release/OMIX009480]. The public data used in this study include RNA-seq data are available from TCGA datasets (https://portal.gdc.cancer.gov/). Previously published scRNA-seq data and gene expression data are available from the GEO datasets (https://www.ncbi.nlm.nih.gov/geo/) under accession codes GSE200996 and GSE93157, respectively. All data are included in the Supplementary Information or available from the authors, as are unique reagents used in this Article. The raw numbers for charts and graphs are available in the Source Data file whenever possible. Source data are provided with this paper.

### Code availability

Codes used in this study are available at the GitHub website (https://github.com/lihaowhusos/TLS_in_HNSCC).

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

## Acknowledgements

This work was supported by the National Natural Science Foundation of China (NSFC) 82303328 (H.L.), 82472818 (Z.-J.S.), 82103336 (W.-W.D.), 82273202 (Z.-J.S.), the Fundamental Research Funds for the Central Universities 2042023kf0141 (H.L.), 2042024kf0021 (Z.-J.S.) and 2042022dx0003 (Z.-J.S.), National Key Research and Development Program 2022YFC2504200 (Z.-J.S.), the Interdisciplinary innovative foundation of Wuhan University XNJC202303 (Z.-J.S.), the Natural Science Foundation of Wuhan 2023020201020516 (H.L.), and the Young Elite Scientist Support Program by CSA 2023PYRC001 (H.L.).

## Author contributions

Z.-J.S., W.-W.D., and H.L. conceived and designed the experiments. H.L., M.-J.Z., B.Z., W.-P.L., S.-J.L., Q.W., Q.-C.Y., and W.-D.W. performed the experiments. H.L., W.-P.L., and D.X. analyzed the data. H.L., M.-J.Z., W.-W.D., C.-F.H., and Z.-J.S. wrote the manuscript. All authors have given approval to the final version of the manuscript.

## Competing interests

The authors declare no competing interests.
