## [Transparent Peer Review file · Nature Communications]

Mature tertiary lymphoid structures evoke intra-tumoral T and B cell responses via progenitor exhausted CD4(+) T cells in head and neck cancer

Corresponding Author: Professor Zhi-Jun Sun

Version 0:

Reviewer comments:

Reviewer #1

(Remarks to the Author)

Aiming at deciphering the underlying mechanisms that trigger antitumor B and T cell responses within tertiary lymphoid structures in HNSCC patients, Liu et al. performed single-cell RNA and antigen receptor sequencing and spatial transcriptomics on patients with different statuses of TLS. The authors claim that primary tumor samples containing mTLS are highly enriched with stem-like T cells, as well as B cells at various maturation stages, which supposedly supports the generation of T and B cell responses in the tumor microenvironment. In addition, CD4 progenitor exhausted/follicular helper T cells are also enriched within mTLS, which, according to the authors, can activate B cells to produce plasma cells in the germinal center, although no clear evidence is shown.

There is no evidence that supports the following conclusions: i) mTLS can generate antitumor B and T cell responses, ii) follicular helper T cells activate B cells to produce plasma cells in the germinal center, iii) DC-LAMP+ dendritic cells support the activation of CD8+ T cells, and iv) imTLS cannot produce in situ anti-tumor immunity due to obstruction by immunosuppressive cells, such as regulatory T cells (Tregs) and M2 macrophage.

The authors always explore the data as following: cell clustering, then enriched genes/pathways, followed by pseudotime analysis, cell abundance according to the TLS status, and then clonotype analysis when possible. Data would perhaps be more robust if analyses were not biased by this formula.

Overall, the manuscript lacks novelty, and it is highly descriptive and circular. Conclusions are not supported by the data provided, and an extensive validation will be needed to fully support the conclusions of the study.

In conclusion, the quality and novelty of the data do not justify their publication at Nature Communications.

(Remarks on code availability)

Reviewer #2

(Remarks to the Author)

Major comments

Do mTLS patients contain imTLS or imTLS patients have a small percentage of mTLS? When stratifying patients by TLS, what is the cutoff point of % mTLS or imTLS?

Please comment on this. If mTLS patients have imTLS, even though the number of imTLS is low in mTLS patients, vice versa for imTLS patients, analyzing mixing cells from different types of TLS patients may potentially produce false results. The description for results is rather simple and, sometimes, too brief. It will be much better to expand the result description to provide enough information to help a better understanding on experiments, data interpretation, etc.

Fig. 1c. What does this figure intend to tell? It will be better to provide an interpretation for this figure in the Result section. Also, it is very hard to distinguish which is which based on patient ID and cell type samples using the different colors, which makes this figure less meaningful. For example, nTLS may be completely absence from B cell clusters of B and Mast cells shown in Figure 1e. The authors failed to provide explanation to those presumably interesting results

Fig. 1d. It will be better to use a different and highly contrasted color combination to make these two sample types distinguishable much easily. Again, what is the figure going to tell? Is the figure to tell that the sample types are different? The Result section mentioned Fig. 1h, 1i, but they are missing in figures.

Fig. 2b presented a lot of results, but the authors did not describe and explain the results. For examples, why some T-cell immune types and signaling components are highly expressed in one subset (cluster), but down-regulated in others.

Fig. 2c. Using TCF7, CCR7 and IL-7R to define stem-like memory T cells (Tscm) is not accurate. These genes are expressed by naïve or central memory cells as well. CD95, CD122 and CXCR3 should be included in the characterization.

Fig. 2c. the authors mentioned ISG in the Results section, but did not show this gene expression in Fig. 2c. The same as the description of other results in the Results section that Fig. 2c does not show any specific genes. It seems like that the Fig. 2b and 2c were flipped and mixed up.

It seems like that “cycling T/NK cells” labeling is missing in Fig. 2b.

Fig. 2d, 2e. It will be better to perform the same analysis in imTLS and nTLS as a comparison.

The description for Fig. 2f should move up based on alphabetical order.

The description for Extended Data Fig. 6-7 is missing in the Results section.

Fig. 2g. The authors combined cluster 1 (naïve/central memory) and 2 (effector memory) and named it as CD8+ T stem-like (Tscm). The literatures say that Tscm is a T-cell subtype on its own and is different from naïve or central memory or effector memory. Please comment on this.

Fig. 2g, clonal expansion figure. What does the numbers with colors mean? It will be better to provide the interpretation for Fig. 2g. In current version, the authors only mentioned they did this experiment.

The description for Fig. 2h should come before the description for Fig. 2i.

Fig. 2j. It will be better to provide an explanation to help understand this figure.

The mTLS cells have both increased expanded clonotypes and shared clonotypes of CD8+ T cells. Please comment on whether these two results are contradictory.

Fig.3a. Technically, when reclustering CD4+ cells, the resulting clusters will be different because of the absence of other cells that contribute to the clustering algorithm. How are the authors able to identify the exact same clusters as Fig. 2b? It will be better to provide more information about this.

Fig. 3g. Possible labeling error. Should CD8+ from title “TCR clonotype sharing across CD8+ T cell subclusters” be CD4+?

Fig. 3. Authors found that mTLS contain more exhausted CD4+ T cells, especially progressive Texh.

Fig.4c. The description for Fig. 4c, 4e should move up based on alphabetical order.

Fig. 5a. It will be better to provide the information on which markers are used to define myeloid cells
What are sample types, whole tumor tissue or CD45+ cells, used for this analysis?

Fig. 6. The authors mentioned Fig.1i. It seems that Fig.1i is missing.

Fig. 6c. The image that staining of B.naive, CD4.T.naive, CD8.cytotoxic and cDC2 was abundant and saturated in the same follicle raises concern that the staining is not specific. Please comment on this.

Fig. 7c-d. The patient numbers in each group are almost the same in Fig. 7c and Fig. 7d. Please double check and make sure those numbers are correct.

Grammar errors:
Line 561, “For tumor tissue” should be “Tumor tissue”.
Line 148. What is “almost phenotypes”?
Line 334. Incomplete sentence: facilitated the identification of a cell cluster associated with TLS, which included B cells, T cells, DCs and myeloid cells.

(Remarks on code availability)

Reviewer #3

(Remarks to the Author)

Manuscript title: Mature tertiary lymphoid structures evoke intra-tumoral T and B cell responses via progenitor exhausted CD4+ T cells in head and neck cancer

Li et al. (Sun)

Submitted to: Nature Communications

Summary:

The manuscript addresses the fine description of the tumor microenvironment (TME) of head and neck squamous cell carcinoma (HNSCC) tumors, using first a combination of single-cell multi-omics and low resolution spatial transcriptomics on a small series of samples, and second multiplex IHC and bioinformatics deconvolution of bulk RNA-seq on larger series of patient samples (either from this study or publicly available).

The authors focus mainly on 14 samples which they have categorized based on their content of tertiary lymphoid structures (TLS) and their degree of maturity. They study 4 samples visibly devoid of TLS (nTLS = non-TLS), 4 samples containing only immature TLS (imTLS), and 6 samples containing mature TLS (mTLS). The authors split the freshly resected tumor specimens into several parts which they processed separately to yield (1) scRNA-seq data on all viable cells from the whole dissociated tumors, (2) multi-omics (scRNA-seq + scBCR-seq + scTCR-seq) on CD45+ cells, and (3) spatial transcriptomics (10x Genomics Visium, 55µm-diameter capture spots, whole transcriptome). The authors use state-of-the-art bioinformatics analyses to qualify and quantify cell types and cell states, first in a broad manner (Figure 1), then focusing specifically on T and NK cells (Figures 2 and 3), B cells (Figure 4), Myeloid cells (Figure 5). They map the cell types and states they have characterized in details in the first figures to their location in spatial transcriptomics dataset (Figure 6) through deconvolution, co-occurrence analysis, and ligand-receptor analyses. Finally, they extend their observations to other cohorts of samples analyzed by distinct methods (Figure 7), confirming their results on the small series of 18 samples, and extending their observations to define biological correlates of survival for patients with HNSCC, also when treated with immune checkpoint blockade therapy.

Assessment:

The manuscript is clearly written, although it would benefit from a thorough editing by a native English speaker. Overall, the authors have presented a very complete dataset, with high quality scRNA-seq and multi-omics data on a carefully selected series of HNSCC samples reflecting distinct TLS infiltration patterns. Given the association of TLS with clinical course in many solid tumor types, including HNSCC, this descriptive study is of major interest to researchers and clinicians interested in understanding and harnessing the TME for obtaining durable response in HNSCC. The bioinformatic methods are clearly explained, and the analysis tools that the authors have used have been carefully selected to fit the purposes of their analyses.

Nevertheless, there are some major points that I believe should be addressed to support the main claims of the article on functional immune mechanisms occurring in mature TLS, and strengthen the study.

First, the authors should relate to previous single-cell studies of HNSCC, including the work of Ruffin et al. (Nature Communications 2021) which suggested that TLS infiltration was tightly link to HPV status. This work is not referenced, and the HPV status of samples in the study is not described.

Second, it is unclear to me whether the clonality assessment for B cells included a “tolerance” for intra-clonal somatic hypermutation. The methods seem to describe that CDR3 identity was used for grouping B cells in clonotypes. If this is indeed the case, I suggest using a more “relaxed” definition of clonotypes that would group cells even if they have diverged from their naïve B cell ancestor through somatic hypermutation. There are many ways to do so, and I could recommend the Immcantation suite in R (<https://immcantation.readthedocs.io/en/stable/>). After revising the B cell clonotype definition, if necessary, it would be interesting to comment on the fact that GC B cells in mTLS do not seem to be clonally linked to plasmablasts and plasma cells, whereas Memory B cells have a slightly higher clonotypic link with antibody-secreting cells. That would suggest that GC B cells in mTLS are not the main or the direct source of plasmablasts and plasma cells, contrary to what is depicted in the summary Figure 7r.

Third, the spatial datasets, either from spatial transcriptomics or from multiplex IHC, would benefit from proper quantitative analyses, with statistically robust hypothesis testing taking into consideration the different samples analyzed. For example, Figure 6 a-c only show examples and are barely readable, while the correlation matrices in Figure 6d-f seem to be built based on all ST datasets from each sample category, and do not offer any statistical analysis. In Figure 7n-o, it is unclear to me what the main result is regarding cell type proximity, and how such result is robustly backed up by the data presented. I believe the authors should extract the relevant metrics (e.g. proximity of CD4+ TCF1+ CXCL13+ cells to B cells, DC-LAMP+ cDCs, or CD8+ T cells) from each slide individually, and analyze the distributions of those metrics in all sample groups with robust statistical analysis.

Fourth, it is unclear to me how the CD4+ Texprog subset differs from bona fide TFH cells. Most studies on TLS thus far have identified CD4+ T cells within mTLS as TFH cells, expressing PD1 and CXCL13 like TFH in secondary lymphoid organs. Here the authors do not identify any TFH cells in their samples, even though the CD4+ Texprog subset would classify as such in my opinion. Is it just a question of nomenclature?

Fifth, any claim on functionality, e.g. crosstalk between cDC2s, CD4+ Texprog and B cells in mTLS, should be backed up by functional assays, not only bioinformatics analyses of scRNA-seq and spatial transcriptomics data. In the absence of functional assays, any conclusion should remain speculative and be stated as such.

I will further detail my concerns and suggestions below.

1. Lines 188-189: Proper quantification of multiplex IHC images is required to support the claims.
2. Line 213: Fig. 3f is an IF image. Please correct Figure 3 citations for proper panels throughout the text.
3. Line 276: the fact that PCs are found in nTLS samples may indicate that PCs are generated outside TLS, but not necessarily outside the TME. In secondary lymphoid organs, PBs and PCs may be generated from naïve or memory B cells in GC-independent and extra-follicular responses.
4. Line 283: see my general comment above on BCR clonotype definition. Please revise conclusions accordingly if the clonotype definition should be revised.
5. Lines 289-290: see my comments above, the results indeed suggest that PCs originate not from GC B cells but from memory B cells. The diagram in Figure 7r should be revised accordingly.
6. Lines 323-326: it is unclear to me what is the relevance of cDC2 cells expressing genes involved in “Ig production”, since cDC2s are not producing Ig themselves. Could the authors precise what those genes are? How do those analyses suggest that cDC2s play a crucial role in regulating antibody responses?
7. Lines 337-353: see my comment above. Those conclusions should be backed up by proper image quantification and statistically robust analyses.
8. Lines 362-364: please explain how CD4+ Texprog differ from TFH.
9. Lines 374-377: those figures are not readable. Please provide a quantification over all analyzed sections.
10. Line 425: Figure 7n is an example. Please provide proper quantification and statistically robust analysis.
11. Line 453: I believe “elucidated” is too strong a word when no functional assay on immune mechanism has been used.
12. Line 485: “nTLS” missing at the beginning of the sentence.
13. Line 500-501: the data presented here seems to show that PCs are not derived from GC B cells, so the link between GCs and PC production should be discussed more critically.
14. Lines 504-507: in the absence of functional assays, this is pure speculation and should be phrased as such.
15. Line 518: I believe the authors mean “bioinformatic tools”, not “biological tools”.
16. Figure 3g: revise header to state “CD4+”, not “CD8+”.

Pierre Milpied, PhD
Centre d’Immunologie de Marseille-Luminy (CIML)

(Remarks on code availability)

Reviewer #4

(Remarks to the Author)

In this study by Li et al, the investigators base their main analyses on a set of 14 tumors with predefined status of TLS (none, immature and mature TLS), and they apply single cell RNA/TCR sequencing and spatial transcriptomics to this cohort of HNSCC. In these analyses, the key determinant is the allocation to the three different TLS statuses and then define the differences in the tumors where these three states are present. They set off then to identify the key subsets of immune cells (eg CD4+ Texprog), their distribution and interaction, and their role in regulating T and B cell responses within mTLS of the TME. Based on these findings, they developed a TLS-based classification (TLS imprint signature) and applied this signature on bulk-RNAseq from TCGA HNSCC dataset for TLS classification, and found that patients with mTLS status have better clinical outcomes (ie overall survival, disease specific survival and progression-free interval). They also applied this signature on datasets derived from SCC patients treated with anti-PD1 and observed tumor samples with high TLS imprints demonstrated a more favorable response to the treatment.

The amount of data generated in this study is amazing with more than 200K single cells sequenced, with TCR and BCR sequencing and spatial data, and this would all be incredibly useful reproduces for subsequent analyses. However, I have trouble following the manuscript for a number of key reasons, which dampens my enthusiasm for this manuscript. It is a case of doing way too much and not having a point! It was also very painful to go through the paper as all the figures are not in order that they are cited and some are even missing, which does not make easy reading or reviewing.

The most important premise for the entire paper from Figures 1-5 is the ability to classify the 14 tumors into 3 groups, and is the basis of any analyses. The problem is that I am not entirely convinced that they have shown this data completely or accurately. I find it hard to believe that tumors neatly fall into the 3 categories with little gradation or overlap between imTLS and mTLS. This is in fact demonstrated in the differences shown for CD8 and CD4, where the p-values between specific subpopulations are so weak, and often driven by one or two outliers. All the data for CD8 and CD4 trajectory and gene analyses are meaningless as what they deem to be 'stem-like' is actually just naive CD8 and CD4 cells, which if you include in any trajectory from any tumor will show the same loss of naive and gain of dysfunctional markers. I find the data from Figures 1-5 suffer from this repeated error based on a false premise and am completely lost with all the claims, with the good work of TCR and BCR sequencing being completely lost in the process. As I am about to completely give up on the paper, the saving grace arrives in the form of Figure 6 which is the spatial analyses that seems to recapitulate their findings in the tumor and TLS microenvironment which is at least more convincing than the aimless scRNAseq data prior. Validating this data on TCGA or even TMA to show association with outcome is an easy trap to fall under, but the only clinical point that matters is the response to PD1 blockade, and this was very limited. All in all I feel that if the authors presented this paper better, it would be more convincing in my mind to sell as a story. I would have started with the Spatial data and from there branched out to specifically look at the different tumor classes as a range rather than discrete groups and focused on TCR and BCR sequencing and T and B cells respectively. In its current form, it has too many fundamental flaws to be acceptable.

The following are additional specific points that need to be addressed as well:

1. As stated before, I am not entirely convinced with the IHC images in Extended Data Fig. 2 used to represent nTLS, imTLS and mTLS, as the authors are selective in the regions they want to show. Please provide more representative images with higher resolution for all the tumor and how it was decided to call each tumor into a discrete vein of n, im or mTLS.
2. TLS of different maturation statuses can be distributed throughout the TME. The presence and density of TLS can also vary across different regions of the tumor. Hence, a scoring or quantification system has to be used to classify TLS more rigorously in the context of their findings. The authors have not mentioned them in this manuscript and this needs to be addressed as this forms a crucial part for the subsequent interpretation of the results.
3. Figure 1h and 1i are either missing or not appropriately labelled
4. Is Figure 3g showing CD8 or CD4 clonotype sharing?
5. "Interestingly, CD4+ Texprog phenotype exhibited the highest expression of cytotoxic markers (PRF1, GZMA, GZMB, IFNG and GNLY) (Fig. 3g, Extended Data Fig. 10e), aligning with recent discoveries indicating the presence of a phenotype of cytotoxic CD4+ T cells within tumors and can directly kill cancer cells. This may suggest that mTLS possess the capability to generate cytotoxic CD4+ T cells in situ in the local TME." Please cite these published findings accordingly.
6. Figure 3p is not found
7. "Gene enrichment analysis revealed that the CD4+ Texprog subset was significantly enriched with features of Tfh-like functions, including B cell chemotaxis and GC formation (Fig. 3e)". Should be referring to Fig 3d
8. "Interestingly, nTLS showed minimal numbers of B cells, except for an enrichment of plasma cells, suggesting that these plasma cells may have originated from outside the TME (Fig. 4b, c)." Please elaborate. How does this finding suggest that nTLS plasma cells may have originated from outside the TME. Can the plasma cells originate from nearby TLS?
9. Line 320-321, the claiming statement "This suggests that mast cells may play a significant role in the process of tertiary lymphoid structure formation" requires further evidence for support. Mast cells may not necessarily play a direct role in TLS formation. Rather, their presence could be a secondary consequence of the inflammation associated with TLS formation. They may act as bystanders that are recruited to the site after TLS formation, instead of actively contributing to the initial development of the structures. Please provide data or references that substantiate the claim.

10. Figure 5e is under-described.

11. The histological images provided are of low resolution, making it difficult to accurately visualize and interpret the features described by the authors.

12. Figure 6h is included in the main figure but is not referenced or described in the text. Please clarify its relevance or consider removing it if it does not contribute to the discussion.

13. Line 368-373: Where is PVRL2 in Fig 6d?

14. Line 415-417: "This investigation uncovered that specific cell types, specifically stem-like CD8+ T cells and CD4+ Texprog demonstrated a strong correlation with B cells (Fig. 7m)." The reported correlations - 0.40 for stem-like CD8+ T cells (CD8+TCF1+) vs. B cells (CD20+), 0.45 for CD4+ Texprog (CD4+CXCL13+) vs. B cells, and 0.56 for CD4+ Texprog (CD4+TCF1+) vs. B cells - raise the question of what constitutes a 'strong' versus 'weak' correlation. Please clarify the criteria used to define the strength of these correlations. In addition, the population CD4+TCF1+CXCL13+ that was described as CD4 progenitor exhausted phenotypes here is missing in Figure 7m.

15. Figure 7e is included in the main figure but is not referenced/described in the text. Please clarify.

16. Figure 7n is too pixelated and cannot be used to support what is described in the text.

17. Line 429-432: "Although there appears to be no difference in the general cellular neighborhoods among CD20+ B cells, CD4+ and CD8+ T cells across different TLS statuses, our findings indicate that the highest frequencies and shortest nearest-neighbor distances between CD4+ Texprog and B cells, as well as stem-like CD8+ T cells occur in mTLS." Please show the data or cite the figure according.

18. Line 433-435: While the use of published dataset for validation is commendable, NSCLC represents a different tumor type, and may limit its relevance to HNSCC.

19. Lines 433-435: "While the use of a published dataset for validation is commendable, NSCLC represents a different tumor type, which may limit its relevance to HNSCC. Please justify the use of this dataset or consider discussing the potential differences between these tumor types that could affect the findings."

20. Line 463-464: "These cells demonstrate the capacity to develop into fully functional CD8+ T cells, suggesting that mTLS fosters an in situ CD8+ T cell response." Please including relevant references.

21. Lines 470-472: The statement mentions "multiple other studies" confirming the association between TLS and TCF1+ T cells. Yet only one citation is provided. Please either provide additional references to support this claim.

(Remarks on code availability)

Version 1:

Reviewer comments:

Reviewer #1

(Remarks to the Author)

I reiterate the fact that the manuscript by Liu et al. provides insights into B and T cell responses associated with tertiary lymphoid structures in HNSCC patients. However, I also reiterate that the conclusions drawn in the study are largely based on correlative analyses, without validation through functional experiments. While correlation can suggest interesting associations, it does not establish causation and, in this case, it is unclear whether the immune cell subpopulations observed in TLS tissue samples actually originate from the mature TLS, or whether they are simply present due to the surrounding inflammatory environment. Additionally, the data provided are not sufficiently robust to prove that B and T cells induce antitumor responses.

At present, the conclusions appear overstated, as they are based on data that do not fully support such broad claims. To ensure accuracy and precision, I recommend that the authors either conduct functional experiments or, at the very least, adjust their conclusions to reflect the limitations of their data, including the title of the manuscript, as there is no solid evidence showing that "mature tertiary lymphoid structures evoke intra-tumoral T and B cell responses via progenitor exhausted CD4+ T cells...".

(Remarks on code availability)

Reviewer #2

(Remarks to the Author)

All the comments and concerns have been addressed.

(Remarks on code availability)

Reviewer #3

(Remarks to the Author)

The authors have addressed all of my comments in a satisfying manner. I believe the manuscript is now much stronger through more appropriate bioinformatics analyses and quantitative analyses.

(Remarks on code availability)

Reviewer #4

(Remarks to the Author)

Most of my queries have been satisfactorily responded to, although I feel the the authors has the opportunity to analyse the TLS maturity as a continuum rather than dividing it binarily according to Vanhersecke et al, and given themselves an advantage in doing so. I still feel that Figure 6 would have benefited from coming on earlier in the presentation as the take off point and make this manuscript more readable. I am glad to see that that all the missing figures and figure order have been rectified, as otherwise it is very difficult to wade through such a dense paper.

(Remarks on code availability)

Not qualified to review code.

Point-by-Point Response to Reviewers' Comments

Reviewers Comments:

=====

Reviewer #1 (Remarks to the Author):

Comment:

Aiming at deciphering the underlying mechanisms that trigger antitumor B and T cell responses within tertiary lymphoid structures in HNSCC patients, Liu et al. performed single-cell RNA and antigen receptor sequencing and spatial transcriptomics on patients with different statuses of TLS. The authors claim that primary tumor samples containing mTLS are highly enriched with stem-like T cells, as well as B cells at various maturation stages, which supposedly supports the generation of T and B cell responses in the tumor microenvironment. In addition, CD4 progenitor exhausted/follicular helper T cells are also enriched within mTLS, which, according to the authors, can activate B cells to produce plasma cells in the germinal center, although no clear evidence is shown.

Response:

We would like to sincerely thank you for your detailed review and valuable comments on our article. We apologize for not being able to effectively present the experimental results and for not using function assays to verify these results. Our research aimed to study the different statuses of TLS in HNSCC tumors employing an integrative approach that combines single-cell and spatial multi-omics technologies. Furthermore, we investigated the role of TLS in the tumor microenvironment and its impact on the tumor immune process.

We also thank you for correcting our use of "CD4 progenitor exhausted". Based on your feedback and to ensure the completeness of the related names and definitions, we have revised the manuscript and renamed all instances of "CD4⁺ Tex^{prog} cells" to "CD4⁺ Tex^{prog}/Tfh cells" in the text and figures.

Regarding your concerns about several results in our manuscript, I would like to address them in separate paragraphs below. We believe that the evidence we present, although not sufficient, is able to support the main viewpoints to a certain extent and lay the foundation for further functional experiments. We appreciate your thorough review and look forward to any further insights you may offer to help improve our manuscript.

Comment 1:

There is no evidence that supports the following conclusions: i) mTLS can generate antitumor B and T cell responses...

Response 1:

We appreciate your valuable and constructive comments. As you mentioned, our current evidence only provides some multi-omic evidence to support the idea that "mTLS can generate antitumor B and T cell responses". Our study actually lacks direct functional experiments to validate the anti-tumor effects of mTLS. I have also addressed these shortcomings in the discussion section of our paper. However, we would like to take this opportunity to present the data from multiple approaches such as scRNA-seq, scTCR/BCR-seq, and spatial transcriptomics, which support this hypothesis. We believe that these data can provide some insights into the role of mTLS in generating antitumor responses.

First, we performed scRNA-seq, scTCR/BCR-seq, and spatial transcriptomics on HNSCC samples with no TLS (nTLS), immature TLS (imTLS), and mature TLS (mTLS). Our analysis revealed that, compared to nTLS and imTLS, mTLS is enriched with CD4⁺ T cells, CD8⁺ T cells, and B cells at various stages of maturity (Fig. R1). The presence of these cells suggests that mTLS may serve as a potential source of anti-tumor immunity¹. Additionally, we observed increased accumulation of GC B cells, CD4⁺ Tex^{prog}/Tfh cells, cDC2, and mregDCs within mTLS. (Fig. R1d-f).

Fig. R1. TLS-associated heterogeneity of T and B cell states in HNSCC. a-c, Pairwise comparisons of kernel

density estimates in UMAP space. The color gradient from red to blue indicates decreasing enrichment of CD4⁺ T cells (a), B cells (b) and DCs (c) in different TLS statuses. d-f, A color gradient, transitioning from red (representing enrichment) to blue (representing depletion), encodes the log₂-transformed odds ratios, while the sizes of the depicted points are governed by the Bonferroni-adjusted -log₁₀(P values), highlighting the statistical significance of observed variations of CD4⁺ T cells (d), B cells (e) and myeloid cells (f).

scTCR/BCR-seq demonstrated that both B and T cells undergo a continuous developmental trajectory from immaturity to maturity in mTLS (Fig. R2), providing stronger evidence than trajectory analysis alone. More importantly, our spatial transcriptome analysis traced the progression of these cells from naïve to functional states within mTLS. Interestingly, we also observed that functional T cells and plasma cells radiate from the mTLS GC into the adjacent tumor tissue, further supporting the conclusion that mTLS have the local capacity to generate effector T cells and plasma cells (Fig. R3).

Fig. R2. TLS induces effects on both B cells and T cells in situ. a-c, PAGA analysis of B cells (a), CD8⁺ T cells (b) and CD4⁺ T cells (c). Each color represents a subcluster of cell. d-f, UMAP plot embedding for B cell (d), CD8⁺ T cells (e) and CD4⁺ T cells (f) subclusters, colored by clone size (upper left) and UMAP embedding in different TLS status, colored by subcluster, indicating the different clone size by dot size. g-i, Heatmap and pie chart showing the number of clonotype sharing between neighboring functional clusters of B cell (g), CD8⁺ T cells

(h) and CD4⁺ T cells (i) in different status of TLS. For heatmap, the color representing the number of shared clonotypes. For pie chart, the color representing the status of TLS and the size representing the number of shared clonotypes.

Fig. R3. Spatial distribution of intratumoral TLS throughout different TLS status of HNSCC. Spatial mapping of granulosa cell types from the scRNA-seq to spatial transcriptomics of nTLS, imTLS and mTLS HNSCC using Cell2location. Spatial mapping of lineage of B cell, CD4⁺ and CD8⁺ T cell and DC from the human scRNA-seq dataset to a respective spatial transcriptomics slide of HNSCC with nTLS, imTLS and mTLS statuses with cell2location. Estimated abundance for cell types (color intensity) to each Visium spot (color) shown over the H&E images. Scale bars = 1 mm.

We acknowledge that the limited sample size for scRNA-seq and spatial transcriptomics may introduce potential biases. To address this, we supplemented our analysis with data from a cohort of 422 FFPE HNSCC tumor samples. These results support our findings from single-cell and spatial transcriptomics, such as the enrichment of CD8⁺ and CD4⁺ subpopulations in mTLS, and the colocalization of CD8, CD4, and DC subclusters within mTLS (Fig. R4a, b). Based on other reviewers' suggestions, we also quantified these results. We demonstrated that CD4⁺TCF1⁺CXCL13⁺ T cells, B cells, DC-LAMP⁺ DCs, and CD8⁺TCF1⁺ T cells are enriched within mTLS and exhibit closer spatial interactions (Fig. R4c). These findings suggest that these cells accumulate in mTLS and form stronger

cellular interactions.

These data align with the results from our single-cell analyses and further confirm the robustness of our conclusions.

Fig. R4. Cell-cell interactions and the average distance of cell-cell interactions in mTLS. a, Images of a mIHC-

stained HNSCC tissue with different status of TLS showing co-localization of B cell (CD20), T cell (CD4, CD8 and TCF1), mregDC (DC-LAMP) and chemokine (CXCL13) in HNSCC across. **b**, The Distance between each type of cell in each core of the TMAs, with core grouped by TLS status. Vertical dotted lines indicate the median distance. Scale bars = 200 μ m. **c-h**, Ratio of cell-cell interactions within overall cellular interactions and average distance of cell-cell interactions between CD4⁺TCF1⁺CXCL13⁺ cells and CD8⁺TCF1⁺ cells (**c, d**), CD20⁺ cells (**e, f**), and DC-LAMP⁺ cells (**g, h**) within overall cellular interactions in HNSCC patient tissues with different TLS status, respectively. (nTLS = 303, imTLS = 83, mTLS = 36, two-tailed Mann-Whitney *U*-test).

Our study revealed that the capacity of mTLS can generate effector T cells and plasma cells (PC). However, this does not directly confirm their TLS-mediated anti-tumor function. Indeed, we understand that the present study is lacking of direct functional experiments to directly prove that mTLS directly generate anti-tumor T and B cell responses. We have tried our best to provided multi-dimensional evidence—through scRNA-seq, scTCR/BCR-seq, spatial transcriptomics, mIHC, and TMA—that T and B cells are present and differentiate within mTLS.

The improper description of content in our article has raised concerns by the reviewers. In order to make the article description more rigorous, we have revised the description in the manuscript. Thank you again for your thoughtful suggestions and for considering our revised manuscript. In the future, we will further explore the functional experiments to clearly explain the underlying mechanism of mTLS formation in ICB response.

Revision made:

In the revised manuscripts, we revised the description in the “Introduction Section”.

“In this study, we explored the function of mTLS as center for the initiation of local immunity in the TME.”

In the revised manuscripts, we added the limitation in the “Discussion Section”.

“Fourthly, our study does not include direct functional assays to confirm the results obtained from single-cell and spatial multi-omics technology data. Our research aimed to study the different statuses of TLS in HNSCC tumors employing an integrative approach that combines single-cell and spatial multi-omics technologies.”

Comment 2:

There is no evidence that supports the following conclusions: ... ii) follicular helper T cells activate B cells to produce plasma cells in the germinal center...

Response 2:

We would like to express our great gratitude for your insightful comments. Indeed, due to the lack of functional experiments in vivo and in vitro, we cannot fully prove this point, especially in the human tumor microenvironment. However, we have still provided some relevant evidence in this work, which we will present in the following.

First, our work demonstrates that CD4⁺ Tex^{prog}/Tfh cells, were found to be more enriched in mTLS (Fig. R5a, b), especially in the GC of the mTLS (Fig. R5c). Gene enrichment analyses further revealed that the CD4⁺ Tex^{prog}/Tfh subset was significantly enriched with features of Tfh-like functions (Fig. R5d). Notably, B cell chemotaxis and germinal center (GC) formation are vital (Fig. R5e). This evidence specifically connects the CD4⁺ Tex^{prog}/Tfh cells’ functionality with the mechanisms necessary for

activating B cells towards plasma cell differentiation within the GC.

Fig. R5. CD4⁺ Tex^{prog}/Tfh cells are able to activate B cells towards plasma cell differentiation. **a**, Left, GLM-based dot plot showing TLS status-specific enrichment of T/NK cell subclusters. A color gradient, transitioning from red (representing enrichment) to blue (representing depletion), encodes the \log_2 -transformed odds ratios, while the sizes of the depicted points are governed by the Bonferroni-adjusted $-\log_{10}(P)$ values), highlighting the statistical significance of observed variations. Middle, heatmap of average T cell state scores across T/NK cell subclusters. Right, heatmap of signaling pathway activity scores across T/NK cell subclusters. **b**, Images of a mIHC-stained different subcluster of CD4⁺ T cells in HNSCC tumor with TLS status. Regions with a high density

of central memory (CXCR5⁺CD4⁺) CD4⁺ T cells and progenitor exhausted (TCF1⁺PD-1⁺CD4⁺) CD4⁺ T cells. Scale bars = 100 μ m. **c**, Heatmap plot depicting scaled expression of genes associated with memory, exhaustion, Tfh and cytotoxicity in CD4⁺ Tconv cells. **d**, Heatmap plot depicting scaled expression of genes associated with memory, exhaustion, Tfh and cytotoxicity in CD4⁺ Tconv cells. **e**, Dot plot showing the analysis of enrichment for the four most significant Gene Ontology (GO) Biological Process terms across CD4⁺ T cell subclusters. A color gradient, transitioning from red (representing enrichment) to blue (representing depletion), encodes the Z-score normalized enrichment score, while the sizes of the depicted points are governed by the Benjamini-Hochberg-adjusted $-\log_{10}(P$ values), highlighting the statistical significance of observed variations.

Subsequently, our study identified all major stages of B cell maturation in mTLS within HNSCC (Fig. R6a-c). Through pseudotemporal ordering and differentiation trajectory analyses, our study highlighted a systematic progression from naive B cells towards plasma cells within mTLS, reinforcing the role of this specific TME in facilitating B cell maturation and subsequent differentiation into plasma-producing cells (Fig. R6d, e). Crucially, GC B cells annotated with higher levels of BCL6, LMO2, RGS13, and HMCES (Fig. R6f) exhibit increased potential to reside in the GC or differentiate into plasma cells, showcasing a clear pathway for B cell maturation mediated by the GC environment within mTLS.

Fig. R6. B cell maturation mediated by the germinal center environment within mTLS. **a**, UMAP plot of B cell subclusters identified from scRNA-seq. Subclusters are numbered and colored by identity: B cells (clusters 1–6) and plasma cells (clusters 7–9). **b**, Left, GLM-based dot plot showing TLS status-specific enrichment of B cell subclusters. A color gradient, transitioning from red (representing enrichment) to blue (representing depletion), encodes the \log_2 -transformed odds ratios, while the sizes of the depicted points are governed by the Bonferroni-adjusted $-\log_{10}(P$ values), highlighting the statistical significance of observed variations. Middle, heatmap of signaling pathway activity scores across B cell and plasma cell subclusters. Right, heatmap of normalized expression of genes associated with immunoglobulin heavy chain constant region, which correlate with types of Ig across B cell subclusters. **c**, Pairwise comparisons of kernel density estimates in UMAP space. The color

gradient from red to blue indicates decreasing enrichment of B cells and plasma cells in different TLS statuses. **d**, PAGA analysis of B cells and plasma cells. Each color represents a subcluster of B cell or plasma cell. **e**, Left, Schematic illustrates that as B cells differentiate from naive B cell to GC B cell, the maturation progressively increased, and the GC reaction time was extended. Right, scaled module scores within these subclusters of B cells with respect to two pseudotime trajectories and genes associated with activation, class switch recombination machinery and class switch recombination interactors. **f**, UMAP plots showing normalized expression profiles of cell-type-specific markers in B/plasma cell T cell subclusters.

In cell-cell communication analysis, we also found that $CD4^+$ Tex^{prog}/Tfh has the ability to attract and activate GC B cells through $CD40LG-CD40$ and $CXCL13-CXCR5$ (Fig. R7a). Leveraging spatial transcriptomics data, we pinpointed highly expressed $CXCL13-CXCR5$ and $CD40LG-CD40$ within the mTLS site (Fig. R7b, c).

Fig. R7. $CD4^+$ Tex^{prog}/Tfh attract and activate GC B cells through $CD40LG-CD40$ and $CXCL13-CXCR5$. **a**, Ligand-receptor pair expression analysis across B cells, $CD4^+$ Tconv cells, DCs and B cell subclusters in B cell zone of mTLS, showing average gene expression by scRNA-seq. **b**, spatial distributed expression of the $CXCL13-CXCR5$, $CD40L$ (encoded by $CD40LG$)- $CD40$, and $CD70-CD27$ signaling axis in different TLS status. The color gradient indicates the gene expression (red, high expression; white, low expression). Scale bars = 1 mm. **c-e**, Average enrichment score of $CD70-CD27$ (c), $CXCL13-CXCR5$ (d), and $CD40LG-CD40$ (e) in spots of relative area (nTLS, imTLS or mTLS) across different TLS status detected by spatial transcriptomics (n = 3 nTLS, n = 3 imTLS, n = 4 mTLS, one-tailed Mann-Whitney U -test; for dot plots: center line, mean; whiskers, standard deviation).

Spatial transcriptomic data in our study also reveal the co-occurrence of $CD4^+$ Tex^{prog}/Tfh and B cells. In mTLS, we delineate the progression from the naive B cell to the GC B cell and subsequently to the plasma cell, establishing a continuum from the inner GC of the mTLS to the adjacent tissue (Fig. R8a). And we also discovered a high level of enrichment in GC B cells, plasma cells, exhausted $CD4^+$ T cells, as well as DCs that were enriched in mTLS (Fig. R8b-k).

Fig. R8. Spatial distribution of intratumoral TLS throughout different TLS status of HNSCC. a, Spatial mapping of granulosa cell types from the scRNA-seq to spatial transcriptomics of mTLS HNSCC using Cell2location. Spatial mapping of lineage of B cell, CD4⁺ and CD8⁺ T cell and DC from the human scRNA-seq dataset to a respective spatial transcriptomics slide of HNSCC with mTLS statuses with cell2location. Estimated abundance for cell types (color intensity) to each Visium spot (color) shown over the H&E images. Scale bars = 1 mm. **b-k**, Average enrichment score of B.GC (**b**), plasma.cell.1 (**c**), CD4.T.prog.exh (**d**), CD4.T.term.exh (**e**), CD8.T.effector.mem (**f**), CD8.T.cytotoxic (**g**), CD8.T.exh (**h**), cDC1 (**i**), cDC2 (**j**), mregDC (**k**) spots in spots of relative area (nTLS, imTLS or mTLS) across different TLS status detected by spatial transcriptomics (n = 3 nTLS, n = 3 imTLS, n = 4 mTLS, one-tailed Mann-Whitney *U*-test; for dot plots: center line, mean; whiskers, standard deviation).

In addition, we used our patient cohort with HNSCC to further support the relationship between CD4⁺ Tex^{prog}/Tfh cells, cDC2, and GC B cells (Fig. R9). We found that both the CD4⁺TCF1⁺PD-1⁺ Tex^{prog}/Tfh cells and CD20⁺BCL6⁺ GC B cells increased in mTLS (Fig. R9a, b). Furthermore, the cell-cell interactions between CD4⁺TCF1⁺PD-1⁺ Tex^{prog}/Tfh cells and CD1C⁺ cDC2, as well as between CD1C⁺ cDC2 and CD20⁺BCL6⁺ B cells, also increased in mTLS (Fig. R9c, d). This data further indicated the close relationship between CD4⁺ Tex^{prog}/Tfh cells and B cells.

Fig. R9. Quantification of mIHC-stained different subclusters of BCL6⁺CD20⁺ B cells, CD4⁺TCF1⁺PD-1⁺ Tex^{prog} cells in HNSCC tumor with different TLS status. a, b, Ratio of CD4⁺TCF1⁺PD-1⁺ cells (a) and CD20⁺BCL6⁺ cells (b) in total cells of HNSCC patient tissues with different TLS status, respectively. c, d, Ratio of CD4⁺TCF1⁺PD-1⁺ cells and CD1c⁺ cells (c), CD20⁺BCL6⁺ cells and CD1c⁺ cells (d) interactions within overall cellular interactions, respectively. (nTLS = 303, imTLS = 83, mTLS = 36, two-tailed Mann-Whitney *U*-test).

In summary, our manuscript presents a suite of evidence across single-cell level and spatial analyses that collectively support the role of CD4⁺ Tex^{prog}/Tfh cells in activating B cells to produce plasma cells within the germinal centers of mTLS in HNSCC. This evidence addresses the feasibility and mechanisms through which these specific T cell subsets contribute to the B cell maturation and differentiation processes critical for effective anti-tumor immune responses. More importantly, we have added the relevant description in the "Discussion Section" to provide a more rigorous explanation of our findings.

Revision made:

In the revised manuscripts, we added the limitation in the "Discussion section".

"Fourthly, our study does not include direct functional assays to confirm the results obtained from single-cell and spatial multi-omics technology data. Our research aimed to study the different statuses of TLS in HNSCC tumors employing an integrative approach that combines single-cell and spatial multi-omics technologies."

Comment 3:

There is no evidence that supports the following conclusions: ...iii) DC-LAMP⁺ dendritic cells support the activation of CD8⁺ T cells...

Response 3:

Thank you for your critical comments. Indeed, one of the shortcomings of the present study is the lack of direct evidence on DC-LAMP⁺ DC (mregDC) support for the activation of CD8⁺ T cells. In this study, by multiple datasets and multi-omics, we examined the close relationship between mregDCs and stem-like CD8⁺ T cells in mTLS from different perspectives to support our findings. Recent studies have shown that T cell immune responses activated by mregDC play a crucial role in the antitumor immunity of TLS^{2,3}. We are thankful for the opportunity to clarify and bolster our findings, detail are as follows.

As mentioned in Response 1, we identified an enrichment of DC-LAMP⁺ mregDCs and stem-like CD8⁺ T cells in mTLS using scRNA-seq and spatial transcriptomics, and mIHC in a small and large HNSCC patient cohort, respectively. Specifically, scRNA-seq data showed that mregDCs and stem-like CD8⁺ T cells were highly enriched in mTLS, suggesting their involvement in mTLS formation and

function (Fig. R10a, b). Further cell interaction analysis revealed that mregDCs expressed high levels of CCR4 ligands (*CCL22*, *CCL17*), the central memory T cell chemokine ligand *CCL19*, co-stimulatory genes (*CD80*, *CD86*, *PVR*, *ICOSLG*, *CD40*), and cytokines such as *IL12B* and *IL15*, which are known to modulate T cells (Fig. R10c). These results indicate that mregDCs interact with stem-like CD8⁺ T cells, facilitating their activation and conversion to functional CD8⁺ T cells.

Fig. R10. The enrichment of mregDC in mTLS and cell interaction analysis between mregDC with CD8⁺ T cells. **a**, GLM-based dot plot showing TLS status-specific enrichment of myeloid cell subclusters. A color gradient, transitioning from red (representing enrichment) to blue (representing depletion), encodes the \log_2 -transformed odds ratios, while the sizes of the depicted points are governed by the Bonferroni-adjusted $-\log_{10}(P)$ values, highlighting the statistical significance of observed variations. **b**, Dot plot of scaled marker gene expression (averaged per cluster) for coarse-grained T cell subclusters, showing differentially expressed genes in columns and clusters in rows. Genes are grouped by cluster and five genes per cluster that are highlighted. A color gradient represents the mean expression within each of the subcluster of T/NK cells, while the sizes of the depicted points indicate the fraction of cells in the categories expressing a gene. **c**, Ligand–receptor pair expression analysis across B cell, CD4⁺ Tconv cell and DC subclusters in in T cell zone of mTLS, showing average gene expression by scRNA-seq.

To further validate the interaction between mregDCs and stem-like CD8⁺ T cells in mTLS, we observed in spatial transcriptomics data that these cells were closely localized within the T cell zone of mTLS, with strong spatial correlation (Fig. R11a, b). Additionally, we observed *CD70* – *CD27* interactions between mregDCs and stem-like CD8⁺ T cells, further supporting the role of mregDCs in activating these cells in mTLS (Fig. R11c, d).

Fig. R11. Spatial distribution data between mregDCs and stem-like CD8+ T cells in mTLS. **a**, Spatial mapping of granulosa cell types from the scRNA-seq to spatial transcriptomics of mTLS HNSCC using Cell₂location. Spatial mapping of lineage of B cell, CD4⁺ and CD8⁺ T cell and DC from the human scRNA-seq dataset to a respective spatial transcriptomics slide of HNSCC with mTLS statuses with cell₂location. Estimated abundance for cell types (color intensity) to each Visium spot (color) shown over the H&E images. Scale bars = 1 mm. **b**, Average enrichment score of mregDC, CD8.T.effector.mem, CD8.T.cytotoxic, CD8.T.exh spots in spots of relative area (nTLS, imTLS or mTLS) across different TLS status detected by spatial transcriptomics (n = 3 nTLS, n = 3 imTLS, n = 4 mTLS, one-tailed Mann-Whitney *U*-test). **c**, Spatial distributed expression of the CD70-CD27 signaling axis in different TLS status. The color gradient indicates the gene expression (red, high expression; white, low expression). Scale bars = 1 mm. **d**, Average enrichment score of CD70-CD27 in spots of relative area (nTLS, imTLS or mTLS) across different TLS status detected by spatial transcriptomics (n = 3 nTLS, n = 3 imTLS, n = 4 mTLS, one-tailed Mann-Whitney *U*-test).

To reduce potential bias from small sample sizes, we also performed mIHC staining on the larger cohort of HNSCC samples. We again found that DC-LAMP⁺ cells were in close proximity to CD8⁺ TCF1⁺ cells within mTLS (Fig. R12).

Fig. R12. The representative mIHC images and quantification of DC-LAMP⁺ cells and CD8⁺TCF1⁺ cells. a, Images of a mIHC-stained HNSCC tissue with different status of TLS showing co-localization of B cell (CD20), T cell (CD4, CD8 and TCF1), mregDC (DC-LAMP) and chemokine (CXCL13) in HNSCC across. **b,** The Distance between each type of cell in each core of the TMAs, with core grouped by TLS status. Vertical dotted lines indicate the median distance. Scale bars = 200 μm . **c-h,** Ratio of cell-cell interactions within overall cellular interactions and average distance of cell-cell interactions between CD4⁺TCF1⁺CXCL13⁺ cells and CD8⁺TCF1⁺ cells (**c, d**), CD20⁺ cells (**e, f**), and DC-LAMP⁺ DCs (**g, h**) within overall cellular interactions in HNSCC tissue with different TLS status, respectively. (nTLS = 303, imTLS = 83, mTLS = 36, two-tailed Mann-Whitney *U*-test).

In summary, our results suggested that mregDCs activate stem-like CD8⁺ T cells in mTLS, contributing to antitumor T cell immunity. More importantly, we have added the relevant description in the "Discussion Section" to provide a more rigorous explanation of our findings. Thank you again for your thoughtful suggestions and for considering our revised manuscript.

Revision made:

In the revised manuscripts, we added the limitation in the "Discussion section".

"Fourthly, our study does not include direct functional assays to confirm the results obtained from single-cell and spatial multi-omics technology data. Our research aimed to study the different statuses of TLS in HNSCC tumors employing an integrative approach that combines single-cell and spatial multi-omics technologies."

Comment 4:

There is no evidence that supports the following conclusions: ...iv) imTLS cannot produce in situ anti-tumor immunity due to obstruction by immunosuppressive cells, such as regulatory T cells (Tregs) and M2 macrophage.

Response 4:

We appreciate your critical feedback and comment. We are thankful for the opportunity to clarify and bolster our findings, detail are as follows.

Our data found a greater infiltration of M2 macrophages and Tregs, and a lack of CD4 Tex^{prog} in imTLS (Fig. R13a, b, d). This cellular composition suggests an immunosuppressive microenvironment that may hinder effective anti-tumor immunity. Moreover, spatial transcriptomics showed Tregs exhibit a close spatial association with both CD8⁺ and CD4⁺ T cells within imTLS (Fig. R13c). This proximity implies that Tregs may directly impede the local anti-tumor immune response within the tumor microenvironment, as they are positioned to exert their suppressive effects on neighboring T cells. Several studies have reported similar findings. For instance, Joshi et al. demonstrated that Tregs within tumor-associated TLSs can suppress antitumor T cell responses⁴. Additionally, the presence of Tregs and M2 macrophages in TLS has been associated with poor prognosis in patients with colorectal liver metastases⁵. This is consistent with the known immunosuppressive functions of these cell types, which can dampen T cell activation and proliferation, thereby limiting the efficacy of the anti-tumor immune response. On the basis of the lack of functional experiments, our conclusions are based on speculation. In the future, and the specific mechanism needs to be further verified.

Fig. R13. The infiltration co-occurrence of M2 macrophages and Tregs. a, UMAP plot embedding for CD4⁺ T cell subclusters, colored by clone size (upper left) and UMAP embedding for CD4⁺ T cell subclusters in different TLS status, colored by subcluster, indicating the different clone size by dot size. **b**, Boxplot showing fraction

clonotypes of Treg in total CD4⁺ T cell clonotypes across different TLS status (n = 4 nTLS, n = 4 imTLS, n = 5 mTLS, one-tailed Mann-Whitney *U*-test; for box plots: box center line, median; box limits, upper and lower quartiles; box whiskers, maximum and minimum values). **c**, Heatmap of co-occurrence of B cells, CD4⁺ T cells, CD8⁺ T cells and DCs across imTLS status. **d**, GLM-based dot plot showing TLS status-specific enrichment of myeloid cell subclusters. A color gradient, transitioning from red (representing enrichment) to blue (representing depletion), encodes the log₂-transformed odds ratios, while the sizes of the depicted points are governed by the Bonferroni-adjusted -log₁₀(*P* values), highlighting the statistical significance of observed variations.

More importantly, we have added the relevant description in the "Discussion Section" to provide a more rigorous explanation of our findings.

Revision made:

In the revised manuscripts, we added the limitation in the "Discussion section".

"Fourthly, our study does not include direct functional assays to confirm the results obtained from single-cell and spatial multi-omics technology data. Our research aimed to study the different statuses of TLS in HNSCC tumors employing an integrative approach that combines single-cell and spatial multi-omics technologies."

Comment 5:

The authors always explore the data as following: cell clustering, then enriched genes/pathways, followed by pseudotime analysis, cell abundance according to the TLS status, and then clonotype analysis when possible. Data would perhaps be more robust if analyses were not biased by this formula.

Response 5:

We appreciate your concerns regarding the structure of our data analysis. We apologize for not being able to organize the article's structure better, making the data more reliable and easier for readers to understand.

In this study, our goal was to investigate the different statuses of TLS in HNSCC tumors using an integrative approach that combines single-cell and spatial multi-omics technologies. Due to the various data modalities, we first aimed to present an initial overview of the expression and function of the different cell types in TLS using a formula. We then further analyzed each cell type and their specific data (such as scTCR/BCR-seq) to comprehensively display the characteristics of TLS status in HNSCC. Additionally, we have also included evidence from other omics in our data to support each aspect. However, this has led to some labeling unclear during the presentation due to the abundance of data, which may have caused confusion for readers. We have thoroughly checked and taken necessary measures to avoid this situation from happening again. Furthermore, we have optimized the presentation order of the data, such as rearranging the paragraphs, to make better understanding by reader.

We sincerely thanks for your valuable feedback and guidance during the review process, which have significantly contributed to enhancing the quality and clarity of our study.

Reviewer #2 (Remarks to the Author):

Comment 1:

Do mTLS patients contain imTLS or imTLS patients have a small percentage of mTLS? When stratifying patients by TLS, what is the cutoff point of % mTLS or imTLS?

Response 1:

We sincerely thank the reviewer for raising the important scientific questions. Based on our current data and existing literature reports, we will answer your questions as follows.

In this study, we followed the classic classification criteria for imTLS and mTLS cases outlined by Vanhersecke et al.⁶, which is a widely accepted method referenced in many TLS studies^{7, 8}. Specifically, if a sample contains only imTLS without mTLS, it is classified as imTLS. If a sample contains mTLS or both mTLS and imTLS (even when the mTLS proportion is lower), it is defined as mTLS. In our study, we found that 16 out of 36 samples exhibited both mTLS and imTLS. This finding aligns with previous report⁶, in which 16 out of 41 mTLS tumor samples showed imTLS structures in a patient cohort including carcinomas and sarcomas derived from different organs.

To optimize the determination of TLS status, we have improved the traditional detection method. Unlike most previous studies that only examined ONE section of tissue using ONE detection method (HE, IHC, or mIHC)^{9, 10, 11}, we took multiple sections from different tumor depths. In addition to sampling from the largest diameter of the tumor, we also collected sections from 1/4 and 3/4 of the tumor depth. These THREE sections were used together to detect the presence and status of TLS. This approach serves to avoid detection bias. Then, the sections to concurrently assess TLS status, following the methods of previous studies^{10, 12}. In order to improve accuracy in evaluating TLS status, we utilized THREE methods including HE, IHC, and mIHC for each sample. Compared to traditional standards, we implemented a more comprehensive and rigorous evaluation process with the goal of accurately identifying mTLS within tumors and minimizing bias.

Furthermore, our analysis of spatial transcriptome to examine the TLS imprint which contains gene clusters associated with the mTLS signature (Fig. 1i) and scRNA-seq data (Extended Data Fig. 6g-i) to further validate the TLS status about immature and mature. Thus, we considered the reasonability of the results in samples consistent with our evaluation criteria. According to reviewer's suggestion, we added the detailed description of detection of TLS status in the revised manuscript highlighted these changes in BLUE, with details as following.

Revision made:

In the revised manuscript, the related method content has been added as following.

“IHC and mIHC to determinethe status of TLS. For hematoxylin and eosin staining, OCT-embedded tissues were cut into 6-8 μm sections, separated from the largest diameter of the tumor. Sections were also taken from 1/4 depth and 3/4 depth of the tumor and stained using a Leica automated slide stainer. The TLS status was then further determined by IHC and mIH^{56,57,58}. CD20, CD3, and CD23 immunostaining were used to determine the status of TLS as previously described¹⁶. Briefly, IHC assay were performed using anti-human CD20 (Cell Signaling Technology), CD3 (Abcam), and CD23 (Abcam) antibodies, as previously described⁹. Then, the multiplex immunofluorescence staining was conducted using the Opal 6-Plex Manual Detection Kit (Akoya Biosciences) to further detect the TLS status as previously describe⁹. Briefly, FFPE tissue slides were first deparaffinized and then incubated

sequentially with primary antibodies CD20 (1:1000, Cell Signaling Technology), CD4 (1:1000, Abcam), CD8 (1:800, Cell Signaling Technology), BCL6 (1:600, Cell Signaling Technology), CD23 (1:900, Abcam), TCF1 (1:600, Cell Signaling Technology). This was followed by incubation with secondary antibodies (Akoya Biosciences) and corresponding reactive Opal fluorophores. Nuclei acids were stained with DAPI. Seven-color stained slides were scanned using a Vectra 3 Imaging System (Akoya Biosciences). Scanning was conducted at a magnification of 20×. The channel employed for multispectral imaging included DAPI, FITC, Cy3, Texas Red and Cy5. The scanned data is subsequently processed by the inForm software (v.2.4, Akoya Biosciences) for analysis of multispectral imaging. Autofluorescence were acquired by assess the region of the slides that not fluorophores. mTLS is GC formation, a dynamic region with a network of CD23⁺ FDCs and CD20⁺BCL6⁺ B cells^{6,10}. The evaluation criteria for TLS status are as follows as previous study described⁵⁶, if a sample contains only imTLS and not mTLS, it will be classified as imTLS. If a sample contains mTLS or both mTLS and imTLS, it will be defined as mTLS.”

.....

“6. Petitprez, F., et al. B cells are associated with survival and immunotherapy response in sarcoma. *Nature* 577, 556-560 (2020).

9. Li, H., et al. Tertiary lymphoid structure raises survival and immunotherapy in HPV(-) HNSCC. *J Dent Res* 102, 678-688 (2023).

10. Fridman, W. H., et al. Tertiary lymphoid structures and B cells: An intratumoral immunity cycle. *Immunity* 56, 2254-2269 (2023).

16. Vanhersecke, L., et al. Mature tertiary lymphoid structures predict immune checkpoint inhibitor efficacy in solid tumors independently of PD-L1 expression. *Nat Cancer* 2, 794-802 (2021).

56. Vanhersecke, L., et al. Standardized Pathology Screening of Mature Tertiary Lymphoid Structures in Cancers. *Lab Invest* 103, 100063 (2023).

57. Haddox, C. L., et al. Phase II Study of Eribulin plus Pembrolizumab in Metastatic Soft-tissue Sarcomas: Clinical Outcomes and Biological Correlates. *Clin Cancer Res* 30, 1281-1292 (2024).

58. Flippot, R., et al. B cells and the coordination of immune checkpoint inhibitor response in patients with solid tumors. *J Immunother Cancer* 12, (2024).”

Comment 2:

Please comment on this. If mTLS patients have imTLS, even though the number of imTLS is low in mTLS patients, vice versa for imTLS patients, analyzing mixing cells from different types of TLS patients may potentially produce false results.

Response 2:

We sincerely thank the reviewer for their valuable comment. Based on Comment 1, we added the details of the classification criteria for imTLS and mTLS cases. In this study, we followed the classic classification criteria for imTLS and mTLS cases outlined by Vanhersecke et al.⁶. Simply, if a sample contains only imTLS without mTLS, it is classified as imTLS; If a sample contains mTLS or both mTLS and imTLS (even when the mTLS proportion is lower), the sample is defined as mTLS. Indeed, considering the traditional definition of TLS status, mTLS may contains imTLS. In our patient cohort of HNSCC, 16 out of 36 samples exhibited both mTLS and imTLS. Similarly, in the previous study, they reported that 16 out of 41 mTLS tumor samples showed imTLS structures in a patient cohort including carcinomas and sarcomas derived from different organs⁶.

Therefore, in our study, we strictly selected tumor samples containing only mTLS, with no imTLS present. Under the strictly rule mentioned above, we performed scRNA-seq, scTCR/BCR-seq, and spatial transcriptomics on these “mTLS” samples. We applied this strictly method to minimize potential bias from mixed detection of mTLS and imTLS within the same sample.

Comment 3:

The description for results is rather simple and, sometimes, too brief. It will be much better to expand the result description to provide enough information to help a better understanding on experiments, data interpretation, etc.

Response 3:

We sincerely thank the reviewer for their careful review. Based on your suggestion, we added the result description to provide enough information to help a better understanding the experiments and highlight in BLUE in the revised manuscript.

Comment 4:

Fig. 1c. What does this figure intend to tell? It will be better to provide an interpretation for this figure in the Result section. Also, it is very hard to distinguish which is which based on patient ID and cell type samples using the different colors, which makes this figure less meaningful. For example, nTLS may be completely absent from B cell clusters of B and Mast cells shown in Figure 1e. The authors failed to provide explanation to those presumably interesting results.

Response 4:

We sincerely thank the reviewer for the insightful comment. We have carefully addressed each point as below.

Figure 1c is a common plot used in scRNA-seq analysis to show patient data sources and sample processing methods. This type of display is often seen in single-cell studies^{13, 14}. We apologize for the lack of clear explanation in the image, as mentioned by the reviewer. The use of confusing colors and unclear explanations decreased its effectiveness. Based on your suggestion, we have revised Fig. 1c by adjusting the color scheme to better differentiate between patient IDs and cell type samples. Each color now represents a specific patient, with higher saturation indicating sorted CD45⁺ cells and lower saturation representing unsorted tumor cells (Fig. R14).

Fig. R14. (Fig. 1c) UMAP plot colored by patients (n=14) from whole tumor cells or sorted CD45⁺ cells profiled by scRNA-seq.

According to reviewer's suggestion, we have added this context in the main text.

Revision made:

In the revised manuscript, we added the description of “Fig. 1c” as following.

“All of the single cells were separated from a set of 14 tumor biopsies, from different treatments, using unsorted tumor tissue and sorted CD45⁺ cells (Fig. 1c).”

Meanwhile, we appreciate your observation regarding the limited presence of B cells and mast cells in nTLS, which is an interesting phenomenon. Our analysis is consistent with previous reports that indicate B cell clusters as major effector cells in mTLS but are found to be scarce in nTLS^{1, 15}. However, the quantity and function of mast cells in TLS have not been reported previously. Mast cells have been shown to possess pro-inflammatory and chemotactic abilities¹⁶, but it has not been reported whether the lack of mast cells in nTLS is due to the extremely cold immune environment. According to the suggestion of the reviewer, we have added this context in the main text.

Revision made:

In the revised manuscript, we added the explanation to those presumably interesting results about B cells and mast cells as following.

“Interestingly, we observed mast cells, were scarce in nTLS but highly enriched in mTLS (Fig. 1g). While the relationship between mast cells and TLS has not been previously reported, recent research suggests that mast cells may recruit cytotoxic T cells through CCL2-CCR2 axis¹⁶. This finding indicates that mast cells enriched in mTLS may possess similar pro-inflammatory and chemotactic properties.”

.....

“18. Fan, F., et al. Elevated mast cell abundance is associated with enrichment of CCR2⁺ Cytotoxic T Cells and Favorable Prognosis in Lung Adenocarcinoma. Cancer Res 83, 2690-2703 (2023).”

Comment 5:

Fig. 1d. It will be better to use a different and highly contrasted color combination to make these two sample types distinguishable much easily. Again, what is the figure going to tell? Is the figure to tell that the sample types are different?

Response 5:

We thank the reviewer for the valuable suggestion. As recommended, we have used a highly contrasted color to make these two sample types distinguishable much easily in Fig. 1d. We have revised the Fig. R2 to replace by the Fig. 1d in the revised manuscript with details as following.

Meanwhile, this figure was colored by cell origin and cell number according to whole tumor cells or sorted CD45⁺ cells for all samples by UMAP plot. It intended to show the distribution of single cell data from two cell types. And the Fig. 1d is the different sample types.

Revision made:

In the revised manuscript, we revised the “Fig. R15” to replace by the “Fig. 1d”.

Fig. R15. (Fig. 1d) UMAP plot colored by origin of the cells and the number of cells, either from whole tumor cells ($n = 123,432$) or sorted $CD45^+$ cells ($n = 124,904$).

Comment 6:

The Result section mentioned Fig. 1h, 1i, but they are missing in figures.

Response 6:

Thank you for pointing this out issue. We apologized for the label missing in this Fig. 1h, 1i. According to your suggestion, we re-checked the manuscript carefully and added the Result section mentioned of Fig. 1h and 1i.

Revision made:

In the revised manuscript, we added the “Fig. 1h, 1i” label in “Fig. 1” and the related description are as following.

Fig. R16. (Fig. 1g-i), g, The cell types for scRNA-seq of whole tumor cells and the composition difference of $CD45^+$ cells (g), H&E and IHC of tumor slides (h), and spatial transcriptomics of tumor tissue (i) in nTLS, imTLS, and mTLS statuses.

Comment 7:

Fig. 2b presented a lot of results, but the authors did not describe and explain the results. For examples, why some T-cell immune types and signaling components are highly expressed in one subset (cluster), but down-regulated in others.

Response 7:

We thank the reviewer for the valuable suggestion. We are sorry insufficient explanation of results in Fig.2b. We have added the detailed result as follows.

Revision made:

In the revised manuscript, we have stated the results section in Fig. 2b with details as following.

“The enrichment analysis revealed that clusters with naive or memory characteristics (clusters 1-2, 9-10, 15) were enriched in the naive T cell signature, while those with effector features (clusters 3, 22-28) were enriched in the cytotoxic T cell signature category (Fig. 2b). Clusters annotated as cycling (clusters 6-8) were enriched in the proliferation T cell signature category, and exhausted clusters exhibiting varying degrees of exhaustion-related gene expression (clusters 4, 12, 13, 16-20) were enriched in different exhausted categories (Fig. 2b). The cell pathway activity scores demonstrated differences in signaling pathway activities across clusters. T cell subsets with high expression of interferon stimulated genes (ISG) (clusters 5, 14, 21) showed enhanced activity in the JAK-STAT pathway, while cycling T/NK cells (clusters 6-8) exhibited increased activity in the MAPK pathway (Fig. 2b), consistent with previous reports²³. Treg subsets (clusters 16-21) displayed enhanced activity in the TNFa pathway (Fig. 2b), aligning with earlier findings²⁴.”

.....

*“23. Zhang, W., Liu, H. T. MAPK signal pathways in the regulation of cell proliferation in mammalian cells. *Cell Res* 12, 9-18 (2002).*

*24. Jung, M. K., et al. Tumor necrosis factor and regulatory T cells. *Yonsei Med J* 60, 126-131 (2019).”*

Comment 8:

Fig. 2c. Using TCF7, CCR7 and IL-7R to define stem-like memory T cells (Tscm) is not accurate. These genes are expressed by naïve or central memory cells as well. CD95, CD122 and CXCR3 should be included in the characterization.

Response 8:

We would like to extend our gratitude to the reviewers for thorough review and valuable suggestions. We apologize for not providing a more detailed explanation of the definition of stem-like T cells and referencing supporting literature.

In our study, we have categorized two clusters - cluster 1 labeled as “naive/central memory T cells” and cluster 2 labeled as “effector memory T cells” - as possessing a “stem-like” phenotype due to their elevated expression of memory/stem-like molecules (*TCF7*, *CCR7*, and *IL7R*). This definition has been widely used in several previous studies^{3, 17, 18} at the single-cell level.

However, we consider that the stem-like T cells described in our article are distinct from T memory stem cells (Tscm). According to existing literature, the differences between these two T cell subtypes lie in their developmental stage and the specific markers used for their definition. Tscm possess characteristics of conventional memory T cells, falling between naive T cells and effector memory T cells^{19, 20}. On the other hand, stem-like T cells encompass a broader concept that includes both naive T cells and effector memory T cell subsets with stem-like properties. In our scRNA-seq analysis, the stem-like T cell cluster was characterized by high expression levels of TCF7, IL-7R, and CCR7. While Tscm, a specific T-cell subset derived from naive T cells, not only highly express CCR7 and IL-7R, but also express high levels of CD95, CD122, and the chemokine receptor CXCR3^{21, 22, 23}.

Based on your suggestion, we have reviewed our data on the expression levels of *CD95 (FAS)*, *CD122 (IL2RB)*, and *CXCR3* in stem-like T cells. We have found that these markers are not significantly enriched in the T cells defined as stem-like by us (Fig. R17).

Fig. R17. UMAP plots showing normalized expression profiles of *FAS*, *IL2RB*, *CXCR3*.

Therefore, we consider that the stem-like T cells defined in our work are not equivalent to Tscm. Currently, most studies showed that stem-like T cells are closely associated with TLS formation^{24, 25}, and our study mainly focuses on these important role of stem-like T cells in mTLS. Your comments were vital in improving our manuscript, and we have now included citations to relevant literature to avoid any confusion among readers.

Revision made:

To prevent misinterpretation by readers, we added references about the definition of “stem-like T cells”.

“To investigate whether cytotoxic CD8⁺ T cells are produced within the local TME, we categorized clusters 1 and 2, characterized by elevated expression of memory/stem-like molecules (*TCF7*, *CCR7* and *IL7R*), as possessing a ‘stem-like’ phenotyp^{26,27}.”

.....

“26. Eberhardt, C. S. et al. Functional HPV-specific PD-1(+) stem-like CD8 T cells in head and neck cancer. *Nature* 597, 279-284, doi:10.1038/s41586-021-03862-z (2021).

27. Cardenas, M. A. et al. Differentiation fate of a stem-like CD4 T cell controls immunity to cancer. *Nature* 636, 224-232, doi:10.1038/s41586-024-08076-7 (2024).”

Comment 9:

Fig. 2c. the authors mentioned ISG in the Results section, but did not show this gene expression in Fig. 2c. The same as the description of other results in the Results section that Fig. 2c does not show any specific genes. It seems like that the Fig. 2b and 2c were flipped and mixed up.

Response 9:

We sincerely thank the reviewer for their careful observations, and apologize for the oversight regarding the result description of Fig. 2c and reverse order of the Fig. 2b and 2c. Following the reviewer’s suggestion, we have added the data on the gene expression in Fig. 2c and revised the order of the Fig. 2c.

Revision made:

In the revised manuscript, we have clearly stated the gene expression in Fig. 2c with details as following.

“Among the CD4⁺ T subclusters, almost *all* phenotypes (clusters 9–13) were found to be more enriched in mTLS, moderately enriched in imTLS, and less abundant in nTLS, except for the *cluster 14 referring to interferon-stimulated gene (ISG) T cell phenotype (Fig.2b, c)*. On the other hand, Treg clusters (clusters 15–21) were found to be reduced in mTLS compared to non-mTLS (nTLS/imTLS),

except for the memory Treg (Fig.2b, c). NK cells (clusters 22–26) were more abundant in nTLS and imTLS, with varying levels of activity among the different NK cell phenotypes (Fig.2b, c).”

Comment 10:

It seems like that “cycling T/NK cells” labeling is missing in Fig. 2b.

Response 10:

Thank you for pointing out the missing “cycling T/NK cells” label in Figure 2b. We intentionally omitted this label in order to maintain clarity and avoid overcrowding the figure, which could negatively impact the layout and readability. The cluster for cycling T/NK cells is relatively small and distinguishable within the figure, so we decided to prioritize maintaining a clear and concise presentation for the other key information depicted in the figure. We hope this explanation clarifies our decision, and we appreciate your understanding.

Comment 11:

Fig. 2d, 2e. It will be better to perform the same analysis in imTLS and nTLS as a comparison.

Response 11:

First, thank you very much for your comment. In Figure 2d, we conducted trajectory and pseudotime analysis on all CD8⁺ T cells without distinguishing TLS status, following the precedent set in studies on T cells (such as those by Zemin Zhang T cells in Science and Nature Cancer)^{26, 27}. Based on the reviewer's suggestion, we categorized the CD8⁺ T cells according to their TLS status and performed trajectory and pseudotime analyses for each of the three groups: nTLS, imTLS, and mTLS. We found that the developmental trajectories of CD8⁺ T cells were largely consistent across these groups, as was the trend in the expression of related genes identified in the pseudotime analysis.

Revision made:

In the revised manuscript, we added the “Fig. R18” about PAGA and pseudotime analysis in nTLS, imTLS, and mTLS of HNSCC.

Fig. R18. PAGA and pseudotime analysis throughout different TLS status of HNSCC. PAGA analysis of CD8⁺ T cells in nTLS (a), imTLS (c), and mTLS (e). Each color represents a subcluster of CD8⁺ T cell. UMAP plot colored by pseudotime across subclusters of CD8⁺ T cells in nTLS (b), imTLS (d), and mTLS (f). Scaled module scores within these subclusters of CD8⁺ T cells with respect to pseudotime and stem-like, functional and exhausted markers.

Comment 12:

The description for Fig. 2f should move up based on alphabetical order.

Response 12:

We sincerely thank the reviewer for such careful reading. As the reviewer suggested, we adjusted the order of the Figure based on alphabetical order (Fig. R19).

Revision made:

In the revised manuscript, the “Fig. 2f” was changed to “Fig. 2d”, “Fig. 2d” was change to “Fig. 2e”, and “Fig.2e” was change to “Fig. 2f”, with details as following.

Fig. R19. (Fig. 2d-f), d, Images of a mIHC-stained different subclusters of CD8⁺ T cells in HNSCC tumor with different TLS status. Regions with a high density of memory (CCR7⁺CD8⁺), stem-like (TCF1⁺CD8⁺), cytotoxic (Granzyme B⁺CD8⁺) and exhausted (PD-1⁺CD8⁺) CD8⁺ T cells. Scale bars = 100 μ m. **e**, PAGA analysis of CD8⁺ T cells. Each color represents a subcluster of CD8⁺ T cell. **f**, UMAP plot colored by pseudotime across subclusters of CD8⁺ T cells. Scaled module scores within these subclusters of CD8⁺ T cells with respect to pseudotime and stem-like, functional and exhausted markers.

Comment 13:

The description for Extended Data Fig. 6-7 is missing in the Results section.

Response 13:

We sincerely thank the reviewer for their careful reading. We apologize for omitting the description for Extended Data Fig. 6-7 in the Results section. We added new Extended Data Figures during the revision process. The previous Extended Data Fig. 6-7 have now become “Extended Data Fig. 13”, and “Extended Data Fig. 15”. Detailed results description has been added in the revised manuscript, with details as following.

Revision made:

In the revised manuscript, we added the related description as following.

“We then subclustered 27,622 cells that were manually annotated as B cells and plasma cells, subsequently obtaining 9 distinct subclusters based on RNA expression profiles of established markers (Fig. 4a, Extended Data Fig. 13 and Supplementary Table 3).”

“We further focused on the myeloid cell compartment and re-clustered 70,093 cells into 4 main clusters of myeloid cells with 16 subclusters, including DCs, macrophages, neutrophils, and mast cells (Fig. 5a and Extended Data Fig. 15).”

Comment 14:

Fig. 2g. The authors combined cluster 1 (naïve/central memory) and 2 (effector memory) and named it as CD8⁺ T stem-like (Tscm). The literatures say that Tscm is a T-cell subtype on its own and is different from naïve or central memory or effector memory. Please comment on this.

Response 14:

We deeply appreciate the reviewer's insightful comment. We, again, apologize for not providing a

more detailed explanation of the definition of stem-like T cells and for not referencing supporting literature. As mentioned in our Comment 8, we labeled cluster 1 as “naive/central memory T cells” and cluster 2 as “effector memory T cells,” defining them as having a 'stem-like' phenotype due to their elevated expression of memory/stem-like molecules (TCF7, CCR7, and IL7R). This definition has been widely used in several previous studies^{3, 17, 28} at the single-cell level.

We consider that the stem-like T cells described in our study are distinct from T memory stem cells (Tscm). According to existing literature, the differences between these two T cell subtypes lie in their developmental stage and the specific markers used for their definition. Tscm possess characteristics of conventional memory T cells, falling between naive T cells and effector memory T cells^{19, 20}. In contrast, stem-like T cells encompass a broader concept that includes both naive T cells and effector memory T cell subsets with stem-like properties. In our scRNA-seq analysis, the stem-like T cell cluster was characterized by high expression levels of TCF7, IL-7R, and CCR7. On the other hand, Tscm, a specific T-cell subset derived from naive T cells, not only highly express CCR7 and IL7R, but also express high levels of CD95, CD122, and the chemokine receptor CXCR3^{21, 22, 23}.

Therefore, we consider that the stem-like T cells described in our work are not equivalent to Tscm. Currently, most studies showed that stem-like T cells are closely associated with TLS formation^{24, 25}, and our study mainly focuses on these important role of stem-like T cells in mTLS. Your comments were essential in improving our manuscript, and we have now included citations to relevant literature to avoid any confusion among readers.

Revision made:

To prevent misinterpretation by readers, we added references about the definition of “stem-like T cells”.

“To investigate whether cytotoxic CD8⁺ T cells are produced within the local TME, we categorized clusters 1 and 2, characterized by elevated expression of memory/stem-like molecules (TCF7, CCR7 and IL7R), as possessing a ‘stem-like’ phenotyp^{26,27}.”

.....

*“26. Eberhardt, C. S. et al. Functional HPV-specific PD-1(+) stem-like CD8 T cells in head and neck cancer. *Nature* 597, 279-284, doi:10.1038/s41586-021-03862-z (2021).*

*27. Cardenas, M. A. et al. Differentiation fate of a stem-like CD4 T cell controls immunity to cancer. *Nature* 636, 224-232, doi:10.1038/s41586-024-08076-7 (2024).”*

Comment 15:

Fig. 2g, clonal expansion figure. What does the numbers with colors mean? It will be better to provide the interpretation for Fig. 2g. In current version, the authors only mentioned they did this experiment.

Response 15:

We sincerely thank the reviewer for the valuable suggestion. The numbers with colors in the upper left corner of the graph represent the clone size. It also shows its distribution across different subclusters and TLS statuses. Indeed, to more clearly characterize the TCR clone information, the detailed interpretation for Fig. 2g is necessary. Based on your suggestion, we added the description of Fig. 2g to refine the manuscript.

Revision made:

In the revised manuscript, the related description of Fig. 2g as following was added.

“The TCR clone information was mapped to the UMAP to facilitate the visualization of clone size and its distribution across different subclusters and TLS statuses (Fig. 2g). It was observed that mTLS status was harbored a larger number of clonotypes of CD8⁺ T cells (Fig. 2g, h).”

Comment 16:

The description for Fig. 2h should come before the description for Fig. 2i.

Response 16:

We thank the reviewer for their careful observations. We apologize for inconsistency between the text description and the Fig. 2h and 2i. In order to read the article more smoothly, we have adjusted the order of the Fig. 2h and 2i (Fig. R7) in the revised manuscript.

Revision made:

In the revised manuscript, the figures order and related description as following were modified.

“It was observed that mTLS status was harbored a larger number of expanded clonotypes of CD8⁺ T cells (Fig. 2g, h). In concordance with the result of enrichment analysis, the naive/central memory and effector memory phenotypes of CD8⁺ T cells showed enrichment for larger clone sizes in mTLS (Fig.2i).”

Fig. R20. (Fig. R2h, i), h, Boxplot showed the number of expanded clonotypes of CD8⁺ T cells in different status of TLS (n = 4 nTLS, n = 4 imTLS, n = 5 mTLS, one-tailed Mann-Whitney *U*-test; for box plots: box center line, median; box limits, upper and lower quartiles; box whiskers, maximum and minimum values). **i,** Clone size in each CD8⁺ T cell subclusters separated by TLS status (two-tailed Mann-Whitney *U*-test).

Comment 17:

Fig. 2j. It will be better to provide an explanation to help understand this figure.

Response 17:

We sincerely thank the reviewer for their valuable suggestion. As the reviewer suggested, we added an explanation of Fig. 2j to help more understand this figure in the revised manuscript, with details as following.

Revision made:

In the revised manuscript, the related description as following was added in the “Result Section”.

“We also used a heatmap to show the sharing of clonotypes among clusters of CD8⁺ T cells (Fig. 2j) and noted a significant sharing of clonotypes among neighboring functional clusters of CD8⁺ T cells (clusters 1–4), consistent with the trajectory established by PAGA analysis (Fig. 2d).”

Comment 18:

The mTLS cells have both increased expanded clonotypes and shared clonotypes of CD8⁺ T cells. Please comment on whether these two results are contradictory.

Response 18:

Thank you for bringing up this important discussion point. In our work, we indeed observed that stem-like CD8⁺ T cells (cluster 1-2) showed enrichment for larger clonotype sizes in mTLS. We also noted a significant sharing of clonotypes among neighboring functional clusters of CD8⁺ T cells (clusters 1-4). Actually, these findings are not contradictory. Instead, we believe the simultaneous presence of expanded and shared clonotypes among CD8⁺ T cells in mTLS demonstrates the capacity of mTLS to generate functional CD8⁺ T cells in situ.

Firstly, the presence of expanded clonotypes indicates selective proliferation of certain T cell clones within the tumor. This suggests that these T cells are likely responding to specific tumor-associated antigens, driving their clonal expansion. The expansion of stem-like CD8⁺ T cell clonotypes in mTLS suggests that mTLS may provide an advantageous environment for the activation and proliferation of stem-like CD8⁺ T cells.

Secondly, the existence of shared clonotypes among different CD8⁺ T cells indicate a common recognition of specific tumor-associated antigens. The aggregation of shared clonotype CD8⁺ T cells in mTLS suggests that mTLS provides an environment conducive to the activation and differentiation of CD8⁺ T cells. This also implies that mTLS can offer a robust antigen presentation and immune cell aggregation environment.

It is worth noting that the simultaneous significant increase of expanded and shared clonotypes of CD8⁺ T cells in mTLS is not a contradictory phenomenon. In our study, we found that stem-like CD8⁺ T cells (clusters 1-2) in mTLS exhibit more clonotype expansion, consistent with the result of enrichment analysis. This suggests that mTLS provides a more favorable local environment for the activation and proliferation of these stem-like CD8⁺ T cells. Additionally, we observe significant clonotype sharing among adjacent functional CD8⁺ T cells (clusters 1-4), which aligns with trajectory analysis results. This indicates a stronger driving force for the differentiation of CD8⁺ T cells from stem-like to functional in mTLS. These results collectively support the conclusion that mTLS enables the in situ generation of functional CD8⁺ T cells. Furthermore, Kusnadi et al.²⁹ also observed a simultaneous enhancement of TCR clonotype expansion and sharing in their study on exhausted SARS-CoV-2-reactive CD8⁺ T cells, similar to our findings, demonstrating that this is not a rare phenomenon.

Comment 19:

Fig.3a. Technically, when reclustering CD4⁺ cells, the resulting clusters will be different because of the absence of other cells that contribute to the clustering algorithm. How are the authors able to identify the exact same clusters as Fig. 2b? It will be better to provide more information about this.

Response 19:

Thank you to the reviewer for their insightful comments. In Figure 3a, we used the annotations from Figure 2a specifically for the CD4⁺ T cells. Just to clarify, we extracted these annotations from the Leiden unsupervised clustering in Figure 2a and recalculated the UMAP to improve the visualization of the overall structural relationships among these CD4⁺ T cells. Therefore, the annotations in Figure 2b and Figure 3a are the same. This method of annotation is also commonly used in other previous studies¹³.

Comment 20:

Fig. 3g. Possible labeling error. Should CD8⁺ from title “TCR clonotype sharing across CD8⁺ T cell subclusters” be CD4⁺?

Response 20:

We appreciate the reviewer’s careful observation. According to the reviewer advice, we have corrected the label of Fig. 3g.

Revision made:

In the revised manuscript, the “Fig. 3g” was replaced by the “Fig. R21” as following.

Fig. R21. (Fig. 3g) Heatmap and pie chart showing the number of clonotype sharing between neighboring functional clusters of CD4⁺ T cells in different status of TLS. For heatmap, the color representing the number of shared clonotypes. For pie chart, the color representing the status of TLS and the size representing the number of shared clonotypes.

Comment 21:

Fig. 3. Authors found that mTLS contain more exhausted CD4⁺ T cells, especially progressive Texh.

Response 21:

Thank you for pointing this out. In this research, we observed the differentiation of exhausted CD4⁺ T cells into two distinct subclusters, namely progenitor exhausted (Tex^{prog}) and terminally exhausted (Tex^{term}) type (Fig. 2a, b), expressing relatively high levels of cytotoxicity markers (*IFNG*, *GZLY*, *GZMB* and *GZMH*) (Fig. 3c, d). We found that CD4⁺ Tex^{prog} cells, with features resembling follicular helper T cells (*CXCL13*, *PD-1*, *IL6ST* and *TCF1*), evoke intra-tumoral T and B cell responses in mTLSs. On one hand, CD4⁺ Tex^{prog} cells, mregDCs, and effector memory CD8⁺ T cells form a cellular triad that is likely to facilitate the local differentiation of CD8⁺ Tex^{prog} cells into effector CD8⁺ T cells within mTLSs (Fig. 7n). On the other hand, CD4⁺ Tex^{prog} has the ability to attract and activate GC B cells through CD40LG–CD40 and CXCL13–CXCR5 (Fig. 6g).

Apart from our data, a recent study reported that PD1⁺TCF1⁺ stem-like CD4⁺ T cells can promote effector CD8 T cell responses via IFN γ production²⁸. These T cells share similar molecular markers with the exhausted CD4⁺ T cells defined in our manuscript. These endogenous stem-like CD4⁺ T cells can play a crucial role in the anti-tumor immune response of CD8⁺ T cells, via IFN γ production induced robust effector differentiation from TCF1⁺CD8⁺ T cells. These PD1⁺TCF1⁺ stem-like CD4⁺ T cells improve the effectiveness of immunotherapy even in the presence of Tregs, which usually suppress immune responses.

In summary, CD4⁺ exhausted T cells indicate favorable anti-cancer immunity, and thus we believe the fact that mTLSs contain more CD4⁺ exhausted T cells is reasonable.

Revision made:

In the revised manuscript, we added the explanation about the enrichment of CD4⁺ Tex^{prog}/Tfh cells and CD4⁺ Tex^{term} cells in mTLS.

“Moreover, recent studies have also shown that CD4⁺ T cells have the ability to promote stem-like TCF1⁺CD8⁺ T cell responses through IFN- γ production²⁹. This evidence is further supported by a high level of overlap in clonotype sharing between CD4⁺ Tex^{term} and CD4⁺ Tex^{prog}/Tfh (Fig. 3g), providing further support for the role of CD4⁺ T cells in promoting CD8⁺ T cell immune responses. mIHC stained images also confirmed that CD4⁺ Tex^{prog}/Tfh (CD4⁺TCF1⁺PD-1⁺) enriched in mTLS, particularly in the GC of the mTLS (Fig. 3f).”

.....

“29. Cardenas, M. A., et al. Differentiation fate of a stem-like CD4 T cell controls immunity to cancer. Nature 636, 224-232 (2024).”

Comment 22:

Fig.4c. The description for Fig. 4c, 4e should move up based on alphabetical order.

Response 22:

We would like to thank the reviewer for the constructive comments. According to the reviewer’s suggestion, we have revised the order of the second and third paragraphs of “Promotion of B cell maturation and plasma cell generation by mTLS” section.

Revision made:

In the revised manuscript, we changed the order of the second and third paragraphs as following.

“We next explored the differences in B cell subclusters among various TLS statuses. Although mTLS and imTLS enrich B cells as expected (Fig. 4b, c), the subclusters of these B cells were different..... In line with this, gene enrichment analysis showed that the mTLS-enriched GC B cells exhibit immunoglobulin (Ig) production and antigen presentation through major histocompatibility complex class II (MHC-II) (Fig. 4e).

Subsequently, we conducted trajectory inference to better understand the differentiation trajectories of B cells and plasma cells. Our results showed a pseudotemporal ordering of naive B cells, memory B cells, GC B cells, plasmablasts and plasma cells (Fig. 4f), which is consistent with previous reports on the generation of B cell immunity within lymphoid follicles..... These findings are consistent with previously reported roles of these genes in B cell activation.”

Comment 23:

Fig. 5a. It will be better to provide the information on which markers are used to define myeloid cells.

Response 23:

Thanks for your kind suggestion. We apologize for not clearly presenting the data on Fig. 5a. As suggested, we have shown the markers for myeloid cells in Extended Data Fig. 15.

Revision made:

In the revised manuscript, the “Fig. R22” was added by “Extended Data Fig. 15” as following.

Fig. R22. (Extended Data Fig. 15) Marker gene expression of myeloid cell phenotypes. **a**, Dot plot of scaled marker gene expression (averaged per cluster) for coarse-grained myeloid cell subclusters, showing differentially expressed genes in columns and clusters in rows. Genes are grouped by cluster and five genes per cluster that are highlighted. A color gradient represents the mean expression within each of the subcluster of myeloid cells, while the sizes of the depicted point indicate the fraction of cells in the categories expressing a gene. **b**, Dot plot showing comparison of myeloid cell subclusters (this study) and published myeloid cell subclusters using AUCell algorithm to verify the cell annotation. A color gradient, transitioning from red (representing high scaled mean change) to blue (representing high scaled mean change), encodes the Z-scaled scores, while the sizes of the depicted points are governed by the Benjamini-Hochberg-adjusted $-\log_{10}(P)$ values.

Comment 24:

What are sample types, whole tumor tissue or CD45⁺ cells, used for this analysis?

Response 24:

We thank the reviewer for the insightful comments. In the scRNA-seq analysis, we used the whole tumor tissue for scRNA-seq and CD45⁺ cells in tumor for scRNA-seq and scTCR/BCR-seq. The information about the sample types used in this study is shown in Fig. 1c (Fig. R23) and Extended Data Fig. 1 (Fig. R24).

Fig. R23. (Fig. 1c) UMAP plot colored by patients (n=14) from whole tumor cells or sorted CD45⁺ cells profiled by scRNA-seq.

Fig. R24. (Extended Data Fig. 1) Clinicopathological parameter analysis of HNSCC patients. Clinical metadata and analyses performed for each patient with different status of TLS. Patient data include TLS status, patient age, TNM classification, pathological state. The data collection methods included scRNA-seq for whole tumor cells, paired scRNA-seq and scTCR/BCR-seq for intra-tumoral CD45⁺ cells, spatial transcriptomics, H&E and IHC.

Comment 25:

Fig. 6. The authors mentioned Fig.1i. It seems that Fig.1i is missing.

Response 25:

Thank you for bringing this serious mistake to our attention. We apologize for the missing label on Fig. 1i in Fig. 1. We have corrected this labeling error, and the correct label for Fig. 1 is displayed below. (Fig. R25 shown below)

Revision made:

In the revised manuscript, we have corrected label for the “Fig. 1i”.

Fig. R25. (Fig. 1h, i) g-i, The cell types for scRNA-seq of whole tumor cells and the composition difference of CD45⁺ cells (g), H&E and IHC of tumor slides (h), and spatial transcriptomics of tumor tissue (i) in nTLS, imTLS, and mTLS statuses. A color gradient, transitioning from red (indicating enrichment) to blue (signifying depletion), encodes the log₂-transformed odds ratios, while the sizes of the depicted points are governed by the Bonferroni-adjusted -log₁₀(P values), accentuating the statistical significance of observed variations.

Comment 26:

Fig. 6c. The image that staining of B.naive, CD4.T.naive, CD8.cytotoxic and cDC2 was abundant and saturated in the same follicle raises concern that the staining is not specific. Please comment on this.

Response 26:

Thank you for your insightful comment. We apologize for the unclear representation of the results in Fig. 6.

Actually, Fig. 6a-c are not staining images. Our study utilized Visium v1 technology, which provides a relatively low resolution of 55 μm in diameter and encompassing 5 to 30 cells per spot. As a result, each spot contains gene information from multiple cells rather than individual cell-level data. Therefore, in Fig. 6c, B.naive, CD4.T.naive, CD8.cytotoxic, and cDC2 cells appear to be in the same area of spots. Actually, other studies employing Visium v1 technology have likewise shown comparable results³⁰ that there are different types of cells in the same area of spots. This observation actually supports our research that these cells cluster aggregate in mTLS, providing a good structural basis for intercellular communication and interaction. The co-localization of these cell types within the same spots underscores the potential for complex cellular interactions and signaling within the mTLS, which is crucial for their functional significance in the tumor immune response.

We have further emphasized this technical limitation in the Discussion section to guide readers towards a more accurate and objective interpretation of our research findings.

Revision made:

In the revised manuscript, we added relevant explanations in Discussion.

“Thirdly, the localization of different immune cell subtypes was determined using Visium v1 technology with relatively low resolution (55 μm in diameter) and may encompassing 5 to 30 cells per spot; thus, we employed bioinformatic tools to deconvolute the spatial transcriptome without directly observing cell-cell interactions.”

Comment 27:

Fig. 7c-d. The patient numbers in each group are almost the same in Fig. 7c and Fig. 7d. Please double check and make sure those numbers are correct.

Response 27:

Thank you for your comment. In Fig. 7c-d, we explored the association between the expression levels of GC B cells and CD8⁺ effector memory T cells and the overall survival rates of HNSCC patients. In the survival analysis, we utilized the median as the cut-off point to distinguish between low and high expression of cell cluster. Using the median as a cutoff to distinguish between low and high expression groups in survival analysis is indeed a classical approach^{31, 32, 33}. This method is commonly used because it divides the sample into two equal-sized groups, ensuring a balanced comparison. Therefore, the number of patients in each group is nearly equal.

Comment 28:

Line 561, “For tumor tissue” should be “Tumor tissue”.

Response 28:

Thank you for pointing out this issue. We apologize for the incorrect description. We have changed the word of “For tumor tissue” to the “Tumor tissue”.

Revision made:

In the revised manuscript, we changed the word of “For tumor tissue” to the “Tumor tissue”.

“Tumor tissue was cut into approximately <2 mm³ in tissue storage solution and then enzymatically digested using RPMI-1640 medium modified with collagenase IV (Biosharp), DNase I (BioFroxx) and hyaluronidase (BioFroxx) in a C-tube using the gentleMACS system (Miltenyi) at 37 °C for 1h.”

Comment 29:

Line 148. What is “almost phenotypes”?

Response 29:

Thanks for the reviewer’s careful observation and we apologize for the incorrect description in our manuscript. The word of “almost phenotypes” has been replaced by “almost all phenotypes”.

Revision made:

In the revised manuscript, the word of “almost phenotypes” has been replaced by “almost all phenotypes”.

“Among the CD4⁺ T subclusters, almost all phenotypes (clusters 9–13) were found to be more enriched in mTLS, moderately enriched in imTLS, and less abundant in nTLS, except for the interferon-stimulated gene (ISG) T cell phenotype (Fig. 2c).”

Comment 30:

Line 334. Incomplete sentence: facilitated the identification of a cell cluster associated with TLS, which included B cells, T cells, DCs and myeloid cells.

Response 30:

We appreciate the reviewer's comment. We apologize for this incomplete statement in the manuscript. In the revised manuscript, we have altered our description to clarify the meaning.

Revision made:

In the revised manuscript, the description treatment was changed as following.

“This method facilitated the identification of a cell cluster associated with TLS, which included B cells, T cells, DCs and myeloid cells (Fig. 6a-c).”

Reviewer #3 (Remarks to the Author):

Summary Comment:

The manuscript addresses the fine description of the tumor microenvironment (TME) of head and neck squamous cell carcinoma (HNSCC) tumors, using first a combination of single-cell multi-omics and low resolution spatial transcriptomics on a small series of samples, and second multiplex IHC and bioinformatics deconvolution of bulk RNA-seq on larger series of patient samples (either from this study or publicly available).

The authors focus mainly on 14 samples which they have categorized based on their content of tertiary lymphoid structures (TLS) and their degree of maturity. They study 4 samples visibly devoid of TLS (nTLS = non-TLS), 4 samples containing only immature TLS (imTLS), and 6 samples containing mature TLS (mTLS). The authors split the freshly resected tumor specimens into several parts which they processed separately to yield (1) scRNA-seq data on all viable cells from the whole dissociated tumors, (2) multi-omics (scRNA-seq + scBCR-seq + scTCR-seq) on CD45⁺ cells, and (3) spatial transcriptomics (10x Genomics Visium, 55µm-diameter capture spots, whole transcriptome). The authors use state-of-the-art bioinformatics analyses to qualify and quantify cell types and cell states, first in a broad manner (Figure 1), then focusing specifically on T and NK cells (Figures 2 and 3), B cells (Figure 4), Myeloid cells (Figure 5). They map the cell types and states they have characterized in details in the first figures to their location in spatial transcriptomics dataset (Figure 6) through deconvolution, co-occurrence analysis, and ligand-receptor analyses. Finally, they extend their observations to other cohorts of samples analyzed by distinct methods (Figure 7), confirming their results on the small series of 18 samples, and extending their observations to define biological correlates of survival for patients with HNSCC, also when treated with immune checkpoint blockade therapy.

Assessment:

The manuscript is clearly written, although it would benefit from a thorough editing by a native English speaker. Overall, the authors have presented a very complete dataset, with high quality scRNA-seq and multi-omics data on a carefully selected series of HNSCC samples reflecting distinct TLS infiltration patterns. Given the association of TLS with clinical course in many solid tumor types, including HNSCC, this descriptive study is of major interest to researchers and clinicians interested in understanding and harnessing the TME for obtaining durable response in HNSCC. The bioinformatic methods are clearly explained, and the analysis tools that the authors have used have been carefully selected to fit the purposes of their analyses.

Nevertheless, there are some major points that I believe should be addressed to support the main claims of the article on functional immune mechanisms occurring in mature TLS, and strengthen the study.

Response:

We are very grateful to the reviewer for their positive comments. We also highly appreciate the reviewer's careful reading and constructive suggestions that helped us to further the quality. By responding to the reviewer's comments in detail and revising the manuscript accordingly, we believe our manuscript has been significantly strengthened. All revisions are highlighted in **BLUE** color in the revised manuscript and Supplementary Information.

Comment 1:

First, the authors should relate to previous single-cell studies of HNSCC, including the work of Ruffin et al. (Nature Communications 2021) which suggested that TLS infiltration was tightly link to HPV status. This work is not referenced, and the HPV status of samples in the study is not described.

Response 1:

Thank you so much for your helpful feedback. As you mentioned, TLS infiltration is indeed closely linked to HPV status. In this study, all 14 patients included were confirmed to be HPV negative through both P16 protein staining and in situ hybridization dual testing. Furthermore, we have included relevant information about the patients' HPV status in the methods section.

Revision made:

In the revised manuscript, the description treatment was changed as following.

“All 14 patients included were confirmed to be human papillomavirus (HPV) negative through both P16 protein staining and in situ hybridization dual testing.”

Comment 2:

Second, it is unclear to me whether the clonality assessment for B cells included a “tolerance” for intra-clonal somatic hypermutation. The methods seem to describe that CDR3 identity was used for grouping B cells in clonotypes. If this is indeed the case, I suggest using a more “relaxed” definition of clonotypes that would group cells even if they have diverged from their naïve B cell ancestor through somatic hypermutation. There are many ways to do so, and I could recommend the Immcantation suite in R (<https://immcantation.readthedocs.io/en/stable/>). After revising the B cell clonotype definition, if necessary, it would be interesting to comment on the fact that GC B cells in mTLS do not seem to be clonally linked to plasmablasts and plasma cells, whereas Memory B cells have a slightly higher clonotypic link with antibody-secreting cells. That would suggest that GC B cells in mTLS are not the main or the direct source of plasmablasts and plasma cells, contrary to what is depicted in the summary Figure 7r.

Response 2:

Thank you for your valuable suggestion. We greatly appreciate your recommendation of the Immcantation suite in R for providing a “relaxed” definition of clonotypes. This suggestion is crucial for improving the quality and professionalism of our work.

Following your advice, we apply a more relaxed definition to our B cell clonotypes. In the Immcantation Portal, we using the hierarchical clustering method for clonal partitioning by SCOPer package. Specifically, we firstly found the optimal threshold for trimming the hierarchical clustering into B cell clones by shazam package. We then employed the hierarchical clustering method to group sequences with similar junction regions, determining shared common B cell ancestors.

However, as you expected, this approach did not reveal a significant increase in shared clonotypes between plasma cells and GC B cells (Fig. R26). This limitation may be due to the algorithm's inability to fully bridge the clonal relationships disrupted by SHM. Interestingly, King et al.³⁴ also observed weak clonotype sharing between B cells and plasmablasts in tonsillar lymphoid follicles using similar methods.

Therefore, we strongly agree with your opinion and have made changes to Fig. 7r and its corresponding statements. We also replaced the previous Fig. 4g with clonotype sharing results based on the immcantation “relaxed” analysis.

Revision made:

In the revised manuscript, the Fig. 4j was changed to “Fig. R26” as following.

Fig. R26. (Fig. 4j) Heatmap chart showing the number of clonotype sharing between neighboring functional subclusters of B cells and plasma cells in different status of TLS. For heatmap, the color gradient from red to blue indicates decreasing clonotype number between B cells and plasma cells.

In the revised manuscript, the Fig. 7r was changed to “Fig. R27” as following.

Fig. R27. (Fig. 7r) Sketch map showing the TME of HNSCC with nTLS, imTLS and mTLS

In the revised manuscript, the methods about definition of clonotypes of B and plasma cells was changed as following.

“Each unique heavy chain (IGH) and light chain (IGK or IGL) pair was detected and a relaxed

definition to B cell and plasma cells clonotypes by Immcantation Portal (version 4.5.0)⁶⁶. Specifically, the optimal threshold for trimming the hierarchical clustering into B cell clones was firstly calculated by shazam package⁶⁶. We then used the hierarchical clustering method⁶⁷ by SCOPer package⁶⁸ to cluster B cell receptor sequences. This clustering was based on the similarity of junction region sequences within partitions that shared the same V gene, J gene, and junction length, while accommodating ambiguous V or J gene annotations.”

.....

“66. Gupta, N. T. et al. Change-O: a toolkit for analyzing large-scale B cell immunoglobulin repertoire sequencing data. *Bioinformatics* 31, 3356-3358 (2015).

67. Gupta, N. T. et al. Hierarchical clustering can identify B cell clones with high confidence in Ig repertoire sequencing data. *J Immunol* 198, 2489-2499 (2017).

68. Nouri, N. & Kleinstei, S. H. A spectral clustering-based method for identifying clones from high-throughput B cell repertoire sequencing data. *Bioinformatics* 34, i341-i349 (2018).”

Comment 3:

Third, the spatial datasets, either from spatial transcriptomics or from multiplex IHC, would benefit from proper quantitative analyses, with statistically robust hypothesis testing taking into consideration the different samples analyzed. For example, Figure 6 a-c only show examples and are barely readable, while the correlation matrices in Figure 6d-f seem to be built based on all ST datasets from each sample category, and do not offer any statistical analysis. In Figure 7n-o, it is unclear to me what the main result is regarding cell type proximity, and how such result is robustly backed up by the data presented. I believe the authors should extract the relevant metrics (e.g. proximity of CD4⁺TCF1⁺CXCL13⁺ cells to B cells, DC-LAMP⁺ cDCs, or CD8⁺ T cells) from each slide individually, and analyze the distributions of those metrics in all sample groups with robust statistical analysis.

Response 3:

We sincerely thank the reviewers for their valuable suggestions and comments. Indeed, proper quantification analysis will help readers and reviewers to better understand the result. According to the reviewer's comments, we provided these statistics analysis for each sample. (Fig. R28-30).

Firstly, we present the “Fig. R28” of the average enrichment score of spots in relative area, which is the quantification of “6a-c”. The results showed a high level of enrichment in GC B cells, plasma cells, exhausted CD4⁺ T cells, and CD8⁺ T cells, as well as DCs that were enriched in mTLS.

Additionally, we present the “Fig. R29” of the co-occurrence score between cell-cell, which is the quantification of “6d-f”. From the quantitative analysis, we found that the co-occurrence scores between CD4⁺ Tex^{prog} cells and DC cells, B cells, or CD8⁺ T cells were higher in mTLS. This result indicates that these four cell subtypes are more aggregated in spatial positions and exhibit closer relationships within mTLS.

Moreover, we also quantified the mIHC images of “Fig. R30”, which is the quantification of “7n-o”. The quantification results showed that the distance of cell-cell interaction between CD4⁺TCF1⁺CXCL13⁺ cells, B cells, DC-LAMP⁺ DC and CD8⁺TCF1⁺T cells were closer. And the proportion of cell-cell interaction was higher in mTLS.

Revision made:

In the revised manuscript, we added the “Fig. R28”, replaced by “Extended Data Fig. 16” as following.

Fig. R28. (Extended Data Fig. 16) Average enrichment score by spatial transcriptomics in HNSCC. a-j. Average enrichment score of B.GC (a), Plasma.cell.1 (b), CD4.T.prog.exh (c), CD4.T.term.exh (d), CD8.T.effector.mem (e), CD8.T.cytotoxic (f), CD8.T.exh (g), cDC1 (h), cDC2 (i), mregDC (j) spots in spots of relative area (nTLS, imTLS or mTLS) across different TLS status detected by spatial transcriptomics (n = 3 nTLS, n = 3 imTLS, n = 4 mTLS, one-tailed Mann-Whitney U-test).

In the revised manuscript, we added the “Fig. R29”, replaced by “Extended Data Fig. 17” as following.

Fig. R29. (Extended Data Fig. 17a-e) Statistics on cell–cell communication and receptor–ligand pairs in HNSCC. a-e, Relative co-occurrence score of CD4⁺ Tex^{prog} cells and GC B cells (a), CD4⁺ Tex^{prog} cells and effector memory CD8⁺ T cells (b), CD4⁺ Tex^{prog} cells and cDC2s (c), CD4⁺ Tex^{prog} cells and mregDC (d), effector memory CD8⁺ T cells and mregDCs (e) across different TLS status detected by spatial transcriptomics (n = 3 nTLS, n = 3 imTLS, n = 4 mTLS, one-tailed Mann-Whitney *U*-test).

In the revised manuscript, we added the “Fig. R30”, replaced by “Extended Data Fig. 20”as following.

Fig. R30. (Extended Data Fig. 20) Statistic of the ratio of cell-cell interactions and the average distance of cell-cell interactions. **a-f**, Ratio of cell-cell interactions within overall cellular interactions and average distance of cell-cell interactions between $CD4^+TCF1^+CXCL13^+$ cells and $CD8^+TCF1^+$ cells (**a, b**), $CD20^+$ cells (**c, d**), and $LAMP^+$ DCs (**e, f**) within overall cellular interactions in HNSCC tissue with different TLS status, respectively. (two-tailed Mann-Whitney U -test; for box plots: box center line, median; box limits, upper and lower quartiles; box whiskers, maximum and minimum values).

Comment 4:

Fourth, it is unclear to me how the $CD4^+$ Tex^{prog} subset differs from bona fide TFH cells. Most studies on TLS thus far have identified $CD4^+$ T cells within mTLS as TFH cells, expressing PD1 and CXCL13 like TFH in secondary lymphoid organs. Here the authors do not identify any TFH cells in their samples, even though the $CD4^+$ Tex^{prog} subset would classify as such in my opinion. Is it just a question of nomenclature?

Response 4:

Thank you for the reviewer's insightful comment on the distinction between CD4⁺ Tex^{prog} cells and Tfh cells. We agree with your perspective that the difference between our CD4⁺ Tex^{prog} cells and Tfh cells in other scRNA-seq studies is mainly in nomenclature. As your comment, most studies about TLS identify CD4⁺ T cells that express PD-1, TCF1 and CXCL13 as Tfh cells. In our study, we identified this type of CD4⁺ T cell as progenitor exhausted CD4⁺ T cells based on this cell is with features of progenitor exhausted (expressing PD-1, CXCL13, TCF1) and can further differentiate into terminal exhausted CD4⁺ T cells (through our analysis of scTCR-seq). However, we hold the view that they may be two distinct cell types. Due to technical limitations of scRNA-seq, there is difficulty in fully distinguishing Tfh cells from CD4⁺ Tex^{prog} cells at the scRNA-seq level. The following are the differences between CD4⁺ Tex^{prog} cells and Tfh cells.

Firstly, classic Tfh cells are found in primary or secondary lymphoid organs³⁵. They regulate B cell immune responses in germinal centers of lymphoid follicles³⁵. Secondly, in our research and previous other studies¹³, trajectory analysis and scTCR-seq analysis showed that CD4⁺ Tex^{prog} cells can differentiate from intratumoral CD4⁺ T cells and can further differentiate into terminally exhausted CD4⁺ T cells.

We have noticed some scRNA-seq studies use vague definitions for these Tfh-like CD4⁺ Tex^{prog} cells. Vázquez-García et al. defined these CD4⁺ T cells as “CD4.T.dysfunc.early”¹³. Liu et al. defined these CD4⁺ T cells as “Tfh-like T cells”³⁶. Magen et al. named these CD4⁺ T cells as CXCL13⁺ TH1/TFH CD4⁺ T cells³⁷. Oliveira et al. defined these CD4⁺ T cells as TFH/TPE³⁸.

Based on your suggestion and recent scRNA-seq studies about CD4⁺ T cells in tumor^{36, 39}, we have decided to rename “CD4⁺ Tex^{prog} cells” to the more precise “CD4⁺ Tex^{prog}/Tfh cells” in our manuscript. This change will avoid potential confusion for readers due to nomenclature issues.

Revision made:

In the revised manuscript, the description of CD4⁺ Tex^{prog} cells was changed as following.

We have revised the manuscript and rename all “CD4⁺ Tex^{prog} cells” to “CD4⁺ Tex^{prog}/Tfh cells” in all the TEXT and FIGURES.

Comment 5:

Fifth, any claim on functionality, e.g. crosstalk between cDC2s, CD4⁺ Tex^{prog} and B cells in mTLS, should be backed up by functional assays, not only bioinformatics analyses of scRNA-seq and spatial transcriptomics data. In the absence of functional assays, any conclusion should remain speculative and be stated as such.

Response 5:

We thank the reviewer for their insightful comment. We are apologized that using inappropriate descriptions about “*These CD4⁺ T cells interactions with cDC2 cells to coordinate the activation and subsequent differentiation of GC B cells into antibody-producing plasma cells*”. As your comment, the conclusions described in the article are too absolute without function assays. We have modified the relevant statements to indicate that this conclusion is based on our speculation.

In order to strengthen the data support, we performed mIHC staining and statistical analysis to speculate the interaction between cDC2s (CD1c⁺), CD4⁺ Tex^{prog} (CD4⁺TCF1⁺PD-1⁺) and B cells (CD20⁺BCL6⁺). Our analysis uncovered that cDC2, characterized by CD1c⁺ expression^{40, 41}, is enriched

in mTLS and resided in close proximity to B cells and CD4⁺ Tex^{prog}. The relationship between cDC2, B cells, and CD4⁺ Tex^{prog} cells showed aggregated structures. Meanwhile, cDC2, CD4⁺ Tex^{prog} and B cells had the highest frequency and the shortest nearest neighbor distance among mTLS.

Revision made:

In the revised manuscript, we added the “Fig. R31” replaced by “Extended Data Fig. 21” as following.

Fig. R31. (Extended Data Fig. 21) Representative images and quantification of a mIHC-stained different subclusters of BCL6⁺CD20⁺ B cells, CD4⁺TCF1⁺PD-1⁺ Tex^{prog} cells in HNSCC tumor with different TLS status. a-c Images of the mIHC-stained HNSCC tissues with different TLS status showing co-localization of B cell (CD20, BCL6), CD4⁺ Tex^{prog} (CD4, PD-1 and TCF1), and cDC2s (CD1c) in HNSCC across. Scale bars = 100 μm. d, e, Ratio of CD4⁺TCF1⁺PD-1⁺ cells (d) and CD20⁺BCL6⁺ cells (e) in total cells of HNSCC tissue

with different TLS status, respectively. f, g, Ratio of CD4⁺ TCF1⁺PD-1⁺ cells and CD1c⁺ cells (f), CD20⁺BCL6⁺ cells and CD1c⁺ cells (g) interactions within overall cellular interactions, respectively. (two-tailed Mann-Whitney U-test).

Comment 6:

I will further detail my concerns and suggestions below.

Lines 188-189: Proper quantification of multiplex IHC images is required to support the claims.

Response 6:

Thanks for the reviewer's constructive suggestion. According to your suggestion, we added the quantification of mIHC images about Fig. 2f and Extended Data Fig. 9.

Revision made:

In the revised manuscript, we added the "Fig. R32" to "Extended Data Fig. 10", related with the quantification of Fig. 2f and Extended Data Fig. 9.

Fig. R32. (Extended Data Fig. 10) a-b, Proportion of naive/central memory CD8⁺CCR7⁺ cells (a) and TCF1⁺ cells (b) in total CD8⁺T cells across different TLS status detected by mIHC (n = 4 nTLS, n = 4 imTLS, n = 6 mTLS, one-tailed Mann-Whitney U-test).

Comment 7:

Line 213: Fig. 3f is an IF image. Please correct Figure 3 citations for proper panels throughout the text.

Response 7:

We thank the reviewer for this helpful comment. According to the reviewer's suggestion, we corrected the Figure 3f citations in the revised manuscript.

Revision made:

In the revised manuscript, we corrected the citations of "Fig. 3f" to "Fig. 3e" in the Result Section as following.

"The phenotypes of Tex and Treg occupied a larger clonal expansion (Fig. 3e)"

Comment 8:

Line 276: the fact that PCs are found in nTLS samples may indicate that PCs are generated outside TLS, but not necessarily outside the TME. In secondary lymphoid organs, PBs and PCs may be generated from naïve or memory B cells in GC-independent and extra-follicular responses.

Response 8:

Thank you very much for your careful review and valuable comments in our article. In our study, nTLS showed a low number of B cells but a high number of plasma cells. According to the reviewer's comments, we also further search the literature reports. Indeed, this description in the original article may be oversimplified and does not fully account for the complexity and diversity of plasma cytogenesis.

In tumors enriched with TLSs, there are evidence that TLSs generate and propagate anti-tumor antibody-producing plasma cells¹⁰. At the same time, we found that a part of plasmablasts/plasma cells are derived from intratumoral GC B cells (Fig. 4f). Thus, as least a part of plasma cells in nTLS-tumors are produced intratumorally.

However, in tumors without TLS, the origin of plasma cells is a little complicated. On the one hand, there is an extra-tumoral origin. Plasma cells can be produced in secondary lymph organs and be recruited to tumors^{42, 43}. On the other hand, there is also a potential intra-tumoral origin. For example, there are reports of the existence of memory B cells, which can give rise to plasma cells^{44, 45}. It means that plasma cells may also be formed by memory B cell differentiation after reactivation in the nTLS tumor microenvironment. Thus, we admit that the observation of a low number of B cells and a high number of plasma cells in nTLS does not directly infer that plasma cells are generated outside the TME. Basing on current knowledge, we can only conclude that plasma cells in nTLS tumors are generated outside TLSs. According to your suggestions, we have revised the manuscript.

Revision made:

In the revised manuscript, the related description of was modified as following.

*“Interestingly, nTLS showed minimal numbers of B cells, except for an enrichment of plasma cells, suggesting that these plasma cells may have originated from **outside the TLS** (Fig. 4b, c).”*

Comment 9:

4. Line 283: see my general comment above on BCR clonotype definition. Please revise conclusions accordingly if the clonotype definition should be revised.

Response 9:

We would like to thank the reviewer for the valuable suggestions. We have fully incorporated your recommendation in Comment 2, using the Immcantation Portal to define BCR clonotypes in a “relaxed” way. The specific methodology we used for BCR clonotype analysis with the Immcantation Portal is thoroughly discussed in Comment 2. In brief, we now identify BCR clonotypes based on the similarity of junction region sequences within partitions that share the same V gene, J gene, and junction length, while accounting for ambiguous V or J gene annotations.

However, as you expected, this approach did not reveal a significant increase in shared clonotypes between plasma cells and B.GC cells (Fig. R33). We strongly agree with your opinion and have made changes to Fig. 7r and its corresponding statements. We also replaced the previous Fig. 4j with clonotype sharing results based on the immcantation “relaxed” analysis (Fig. R33).

Revision made:

In the revised manuscript, the Fig. 4j was changed as Fig. R33.

Fig. R33. (Fig. 4j) Heatmap chart showing the number of clonotype sharing between neighboring functional subclusters of B cells and plasma cells in different status of TLS. For heatmap, the color gradient from red to blue indicates decreasing clonotype number between B cells and plasma cells.

In the revised manuscript, the Fig. 7r was changed as Fig. R34.

Fig. R34. (Fig. 7r) Sketch map showing the TME of HNSCC with nTLS, imTLS and mTLS.

In the revised manuscript, the methods about definition of clonotypes of B and plasma cells was changed as following.

“Each unique heavy chain (*IGH*) and light chain (*IGK* or *IGL*) pair was detected and a relaxed definition to B cell and plasma cells clonotypes by Immcantation Portal (version 4.5.0)⁶⁴. Specifically, the optimal threshold for trimming the hierarchical clustering into B cell clones was firstly calculated by shazam package⁶⁴. We then used the hierarchical clustering method⁶⁵ by SCOPer package⁶⁶ package to cluster B cell receptor sequences. This clustering was based on the similarity of junction region

sequences within partitions that shared the same V gene, J gene, and junction length, while accommodating ambiguous V or J gene annotations.”

.....

“64 Gupta, N. T. et al. Change-O: a toolkit for analyzing large-scale B cell immunoglobulin repertoire sequencing data. *Bioinformatics* 31, 3356-3358 (2015).

65 Gupta, N. T. et al. Hierarchical Clustering Can Identify B Cell Clones with High Confidence in Ig Repertoire Sequencing Data. *J Immunol* 198, 2489-2499 (2017).

66 Nouri, N. & Kleinstein, S. H. A spectral clustering-based method for identifying clones from high-throughput B cell repertoire sequencing data. *Bioinformatics* 34, i341-i349 (2018).”

Comment 10:

Lines 289-290: see my comments above, the results indeed suggest that PCs originate not from GC B cells but from memory B cells. The diagram in Figure 7r should be revised accordingly.

Response 10:

We would like to thank the reviewer for the valuable suggestions. We fully agree with your comment regarding the description of plasma cells originate only from GC B cells in Fig. 7r. Based on your comments in Comment 2 and 9, we have defined of BCR clonotypes in a “relaxed” way. However, as your expected, this approach did not reveal a significant increase in shared clonotypes between plasma cells and GC B cells (Fig. R35). In light of the latest scBCR-seq analysis, we have revised Fig. 7r and its corresponding statements.

Revision made:

In the revised manuscript, we revised the “Fig. 7r” replaced by the “Fig. R35”.

Fig. R35. (Fig. 7r) Sketch map showing the TME of HNSCC with nTLS, imTLS and mTLS

Comment 11:

Lines 323-326: it is unclear to me what is the relevance of cDC2 cells expressing genes involved in “Ig production”, since cDC2s are not producing Ig themselves. Could the authors precise what those

genes are? How do those analyses suggest that cDC2s play a crucial role in regulating antibody responses?

Response 11:

Thank you for your insightful comment. We appreciate the opportunity to clarify our findings regarding cDC2 cells and their role in Ig production. In fact, the term “Ig production” (GO:0002381) refers to the regulation of immunoglobulin production, not the direct production of Ig itself.

We calculated functional enrichment scores for cDC2 using manually curated lists or GO Biological Process (GO BP) terms by decoupleR package. The “Ig production” is a GO BP term⁴⁶ (GO:0002381, PMID: 9185563) includes genes such as *AICDA*, *APLF*, *ATAD5*, *BATF*, *BCL6*, *BTK*, *CD28*, *CD40*, *IL10*, *IL2*, *IL4*, *STAT6*, *TGFB1*, and *TNFSF13*.

It is worth noting that this “Ig production” term states: “Note that this term is in the subset of terms that should not be used for direct gene product annotation. Instead, select one of the 'regulation' children's terms” (<https://amigo.geneontology.org/amigo/term/GO:0002381>). It means that this annotation further supports that the term “Ig production” specifically refers to the regulation of immunoglobulin production, rather than direct Ig synthesis.

Therefore, the enrichment of the “Ig production” term in cDC2 cells suggests their role in regulating B cell-mediated Ig production, rather than directly producing Ig themselves. cDC2 cells express these genes to modulate and support B cell-mediated Ig production.

Comment 12:

Lines 337-353: see my comment above. Those conclusions should be backed up by proper image quantification and statistically robust analyses.

Response 12:

We sincerely thank the reviewers for their valuable comments. According to your suggestion, we added the image quantification and statistical analysis of Fig. 6a-c. Details are in the “Results section” and highlight in **BLUE** in the revised manuscript as following.

Revision made:

In the revised manuscript, we added the quantification of “Fig. R36.” as following.

Fig. R36. (Extended Data Fig. 16) a-j, Average enrichment score of B.GC (a), plasma.cell.1 (b), CD4.T.prog.exh (c), CD4.T.term.exh (d), CD8.T.effector.mem (e), CD8.T.cytotoxic (f), CD8.T.exh (g), cDC1 (h), cDC2 (i), mregDC (j) spots in relative area across different TLS status detected by spatial transcriptomics (n = 3 nTLS, n = 3 imTLS, n = 4 mTLS, one-tailed Mann-Whitney U-test).

Comment 13:

Lines 362-364: please explain how CD4⁺ Tex^{prog} differ from TFH.

Response 13:

Thank you for bringing up this important question regarding the distinction between CD4⁺ Tex^{prog} cells and Tfh cells. Actually, the difference between our CD4⁺ Tex^{prog} cells and Tfh cells in other scRNA-seq studies is mainly in nomenclature. Most studies about TLS identify CD4⁺ T cells in that express PD-1, TCF1 and CXCL13 as Tfh cells. In our study, we identified this type of CD4⁺ T cell as progenitor exhausted CD4⁺ T cells based on this cell is with features of progenitor exhausted (expressing PD-1, CXCL13, TCF1) and can further differentiate into terminal exhausted CD4⁺ T cells (through our analysis of scTCR-seq). However, we hold the view that they may be two distinct cell types. Due to technical limitations of scRNA-seq, there is difficulty in fully distinguishing Tfh cells from CD4⁺ Tex^{prog} cells at the scRNA-seq level. The following are the differences between CD4⁺ Tex^{prog} cells and Tfh cells.

Firstly, classic Tfh cells are found in primary or secondary lymphoid organs³⁵. They regulate B cell immune responses in germinal centers of lymphoid follicles³⁵. Secondly, in our research, CD4⁺ Tex^{prog} cells in mTLS not only exhibit Tfh-like functions in regulating B and T cell responses but also show significant progenitor exhaustion features. Furthermore, trajectory analysis and scTCR-seq analysis confirm that CD4⁺ Tex^{prog} cells can differentiate from effector CD4⁺ T cells and can further differentiate into terminally exhausted CD4⁺ T cells.

We have noticed some scRNA-seq studies use vague definitions for these Tfh-like CD4⁺ Tex^{prog} cells. Vázquez-García et al. defined these CD4⁺ T cells as “CD4.T.dysfunc.early”¹³. Liu et al. defined these CD4⁺ T cells as “Tfh-like T cells”³⁶. Magen et al. named these CD4⁺ T cells as “CXCL13⁺ TH1/TFH CD4⁺ T cells”³⁷. Oliveira et al. defined these CD4⁺ T cells as “TFH/TPE”³⁸.

Based on your suggestion and recent scRNA-seq studies about CD4⁺ T cells in tumor^{36, 39}, we have decided to rename “CD4⁺ Tex^{prog} cells” to the more precise “CD4⁺ Tex^{prog}/Tfh cells” in our manuscript. This change will avoid potential confusion for readers due to nomenclature issues.

Revision made:

In the revised manuscript, the content was changed as following.

We have revised the manuscript and rename all “CD4⁺ Tex^{prog} cells” to “CD4⁺ Tex^{prog}/Tfh cells” in all the TEXT and FIGURES.

Comment 14:

Lines 374-377: those figures are not readable. Please provide a quantification over all analyzed sections.

Response 14:

We thank the reviewer for their insightful comment. According to your suggestion, we added the quantification of Fig. 6i in the revised manuscript (Fig. R37).

Revision made:

In the revised manuscript, we added the quantification of Fig. 6i as following.

Fig. R37. (Extended Data Fig. 17f-h) a-c, Average enrich score of CD70⁺CD27⁺ (a), CXCL13⁺CXCR5⁺ (b), and CD40LG⁺CD40⁺ (c) spots in relative area across different TLS status detected by spatial transcriptomics (n = 3 nTLS, n = 3 imTLS, n = 4 mTLS, one-tailed Mann-Whitney *U*-test).

Comment 15:

Line 425: Figure 7n is an example. Please provide proper quantification and statistically robust analysis.

Response 15:

We thank the reviewer for their insightful comment. We apologized for the unclear description quantitation and analysis of Figure 7n. In our manuscript, Fig. 7n is mIHC images and Fig. 7g-l is the quantification data.

Revision made:

In the revised manuscript, we showed the quantification of Fig. 7n as following.

Fig. R38. (Fig. 7g-l) g-l, Proportion of CD20⁺, CD4⁺, CD8⁺, DC-LAMP⁺, CD8⁺TCF1⁺ and CD4⁺TCF1⁺CXCL13⁺ cells in total cells of HNSCC tissue with different TLS status, respectively (nTLS = 303, imTLS = 83, mTLS = 36, two-tailed Mann-Whitney *U*-test).

In revised manuscript, we present the description of the Fig. 7g-l in the “Results section” as follows.

“In accordance with our earlier analyses, mTLS are associated with high density of CD20⁺ B cells, CD4⁺ T cells, CD8⁺ T cells and DC-LAMP⁺ mregDC (Fig. 7g-j). Notably, this includes a high concentration of CD8⁺TCF1⁺ T cells and CD4⁺TCF1⁺CXCL13⁺ T cells, which are primarily categorized as stem-like CD8⁺ T cells containing naive/central memory and effector memory phenotypes, and as progenitor exhausted phenotypes within CD4⁺ T cells, respectively, according to scRNA-seq data (Fig. 7k, l).”

Comment 16:

Line 453: I believe “elucidated” is too strong a word when no functional assay on immune

mechanism has been used.

Response 16:

We thank the reviewer for their insightful comment. Indeed, the word “elucidated” is too strong in Line 453. Therefore, we changed the word “elucidated” to “explore” in the revised manuscript.

Revision made:

In the revised manuscript, we changed the related description as following.

“Here, we investigated the role of TLS status in HNSCC and explored their respective roles in antitumor immunity.”

Comment 17:

Line 485: “nTLS” missing at the beginning of the sentence.

Response 17:

Thanks for the reviewer’s careful reading. As the reviewer's suggested, we added this description in the manuscript, with details as following.

Revision made:

In the revised manuscript, the “nTLS” was added in the beginning of the sentence in Line 485 as following.

“nTLS showed a predominance of plasma cells without corresponding B cells, suggesting that these plasma cells might be produced at alternative sites, such as tumor-draining lymph nodes (tdLN), as reported previously.”

Comment 18:

Line 500-501: the data presented here seems to show that PCs are not derived from GC B cells, so the link between GCs and PC production should be discussed more critically.

Response 18:

We thank the reviewer for their insightful comment. In our study, we described that “mTLS commonly harbor germinal centers, indicating their capacity to efficiently produce plasma cells capable of generating in situ antibodies within the TME”. According to your comments, we further search the literature reports. And we summarized the origin of PC into several aspects.

In tumors enriched with TLSs, there are evidence that TLSs generate and propagate anti-tumor antibody-producing plasma cells¹⁰. At the same time, we found that a part of plasmablasts/plasma cells are derived from intratumoral GC B cells (Fig. 4f). Thus, as least a part of plasma cells in mTLS-tumors are produced intratumorally.

In tumors without TLS, the origin of plasma cells is a little complicated. Outside tumors, plasma cells can be produced in secondary lymph organs and be recruited to tumors^{42, 43}. In tumors, there are reports of the existence of memory B cells, which can give rise to plasma cells^{44, 45}. It means that plasma cells may also be formed by memory B cell differentiation after reactivation in the nTLS tumor microenvironment. Thus, we added more content to discussed PC production critically.

Comment 19:

Lines 504-507: in the absence of functional assays, this is pure speculation and should be phrased as such.

Response 19:

We thank the reviewer for their insightful comment. Indeed, we strongly agree with the reviewer's insightful suggestion regarding the use of speculative words, due to the absence of functional assays. We revised the text description of CD4⁺ Tex^{prog} cells in the Results section of the revised manuscript, with details as following.

Revision made:

In the revised manuscript, we revised the detailed description as following.

“These CD4⁺ T cells, which exhibit Tfh-like functionality, engage in collaborative interactions with cDC2 cells. They have the potential to promote the activation and subsequent differentiation of GC B cells into antibody-producing plasma cells.”

Comment 20:

Line 518: I believe the authors mean “bioinformatic tools”, not “biological tools”.

Response 20:

Thanks for the reviewer’s careful observation. According to your suggested, we revised the word of “biological tools” replaced by “bioinformatic tools”.

Revision made:

In the revised manuscript, we revised the word “biological” as “bioinformatic”.

*“thus, we employed **bioinformatic** tools to deconvolute the spatial transcriptome without directly observing cell-cell interactions.”*

Comment 21:

Figure 3g: revise header to state “CD4⁺”, not “CD8⁺”.

Response 21:

Thanks for the reviewer’s careful reading and we apologize for the incorrect spelling in our manuscript. Based on the reviewer's suggestion, we revised this header to state “CD4⁺”, not “CD8⁺”.

Revision made:

In the revised manuscript, the “Fig. 3g” was replaced by the “Fig. R39” as following.

Fig. R39. (Fig. 3g) g, Heatmap and pie chart showing the number of clonotype sharing between neighboring functional clusters of CD4⁺ T cells in different status of TLS. For heatmap, the color representing the number of shared clonotypes. For pie chart, the color representing the status of TLS and the size representing the number of shared clonotypes.

Reviewer #4 (Remarks to the Author):

Comment 1:

In this study by Li et al, the investigators base their main analyses on a set of 14 tumors with predefined status of TLS (none, immature and mature TLS), and they apply single cell RNA/TCR sequencing and spatial transcriptomics to this cohort of HNSCC. In these analyses, the key determinant is the allocation to the three different TLS statuses and then define the differences in the tumors where these three states are present. They set off then to identify the key subsets of immune cells (eg CD4⁺ Tex^{prog}), their distribution and interaction, and their role in regulating T and B cell responses within mTLS of the TME. Based on these findings, they developed a TLS-based classification (TLS imprint signature) and applied this signature on bulk-RNAseq from TCGA HNSCC dataset for TLS classification, and found that patients with mTLS status have better clinical outcomes (ie overall survival, disease specific survival and progression-free interval). They also applied this signature on datasets derived from SCC patients treated with anti-PD1 and observed tumor samples with high TLS imprints demonstrated a more favorable response to the treatment.

Response 1:

We sincerely thank you for your thorough review and insightful suggestion. Your summary accurately captures the key aspects of our study, including our analysis of HNSCC tumors with three different TLS statuses, and its potential implications for patient outcomes and immunotherapy response prediction.

Our study aims to contribute to a better understanding of the role of TLS in HNSCC and potentially improve patient stratification and treatment strategies. Your insights are invaluable in helping us refine and strengthen our findings.

We greatly appreciate the time and effort you have invested in evaluating our work.

Comment 2:

The amount of data generated in this study is amazing with more than 200K single cells sequenced, with TCR and BCR sequencing and spatial data, and this would all be incredibly useful reproduces for subsequent analyses. However, I have trouble following the manuscript for a number of key reasons, which dampens my enthusiasm for this manuscript. It is a case of doing way too much and not having a point! It was also very painful to go through the paper as all the figures are not in order that they are cited and some are even missing, which does not make easy reading or reviewing.

Response 2:

We sincerely appreciate your further comments and the time you have invested in reviewing our manuscript. We sincerely apologize for any confusion caused by the disorganized figure order and missing figures. This was an oversight on our part, and we are grateful for your patience in reviewing the manuscript despite these issues. To address your concerns, we revised the manuscript as detailed below.

1. We restructured some parts of the manuscript to present our findings in a more logical and easy-to-follow manner.
2. We revised and ensured all figures are present, properly ordered, and aligned with the text.
3. We provided a clearer narrative that guides the reader through our analyses and their implications.

We are grateful for your constructive criticism, as it will help us significantly improve the quality and clarity of our manuscript. We look forward to submitting a revised version that addresses these issues and better communicates the significance of our findings.

Comment 3:

The most important premise for the entire paper from Figures 1-5 is the ability to classify the 14 tumors into 3 groups, and is the basis of any analyses. The problem is that I am not entirely convinced that they have shown this data completely or accurately. I find it hard to believe that tumors neatly fall into the 3 categories with little gradation or overlap between imTLS and mTLS. This is in fact demonstrated in the differences shown for CD8 and CD4, where the p-values between specific subpopulations are so weak, and often driven by one or two outliers. All the data for CD8 and CD4 trajectory and gene analyses are meaningless as what they deem to be 'stem-like' is actually just naive CD8 and CD4 cells, which if you include in any trajectory from any tumor will show the same loss of naive and gain of dysfunctional markers. I find the data from Figures 1-5 suffer from this repeated error based on a false premise and am completely lost with all the claims, with the good work of TCR and BCR sequencing being completely lost in the process. As I am about to completely give up on the paper, the saving grace arrives in the form of Figure 6 which is the spatial analyses that seems to recapitulate their findings in the tumor and TLS microenvironment which is at least more convincing than the aimless scRNAseq data prior. Validating this data on TCGA or even TMA to show association with outcome is an easy trap to fall under, but the only clinical point that matters is the response to PD1 blockade, and this was very limited. All in all I feel that if the authors presented this paper better, it would be more convincing in my mind to sell as a story. I would have started with the Spatial data and from there branched out to specifically look at the different tumor classes as a range rather than discrete groups and focused on TCR and BCR sequencing and T and B cells respectively. In its current form, it has too many fundamental flaws to be acceptable.

Response 3:

We are deeply grateful for your thorough and critical evaluation of our manuscript. Your insights have provided us with valuable perspectives on the strengths and weaknesses of our study. We sincerely appreciate the time and effort you have invested in this detailed review. Your feedback has highlighted several critical areas, and we will address them through the following aspects.

Firstly, regarding about “your confusion about tumor TLS status classification” as you mentioned, we recognize your skepticism about the categorization of tumors into three different TLS statuses. In this study, we followed the classic classification criteria for imTLS and mTLS cases outlined by Vanhersecke et al.⁶, which is a widely accepted method referenced in many TLS studies^{7, 8}. Specifically, if a sample contains only imTLS without mTLS, it is classified as imTLS. If a sample contains mTLS or both mTLS and imTLS (even when the mTLS proportion is lower), it is defined as mTLS. In fact, there is a situation where both mTLS and imTLS are present in mTLS. In our study, 16 out of 36 samples showed both mTLS and imTLS structures. This finding is consistent with previous report⁶, where 16 out of 41 mTLS tumor samples also displayed imTLS structures in a patient cohort that includes carcinomas and sarcomas. Therefore, we strictly selected tumor samples containing only mTLS, with no imTLS present for scRNA-seq, scTCR/BCR-seq, and spatial transcriptomics on these “mTLS” samples. We applied this strictly method to minimize potential bias from mixed detection of mTLS and imTLS within the same sample.

Secondly, regarding to the “*P*-values between specific subpopulations are so weak” as you mentioned. We appreciate your point about the potential influence of outliers in our analysis. This is mainly due to the inclusion of a limited number of scRNA-seq and spatial transcriptome sequencing samples in our study. Therefore, to address these limitations, we have supplemented our analysis by incorporating data from our cohort of 422 FFPE tumors of HNSCC patients. The results from our cohort study support some of our findings from the single-cell and spatial transcriptome analyses, such as the enrichment of CD8⁺ and CD4⁺ subpopulations in mTLS and the colocalization of CD8, CD4, and DC subclusters in mTLS. These results were in concordance with the outcomes derived from scRNA-seq analysis, providing further support for the robustness and reliability of our study's conclusions.

Thirdly, regarding about the “your concern to trajectories and genetic analyses of CD4 and CD8 T cells” as you mentioned. Indeed, the CD4 and CD8 trajectory and genetic analysis cannot fully demonstrated the differentiation in mTLS. In our study, scTCR/BCR-seq result demonstrated that both B and T cells undergo a continuous differentiation trajectory from stem-like cells to functional cells. (**Fig. R40**). Moreover, our spatial transcriptome analysis traced the progression of these cells from naive to functional states within mTLS. Interestingly, we also observed that functional T cells and plasma cells radiate from the GC of mTLS into the adjacent tumor tissue, further supporting the conclusion that mTLS have the local capacity to functional T cells and plasma cell (**Fig. R41**). Thank you again to the reviewer for your constructive comments.

Fig. R40. TLS induces effects on both B cells and T cells in situ. **a-c**, PAGAs of B cells (**a**), CD8⁺ T cells (**b**) and CD4⁺ T cells (**c**). Each color represents a subcluster of cell. **d-f**, UMAP plot embedding for B cell (**d**), CD8⁺ T cells (**e**) and CD4⁺ T cells (**f**) subclusters, colored by clone size (upper left) and UMAP embedding in different TLS status, colored by subcluster, indicating the different clone size by dot size. **g-i**, Heatmap and pie chart showing the number of clonotype sharing between neighboring functional clusters of B cell (**g**), CD8⁺ T cells (**h**) and CD4⁺ T cells (**i**) in different status of TLS. For heatmap, the color representing the number of shared clonotypes. For pie chart, the color representing the status of TLS and the size representing the number of shared clonotypes.

Fig. R41. Spatial distribution of intratumoral TLS throughout different TLS status of HNSCC. Spatial mapping of granulosa cell types from the scRNA-seq to spatial transcriptomics of nTLS, imTLS and mTLS HNSCC using Cell2location. Spatial mapping of lineage of B cell, CD4⁺ and CD8⁺ T cell and DC from the human scRNA-seq dataset to a respective spatial transcriptomics slide of HNSCC with nTLS, imTLS and mTLS statuses with cell2location. Estimated abundance for cell types (color intensity) to each Visium spot (color) shown over the H&E images. Scale bars = 1 mm.

Fourth, regarding about the “validating this data on TCGA or TMA to show association with the response to PD-1 blockade outcome is limited” as you mentioned. In our study, our analysis of HNSCC tumors with different TLS status and its potential impact on patient outcome and prediction of response to immunotherapy by TCGA-HNSC and TMA cohort. We understand your point about the limited data regarding the response to PD-1 blockade. Therefore, we employed SCC datasets as a proxy to infer potential associations between the presence of mTLS and response to ICB therapy in HNSCC.

Fifth, regarding about the “manuscript structure should present better in this paper” as you mentioned. We appreciate your suggestion for a more effective presentation of our data. We aim to present a comprehensive and systematic single-cell and spatial map of HNSCC, delineating different TLS statuses, in relation to improved outcomes and responses to ICB. Therefore, after analyzing the single-cell data, we plan to unfold our story through the three stages of TLS, showcasing different levels

and types of tumor immune cells. We then combined with the cell-cell interaction and spatial transcriptome to further verified mTLS activate T and B cell responses. We sincerely thank you for your constructive criticism and guidance. According to your feedback, we have also added mIHC staining and mor description content to make our story more cohesive.

Comment 4:

The following are additional specific points that need to be addressed as well:

As stated before, I am not entirely convinced with the IHC images in Extended Data Fig. 2 used to represent nTLS, imTLS and mTLS, as the authors are selective in the regions they want to show. Please provide more representative images with higher resolution for all the tumor and how it was decided to call each tumor into a discreet vein of n, im or mTLS.

Response 4:

Thank you for your careful review and valuable comments on our article. We understand your concern about our image in Extended data Fig. 2. According to your suggestion, we provide all representative mIHC images of CD20, CD23, BCL6, CD4, CD8, and TCF1 with higher resolution to define TLS status.

In this study, we followed the classic classification criteria for imTLS and mTLS cases outlined by Vanhersecke et al.⁶, which is a widely accepted method referenced in many TLS studies. Specifically, if a sample contains only imTLS without mTLS, it is classified as imTLS. If a sample contains mTLS or both mTLS and imTLS (even when the mTLS proportion is lower), it is defined as mTLS.

According to your suggestion, we also add detailed descriptions n for all the tumor used for scRNA-seq and spatial transcriptomics as follow. Finally, thank you again for your valuable comments.

Revision made:

In the revised manuscript, we added the “Fig. R42-44” and the related description as following.

Fig. R42. (Extended Data Fig. 3) Assessment of the presence and status of mTLS by mIHC. Regions with a high density of GC B cells (CD20, BCL6) and T cells (CD4, CD8, TCF1), and follicular dendritic cells (FDCs) (CD23). Scale bars = 100 μ m. Images of a mIHC-stained tumor of mTLS showed aggregation of CD20⁺BCL6⁺ B cells surround by CD4⁺TCF1⁺ T cells and CD8⁺ T cells. CD23⁺ FDCs were distributed in networks or scattered lymphoid aggregates.

Fig. R43. (Extended Data Fig. 4) Assessment of the presence and status of imTLS by mIHC. Regions with a high density of GC B cells (CD20, BCL6) and T cells (CD4, CD8, TCF1), and follicular dendritic cells (FDCs) (CD23). Scale bars = 100 μ m. Images of a mIHC-stained tumor of imTLS showed aggregation of CD20⁺BCL6 B cells surround by CD4⁺TCF1⁺ T cells and CD8⁺ T cells. There was no distribution of CD23⁺ FDCs in lymphoid aggregates.

Fig. R44. (Extended Data Fig. 5) Assessment of the presence and status of imTLS by mIHC. Regions with a high density of GC B cells (CD20, BCL6) and T cells (CD4, CD8, TCF1), and follicular dendritic cells (FDCs) (CD23). Scale bars = 100 μ m. Images of a mIHC-stained tumor of nTLS showed a scattered distribution of B cells and T cells. There was no distribution of CD23⁺ FDCs in lymphoid aggregates.

Revision made:

In the revised manuscript, we added the description about the criteria of define the TLS status.

“IHC and mIHC to determinethe status of TLS. For hematoxylin and eosin staining, OCT-embedded tissues were cut into 6-8 μm sections, separated from the largest diameter of the tumor. Sections were also taken from 1/4 depth and 3/4 depth of the tumor and stained using a Leica automated slide stainer. The TLS status was then further determined by IHC and mIH^{56,57,58}. CD20, CD3, and CD23 immunostaining were used to determine the status of TLS as previously described¹⁶. Briefly, IHC assay were performed using anti-human CD20 (Cell Signaling Technology), CD3 (Abcam), and CD23 (Abcam) antibodies, as previously described⁹. Then, the multiplex immunofluorescence staining was conducted using the Opal 6-Plex Manual Detection Kit (Akoya Biosciences) to further detect the TLS status as previously describe⁹. Briefly, FFPE tissue slides were first deparaffinized and then incubated sequentially with primary antibodies CD20 (1:1000, Cell Signaling Technology), CD4 (1:1000, Abcam), CD8 (1:800, Cell Signaling Technology), BCL6 (1:600, Cell Signaling Technology), CD23 (1:900, Abcam), TCF1 (1:600, Cell Signaling Technology). This was followed by incubation with secondary antibodies (Akoya Biosciences) and corresponding reactive Opal fluorophores. Nuclei acids were stained with DAPI. Seven-color stained slides were scanned using a Vectra 3 Imaging System (Akoya Biosciences). Scanning was conducted at a magnification of 20×. The channel employed for multispectral imaging included DAPI, FITC, Cy3, Texas Red and Cy5. The scanned data is subsequently processed by the inForm software (v.2.4, Akoya Biosciences) for analysis of multispectral imaging. Autofluorescence were acquired by assess the region of the slides that not fluorophores. mTLS is GC formation, a dynamic region with a network of CD23⁺ FDCs and CD20⁺BCL6⁺ B cells^{6,10}. The evaluation criteria for TLS status are as follows as previous study described⁵⁶, if a sample contains only imTLS and not mTLS, it will be classified as imTLS. If a sample contains mTLS or both mTLS and imTLS, it will be defined as mTLS.”

.....

“6. Petitprez, F., et al. B cells are associated with survival and immunotherapy response in sarcoma. Nature 577, 556-560 (2020).

9. Li, H., et al. Tertiary lymphoid structure raises survival and immunotherapy in HPV(-) HNSCC. J Dent Res 102, 678-688 (2023).

10. Fridman, W. H., et al. Tertiary lymphoid structures and B cells: An intratumoral immunity cycle. Immunity 56, 2254-2269 (2023).

16. Vanhersecke, L., et al. Mature tertiary lymphoid structures predict immune checkpoint inhibitor efficacy in solid tumors independently of PD-L1 expression. Nat Cancer 2, 794-802 (2021).

56. Vanhersecke, L., et al. Standardized Pathology Screening of Mature Tertiary Lymphoid Structures in Cancers. Lab Invest 103, 100063 (2023).

57. Haddox, C. L., et al. Phase II Study of Eribulin plus Pembrolizumab in Metastatic Soft-tissue Sarcomas: Clinical Outcomes and Biological Correlates. Clin Cancer Res 30, 1281-1292 (2024).

58. Flippot, R., et al. B cells and the coordination of immune checkpoint inhibitor response in patients with solid tumors. J Immunother Cancer 12, (2024).”

Comment 5:

TLS of different maturation statuses can be distributed throughout the TME. The presence and density of TLS can also vary across different regions of the tumor. Hence, a scoring or quantification system has to be used to classify TLS more rigorously in the context of their findings. The authors have not mentioned them in this manuscript and this needs to be addressed as this forms a crucial part for the subsequent interpretation of the results.

Response 5:

Thank you for your valuable comments and suggestions. Based your advice, we added detailed descriptions about TLS classification standards as following.

In this study, we followed the classic classification criteria for imTLS and mTLS cases outlined by Vanhersecke et al.⁶, which is a widely accepted method referenced in many TLS studies^{7, 8}. Specifically, if a sample contains only imTLS without mTLS, it is classified as imTLS. If a sample contains mTLS or both mTLS and imTLS (even when the mTLS proportion is lower), it is defined as mTLS.

Indeed, considering the traditional definition of TLS status and differences in the distribution and density of TLS in tumors, mTLS will contains imTLS. To optimize the determination of TLS status, we have improved the traditional detection method. Unlike most previous studies that only examined ONE section of tissue using ONE detection method (HE, IHC, or mIHC)^{9, 10, 11}, We, in addition to taking tissue sections from the largest diameter of the tumor, also took sections from 1/4 depth and 3/4 depth of the tumor. These THREE sections were used together to detect the presence and status of TLS. This approach serves to avoid detection bias. Then, the sections to concurrently assess TLS status, following the methods of previous studies^{10, 12}. In order to improve accuracy in evaluating TLS status, we utilized THREE methods including HE, IHC, and mIHC for each sample. Compared to traditional standards, we implemented a more comprehensive and rigorous evaluation process with the goal of accurately identifying mTLS within tumors and minimizing bias.

Therefore, in our study, we strictly selected tumor samples containing only mTLS, with no imTLS present. Under the strictly standards mentioned above, we performed scRNA-seq, scTCR/BCR-seq, and spatial transcriptomics on these “mTLS” samples. We applied this strictly method to minimizes potential bias from mixed detection of mTLS and imTLS within the same sample.

Meanwhile, we also added detailed descriptions for TLS classification standards as follows. Finally, thank you again for your valuable comments.

Revision made:

In the revised manuscript, we added detailed description of TLS classification standards as following.

“IHC and mIHC to determinethe status of TLS. For hematoxylin and eosin staining, OCT-embedded tissues were cut into 6-8 μm sections, separated from the largest diameter of the tumor. Sections were also taken from 1/4 depth and 3/4 depth of the tumor and stained using a Leica automated slide stainer. The TLS status was then further determined by IHC and mIH^{56,57,58}. CD20, CD3, and CD23 immunostaining were used to determine the status of TLS as previously described¹⁶. Briefly, IHC assay were performed using anti-human CD20 (Cell Signaling Technology), CD3 (Abcam), and CD23 (Abcam) antibodies, as previously described⁹. Then, the multiplex immunofluorescence staining was conducted using the Opal 6-Plex Manual Detection Kit (Akoya Biosciences) to further detect the TLS status as previously describe⁹. Briefly, FFPE tissue slides were first deparaffinized and then incubated sequentially with primary antibodies CD20 (1:1000, Cell Signaling Technology), CD4 (1:1000, Abcam), CD8 (1:800, Cell Signaling Technology), BCL6 (1:600, Cell Signaling Technology), CD23 (1:900,

Abcam), TCF1 (1:600, Cell Signaling Technology). This was followed by incubation with secondary antibodies (Akoya Biosciences) and corresponding reactive Opal fluorophores. Nuclei acids were stained with DAPI. Seven-color stained slides were scanned using a Vectra 3 Imaging System (Akoya Biosciences). Scanning was conducted at a magnification of 20×. The channel employed for multispectral imaging included DAPI, FITC, Cy3, Texas Red and Cy5. The scanned data is subsequently processed by the inForm software (v.2.4, Akoya Biosciences) for analysis of multispectral imaging. Autofluorescence were acquired by assess the region of the slides that not fluorophores. mTLS is GC formation, a dynamic region with a network of CD23⁺ FDCs and CD20⁺BCL6⁺ B cells^{6,10}. The evaluation criteria for TLS status are as follows as previous study described⁵⁶, if a sample contains only imTLS and not mTLS, it will be classified as imTLS. If a sample contains mTLS or both mTLS and imTLS, it will be defined as mTLS.”

.....

“6. Petitprez, F., et al. B cells are associated with survival and immunotherapy response in sarcoma. *Nature* 577, 556-560 (2020).

9. Li, H., et al. Tertiary lymphoid structure raises survival and immunotherapy in HPV(-) HNSCC. *J Dent Res* 102, 678-688 (2023).

10. Fridman, W. H., et al. Tertiary lymphoid structures and B cells: An intratumoral immunity cycle. *Immunity* 56, 2254-2269 (2023).

16. Vanhersecke, L., et al. Mature tertiary lymphoid structures predict immune checkpoint inhibitor efficacy in solid tumors independently of PD-L1 expression. *Nat Cancer* 2, 794-802 (2021).

56. Vanhersecke, L., et al. Standardized Pathology Screening of Mature Tertiary Lymphoid Structures in Cancers. *Lab Invest* 103, 100063 (2023).

57. Haddox, C. L., et al. Phase II Study of Eribulin plus Pembrolizumab in Metastatic Soft-tissue Sarcomas: Clinical Outcomes and Biological Correlates. *Clin Cancer Res* 30, 1281-1292 (2024).

58. Flippot, R., et al. B cells and the coordination of immune checkpoint inhibitor response in patients with solid tumors. *J Immunother Cancer* 12, (2024).”

Comment 6:

Figure 1h and 1i are either missing or not appropriately labelled

Response 6:

Thank you for bringing this serious mistake to our attention. We apologize for the missing label on Fig. 1h and i in Fig. 1. We have corrected this labeling error, and the correct label for Fig. 1 is displayed below.

Revision made:

In the revised manuscript, we correct the label for “Fig. 1h, 1i” and the related description as following was added.

Fig. R45. (Fig. 1g-i), The cell types for scRNA-seq of whole tumor cells and the composition difference of CD45⁺ cells (g), H&E and IHC of tumor slides (h), and spatial transcriptomics of tumor tissue (i) in nTLS, imTLS, and mTLS statuses. A color gradient, transitioning from red (indicating enrichment) to blue (signifying depletion), encodes the log₂-transformed odds ratios, while the sizes of the depicted points are governed by the Bonferroni-adjusted -log₁₀(P values), accentuating the statistical significance of observed variations.

Comment 7:

Is Figure 3g showing CD8 or CD4 clonotype sharing?

Response 7:

Thanks for the reviewer’s careful reading and we apologize for the incorrect spelling in our manuscript. In Figure 3g, we revised this header to state “CD4⁺”, not “CD8⁺” with details as following.

Revision made:

In the revised manuscript, the “Fig. 3g” was replaced by the “Fig. R46” as following.

Fig. R46. (Fig. 3g) g, Heatmap and pie chart showing the number of clonotype sharing between neighboring functional clusters of CD4⁺ T cells in different status of TLS. For heatmap, the color representing the number of shared clonotypes. For pie chart, the color representing the status of TLS and the size representing the number of shared clonotypes.

Comment 8:

“Interestingly, CD4⁺ Tex^{term} phenotype exhibited the highest expression of cytotoxic markers (PRF1, GZMA, GZMB, IFNG and GNLY) (Fig. 3g, Extended Data Fig. 10e), aligning with recent discoveries indicating the presence of a phenotype of cytotoxic CD4⁺ T cells within tumors and can directly kill cancer cells. This may suggest that mTLS possess the capability to generate cytotoxic CD4⁺ T cells in situ in the local TME.” Please cite these published findings accordingly.

Response 8:

We sincerely thank the reviewer for their careful reading. As the reviewer suggested, we cited

additional published findings to support this claim in the revised manuscript, with details as following.

Revision made:

In the revised manuscript, the related published findings as following were cited.

“Interestingly, CD4⁺ Tex^{term} phenotype exhibited the highest expression of cytotoxic markers (PRF1, GZMA, GZMB, IFNG and GNLY) (Extended Data Fig. 12e), aligning with recent discoveries indicating the presence of a phenotype of cytotoxic CD4⁺T cells within tumors and can directly kill cancer cells^{27, 28}”

.....

“27. Oh, D. Y., Fong, L. Cytotoxic CD4(+) T cells in cancer: Expanding the immune effector toolbox. Immunity 54, 2701-2711 (2021).

“28. Franken, A., et al. CD4(+) T cell activation distinguishes response to anti-PD-L1+anti-CTLA4 therapy from anti-PD-L1 monotherapy. Immunity 57, 541-558 e547 (2024).”

Comment 9:

Figure 3p is not found

Response 9:

We would like to extend our sincere gratitude to the reviewer for carefully reviewing our work. We apologize for the incorrect description of the label of Figure 3 in the manuscript. The “Fig. 3p” actually refers to “Fig. 3i” in the manuscript. After addressing other comments, some labels in Fig. 3 have changed. After carefully rechecking the manuscript, we have made the necessary revisions to the text description of Figure 3 in the Results section.

Revision made:

In the revised manuscript, the related description of “Figure 3h-o” as following was changed.

“Meanwhile, mTLS accumulated total CD4⁺ T cells and subclusters of CD4⁺ Tconv cells along with the exhaustion trajectory including naive/central memory and progenitor exhausted clusters (Fig.3h–n). On the contrary, nTLS had a larger clone size and made up the major proportion of clusters of Treg (Fig.3e, o), while imTLS developed along both pathways of Tex and Treg and was in between mTLS and nTLS (Fig.3l–o).”

Comment 10:

“Gene enrichment analysis revealed that the CD4⁺ Tex^{prog} subset was significantly enriched with features of Tfh-like functions, including B cell chemotaxis and GC formation (Fig. 3e)”. Should be referring to Fig 3d

Response 10:

We sincerely thank the reviewer for the careful reading. As suggested, we corrected the referring of “Fig. 3e” to “Fig. 3d”.

Revision made:

In the revised manuscript, the referring of “Fig. 3e” was replaced by “Fig. 3d” as following.

“Gene enrichment analysis revealed that the CD4⁺ Tex^{prog} subset was significantly enriched with

features of Tfh-like functions, including B cell chemotaxis and GC formation (Fig. 3d), which is consistent with previous reported studies.”

Comment 11:

“Interestingly, nTLS showed minimal numbers of B cells, except for an enrichment of plasma cells, suggesting that these plasma cells may have originated from outside the TME (Fig. 4b, c).” Please elaborate. How does this finding suggest that nTLS plasma cells may have originated from outside the TME. Can the plasma cells originate from nearby TLS?

Response 11:

We thank the reviewers for the valuable questions. In response to your question about the origin of plasma cells in nTLS, we extensively searched the literature. Indeed, this description in the original article may be oversimplified and does not fully account for the complexity and diversity of plasma cytogenesis.

In tumors enriched with TLSs, there are evidence that TLSs generate and propagate anti-tumor antibody-producing plasma cells¹⁰. At the same time, we found that a part of plasmablasts/plasma cells are derived from intratumoral GC B cells (Fig. 4f). Thus, at least a part of plasma cells in mTLS-tumors are produced intratumorally.

However, in tumors without TLS, the origin of plasma cells is a little complicated. In extra-tumor, plasma cells can be produced in secondary lymph organs and be recruited to tumors^{42, 43}. In tumors, for example, there are reports of the existence of memory B cells, which can give rise to plasma cells^{44, 45}. It means that plasma cells may also be formed by memory B cell differentiation after reactivation in the nTLS tumor microenvironment.

Thus, we admit that the observation of a low number of B cells and a high number of plasma cells in nTLS does not directly infer that plasma cells are generated outside the TME. Basing on current knowledge, we can only conclude that plasma cells in nTLS tumors are generated outside TLSs. According to your suggestions, we have revised the manuscript.

Revision made:

In the revised manuscript, the related description of was modified as following.

“Interestingly, nTLS showed minimal numbers of B cells, except for an enrichment of plasma cells, suggesting that these plasma cells may have originated from outside the TLS (Fig. 4b, c).”

Comment 12:

Line 320-321, the claiming statement “This suggests that mast cells may play a significant role in the process of tertiary lymphoid structure formation” requires further evidence for support. Mast cells may not necessarily play a direct role in TLS formation. Rather, their presence could be a secondary consequence of the inflammation associated with TLS formation. They may act as bystanders that are recruited to the site after TLS formation, instead of actively contributing to the initial development of the structures. Please provide data or references that substantiate the claim.

Response 12:

Thank you for the constructive suggestion from the reviewer. We strongly agree with your insightful comment about the direct and indirect relationship between mast cells and TLS formation. It has not been reported previously about the amount and function of mast cells within TLS. Mast cells

have been proven to have pro-inflammatory and chemotactic abilities¹⁶, but they may act as bystanders and be recruited to the site after TLS formation rather than actively contributing to the initial development of such structures. Based on the reviewer's constructive suggestion, we have removed this erroneous sentence.

Comment 13:

Figure 5e is under-described.

Response 13:

Thanks for the reviewer's careful reading and we apologize for omitting the description of Figure 5e in our manuscript. Detailed information was added and highlight in BLUE in the revised manuscript.

Revision made:

In the revised manuscript, the related description about "Fig. 5e" as following was added.

"The gene enrichment analysis demonstrated that the cDC2 subset, prominently present in mTLS, showed significant enrichment for features associated with the processing and presentation of exogenous antigens, as well as Ig production (Fig. 5e)."

Comment 14:

The histological images provided are of low resolution, making it difficult to accurately visualize and interpret the features described by the authors.

Response 14:

Thank you for your comprehensive review and valuable feedback on our manuscript. Regarding your comment about the histological images having low resolution, we have made adjustments to enhance the clarity of all images. The high-resolution images we initially submitted were of adequate quality. However, it appears that the resolution may have been compromised during the merging of the file. Please review the original files we submitted to access the uncompressed images. We are confident that, with our collaborative efforts with the publisher, the final version of the manuscript will meet the necessary standards for image clarity.

Comment 15:

Figure 6h is included in the main figure but is not referenced or described in the text. Please clarify its relevance or consider removing it if it does not contribute to the discussion.

Response 15:

We sincerely thank the reviewer for their careful observations and apologize for the incorrectly referred to the description of "Figure 6h" as "Figure 6d" in the text. Based on your suggestion, we changed the reference depicted in "Figure 6d" to "Figure 6h".

Revision made:

In the revised manuscript, we added the details as following.

"Using ligand-receptor mapping, we showed the expression of candidate molecules that might promote interactions between mregDC, CD4⁺ Tex^{prog} and effector memory CD8⁺ T cells (Fig. 6h)."

Comment 16:

Line 368-373: Where is PVRL2 in Fig 6d?

Response 16:

We would like to sincerely thank the reviewer for their careful reading. We apologize for including “PVRL2” in the gene list by mistake, and have since deleted it in the context.

Comment 17:

Line 415-417: “This investigation uncovered that specific cell types, specifically stem-like CD8⁺ T cells and CD4⁺ Tex^{prog} demonstrated a strong correlation with B cells (Fig. 7m).” The reported correlations - 0.40 for stem-like CD8⁺ T cells (CD8⁺TCF1⁺) vs. B cells (CD20⁺), 0.45 for CD4⁺ Tex^{prog} (CD4⁺CXCL13⁺) vs. B cells, and 0.56 for CD4⁺ Tex^{prog} (CD4⁺TCF1⁺) vs. B cells - raise the question of what constitutes a 'strong' versus 'weak' correlation. Please clarify the criteria used to define the strength of these correlations. In addition, the population CD4⁺TCF1⁺CXCL13⁺ that was described as CD4 progenitor exhausted phenotypes here is missing in Figure 7m.

Response 17:

We sincerely appreciate the reviewer’s insightful comment regarding our description of correlations. We fully agree with your observation that we did not clearly define the terms “strong” and “weak” in the correlation analysis. In our study, we utilized a cohort of 422 FFPE tumor samples and applied mIHC staining to identify CD20⁺ B cells, CD4⁺ T cells, CD8⁺ T cells, and DC-LAMP⁺ mregDC in each slide. We calculated the Pearson correlation and the strength of correlation were categorized as weak ($r < 0.4$), moderate ($0.4 < r < 0.6$), or strong ($r > 0.6$), following the general consensus and relevant literature^{47, 48, 49}. The significance of correlations was assessed using p-values ($p < 0.05$). We have revised the manuscript to clarify this criterion of correlation coefficients in accordance with the reviewer’s suggestion.

Revision made:

In the revised manuscript, the description of Fig. 7m was revised as following.

“This investigation uncovered that specific cell types, specifically stem-like CD8⁺ T cells and CD4⁺ Tex^{prog} demonstrated a moderate correlation with B cells (Fig. 7m)”

In the revised manuscript, the figure legend of Fig. 7m was revised as following.

“m, Heatmaps showing the pearson correlation among cell subcluster abundances. The color gradient indicates the P value of pearson correlation (red, high expression; white, low expression). Correlation strength was examined using Pearson’s correlation coefficient ($r < 0.40$ weak; $0.40 \leq r < 0.60$ moderate; $r \geq 0.60$ strong); significance of correlation was determined by the p value ($p < 0.05$)”

Comment 18:

Figure 7e is included in the main figure but is not referenced/described in the text. Please clarify.

Response 18:

We sincerely thank the reviewer for their careful observations and apologize for the oversight regarding the main figure 7e is not referenced in the text. As suggested, we have cited the figure 7e according in the revised manuscript.

Revision made:

In the revised manuscript, the details as following were added.

“Additionally, when clinical factors were integrated into a multivariate model, the status of mTLS was found to be significantly correlated with enhanced overall survival, disease specific survival and progression-free interval (Fig. 7e and Extended Data Fig. 13b, c).”

Comment 19:

Figure 7n is too pixelated and cannot be used to support what is described in the text.

Response 19:

Thank you for your thorough review and valuable feedback on our manuscript. Regarding your comment about the histological images having low resolution, we have made adjustments to enhance the clarity of all images. The high-resolution images we initially submitted were of adequate quality. However, it appears that the resolution may have been compromised during the merging of the PDF file. Please review the original files we submitted to access the uncompressed images. We are confident that, with our collaborative efforts with the publisher, the final version of the manuscript will meet the necessary standards for image clarity.

Comment 20:

Line 429-432: “Although there appears to be no difference in the general cellular neighborhoods among CD20⁺ B cells, CD4⁺ and CD8⁺ T cells across different TLS statuses, our findings indicate that the highest frequencies and shortest nearest-neighbor distances between CD4⁺ Tex^{prog} and B cells, as well as stem-like CD8⁺ T cells occur in mTLS.” Please show the data or cite the figure according.

Response 20:

We thank the reviewer for their careful observations and we apologize for omitting to refer related figure. As suggested, we have cited the figure 7o according in the revised manuscript.

Revision made:

In the revised manuscript, the details as following were added.

“Although there appears to be no difference in the general cellular neighborhoods among CD20⁺ B cells, CD4⁺ and CD8⁺ T cells across different TLS statuses, our findings indicate that the highest frequencies and shortest nearest-neighbor distances between CD4⁺ Tex^{prog} and B cells, as well as stem-like CD8⁺ T cells occur in mTLS (Fig. 7o).”

Comment 21:

Lines 433-435: “While the use of a published dataset for validation is commendable, NSCLC represents a different tumor type, which may limit its relevance to HNSCC. Please justify the use of this dataset or consider discussing the potential differences between these tumor types that could affect the findings.”

Response 21:

Thank you for your valuable comments on our research. As your comment, we agree that lung SCC

is a different tumor type compared with HNSCC, although both of them share similar mutation signatures^{50, 51}. Previous research also used RNA-seq data from other cancer types to predict ICB treatment responses^{31, 52}. However, we acknowledge that this approach may introduce potential bias, which is also a limitation of our work.

Comment 22:

Line 463-464: “These cells demonstrate the capacity to develop into fully functional CD8⁺ T cells, suggesting that mTLS fosters an in situ CD8⁺ T cell response.” Please including relevant references.

Response 22:

We thank the reviewer for their insightful comment. As the reviewer suggested, we add relevant references to support this claim in the revised manuscript, with details as following.

Revision made:

In the revised manuscript, the references details as following were added.

“These cells demonstrate the capacity to develop into fully functional CD8⁺ T cells^{30, 45, 46}, suggesting that mTLS fosters an in situ CD8⁺ T cell response.”

.....

*“30. Magen, A., et al. Intratumoral dendritic cell-CD4(+) T helper cell niches enable CD8(+) T cell differentiation following PD-1 blockade in hepatocellular carcinoma. *Nat Med* 29, 1389-1399 (2023).*

*45. Mogilenko, D. A., et al. Comprehensive Profiling of an Aging Immune System Reveals Clonal GZMK(+) CD8(+) T Cells as Conserved Hallmark of Inflammaging. *Immunity* 54, 99-115 e112 (2021).*

*46. Liu, S., et al. Tissue-resident memory CD103⁺CD8⁺ T cells in colorectal cancer: its implication as a prognostic and predictive liver metastasis biomarker. *Cancer Immunol Immunother* 73, 176 (2024).”*

Comment 23:

Lines 470-472: The statement mentions “multiple other studies” confirming the association between TLS and TCF1⁺ T cells. Yet only one citation is provided. Please either provide additional references to support this claim.

Response 23:

We sincerely thank the reviewer for such careful reading. As the reviewer suggested, we added additional references to support this claim in the revised manuscript, with details as following.

Revision made:

In the revised manuscript, the references details as following were added.

“Our earlier study and multiple other studies have confirmed the association between TLS and TCF1⁺ T cells, demonstrating their role in predicting favorable prognoses^{9, 48, 49, 50}.”

.....

*“9. Li, H., et al. Tertiary lymphoid structure raises survival and immunotherapy in HPV(-) HNSCC. *J Dent Res* 102, 678-688 (2023).*

*48. van der Leun, A. M., et al. CD8(+) T cell states in human cancer: insights from single-cell analysis. *Nat Rev Cancer* 20, 218-232 (2020).*

49. Peng, Y., et al. Single-cell profiling of tumor-infiltrating TCF1/TCF7(+) T cells reveals a T

lymphocyte subset associated with tertiary lymphoid structures/organs and a superior prognosis in oral cancer. Oral Oncol 119, 105348 (2021).

50. *Kasikova, L., et al. Tertiary lymphoid structures and B cells determine clinically relevant T cell phenotypes in ovarian cancer. Nat Commun 15, 2528 (2024)."*

References:

1. Teillaud, J. L., et al. Tertiary lymphoid structures in anticancer immunity. *Nat Rev Cancer* **24**, 629-646 (2024).
2. Cabrita, R., et al. Tertiary lymphoid structures improve immunotherapy and survival in melanoma. *Nature* **577**, 561-565 (2020).
3. Jansen, C. S., et al. An intra-tumoral niche maintains and differentiates stem-like CD8 T cells. *Nature* **576**, 465-470 (2019).
4. Joshi, N. S., et al. Regulatory T Cells in Tumor-Associated Tertiary Lymphoid Structures Suppress Anti-tumor T Cell Responses. *Immunity* **43**, 579-590 (2015).
5. Zhang, C., et al. Localization and density of tertiary lymphoid structures associate with molecular subtype and clinical outcome in colorectal cancer liver metastases. *J Immunother Cancer* **11**, (2023).
6. Vanhersecke, L., et al. Standardized Pathology Screening of Mature Tertiary Lymphoid Structures in Cancers. *Lab Invest* **103**, 100063 (2023).
7. Haddox, C. L., et al. Phase II Study of Eribulin plus Pembrolizumab in Metastatic Soft-tissue Sarcomas: Clinical Outcomes and Biological Correlates. *Clin Cancer Res* **30**, 1281-1292 (2024).
8. Flippot, R., et al. B cells and the coordination of immune checkpoint inhibitor response in patients with solid tumors. *J Immunother Cancer* **12**, (2024).
9. Shu, D. H., et al. Immunotherapy response induces divergent tertiary lymphoid structure morphologies in hepatocellular carcinoma. *Nat Immunol* **25**, 2110-2123 (2024).
10. Meylan, M., et al. Tertiary lymphoid structures generate and propagate anti-tumor antibody-producing plasma cells in renal cell cancer. *Immunity* **55**, 527-541 e525 (2022).
11. MacFawn, I. P., et al. The activity of tertiary lymphoid structures in high grade serous ovarian cancer is governed by site, stroma, and cellular interactions. *Cancer Cell* **42**, 1864-1881 e1865 (2024).
12. Vanhersecke, L., et al. Mature tertiary lymphoid structures predict immune checkpoint inhibitor efficacy in solid tumors independently of PD-L1 expression. *Nat Cancer* **2**, 794-802 (2021).
13. Vazquez-Garcia, I., et al. Ovarian cancer mutational processes drive site-specific immune evasion. *Nature* **612**, 778-786 (2022).
14. Hwang, W. L., et al. Single-nucleus and spatial transcriptome profiling of pancreatic cancer identifies multicellular dynamics associated with neoadjuvant treatment. *Nat Genet* **54**, 1178-1191 (2022).
15. Patil, N. S., et al. Intratumoral plasma cells predict outcomes to PD-L1 blockade in non-small cell lung cancer. *Cancer Cell* **40**, 289-300 e284 (2022).
16. Fan, F., et al. Elevated Mast Cell Abundance Is Associated with Enrichment of CCR2+ Cytotoxic T Cells and Favorable Prognosis in Lung Adenocarcinoma. *Cancer Res* **83**, 2690-2703 (2023).
17. Eberhardt, C. S., et al. Functional HPV-specific PD-1(+) stem-like CD8 T cells in head and neck cancer. *Nature* **597**, 279-284 (2021).
18. van der Leun, A. M., et al. CD8(+) T cell states in human cancer: insights from single-cell analysis. *Nat Rev Cancer* **20**, 218-232 (2020).
19. Gattinoni, L., et al. T memory stem cells in health and disease. *Nat Med* **23**, 18-27 (2017).

20. Brightman, S. E., et al. Neoantigen-specific stem cell memory-like CD4(+) T cells mediate CD8(+) T cell-dependent immunotherapy of MHC class II-negative solid tumors. *Nat Immunol* **24**, 1345-1357 (2023).
21. Gattinoni, L., et al. A human memory T cell subset with stem cell-like properties. *Nat Med* **17**, 1290-1297 (2011).
22. Li, Y., et al. Targeting IL-21 to tumor-reactive T cells enhances memory T cell responses and anti-PD-1 antibody therapy. *Nat Commun* **12**, 951 (2021).
23. Lugli, E., et al. Superior T memory stem cell persistence supports long-lived T cell memory. *J Clin Invest* **123**, 594-599 (2013).
24. Sato, Y., et al. Stem-like CD4(+) T cells in perivascular tertiary lymphoid structures sustain autoimmune vasculitis. *Sci Transl Med* **15**, eadh0380 (2023).
25. Ramachandran, M., et al. Tailoring vascular phenotype through AAV therapy promotes anti-tumor immunity in glioma. *Cancer Cell* **41**, 1134-1151 e1110 (2023).
26. Zheng, X., et al. Single-cell analyses implicate ascites in remodeling the ecosystems of primary and metastatic tumors in ovarian cancer. *Nat Cancer* **4**, 1138-1156 (2023).
27. Zheng, L., et al. Pan-cancer single-cell landscape of tumor-infiltrating T cells. *Science* **374**, abe6474 (2021).
28. Cardenas, M. A., et al. Differentiation fate of a stem-like CD4 T cell controls immunity to cancer. *Nature* **636**, 224-232 (2024).
29. Kusnadi, A., et al. Severely ill COVID-19 patients display impaired exhaustion features in SARS-CoV-2-reactive CD8(+) T cells. *Sci Immunol* **6**, (2021).
30. Molina-Alejandre, M., et al. Perioperative chemoimmunotherapy induces strong immune responses and long-term survival in patients with HLA class I-deficient non-small cell lung cancer. *J Immunother Cancer* **12**, (2024).
31. Helmink, B. A., et al. B cells and tertiary lymphoid structures promote immunotherapy response. *Nature* **577**, 549-555 (2020).
32. Kasikova, L., et al. Tertiary lymphoid structures and B cells determine clinically relevant T cell phenotypes in ovarian cancer. *Nat Commun* **15**, 2528 (2024).
33. Lv, J., et al. The tumor immune microenvironment of nasopharyngeal carcinoma after gemcitabine plus cisplatin treatment. *Nat Med* **29**, 1424-1436 (2023).
34. King, H. W., et al. Single-cell analysis of human B cell maturation predicts how antibody class switching shapes selection dynamics. *Sci Immunol* **6**, (2021).
35. Merckenschlager, J., et al. Dynamic regulation of T(FH) selection during the germinal centre reaction. *Nature* **591**, 458-463 (2021).
36. Liu, W., et al. An immune cell map of human lung adenocarcinoma development reveals an anti-tumoral role of the Tfh-dependent tertiary lymphoid structure. *Cell Rep Med* **5**, 101448 (2024).
37. Magen, A., et al. Intratumoral dendritic cell-CD4(+) T helper cell niches enable CD8(+) T cell differentiation following PD-1 blockade in hepatocellular carcinoma. *Nat Med* **29**, 1389-1399 (2023).
38. Oliveira, G., et al. Landscape of helper and regulatory antitumour CD4(+) T cells in melanoma. *Nature* **605**, 532-538 (2022).
39. Liu, Y., et al. Single-cell and spatial transcriptome analyses reveal tertiary lymphoid structures linked to tumour progression and immunotherapy response in nasopharyngeal carcinoma. *Nat Commun* **15**, 7713 (2024).
40. Guilliams, M., et al. Unsupervised High-Dimensional Analysis Aligns Dendritic Cells across Tissues and Species. *Immunity* **45**, 669-684 (2016).
41. Anderson, D. A., 3rd, et al. Genetic models of human and mouse dendritic cell development and function. *Nat Rev Immunol* **21**, 101-115 (2021).
42. Downs-Canner, S. M., et al. B Cell Function in the Tumor Microenvironment. *Annu Rev*

- Immunol* **40**, 169-193 (2022).
43. Chen, P., et al. Tumour-reactive plasma cells in antitumour immunity: current insights and future prospects. *Immunother Adv* **4**, ltae003 (2024).
 44. Weisel, N. M., et al. Comprehensive analyses of B-cell compartments across the human body reveal novel subsets and a gut-resident memory phenotype. *Blood* **136**, 2774-2785 (2020).
 45. Magri, G., et al. Human Secretory IgM Emerges from Plasma Cells Clonally Related to Gut Memory B Cells and Targets Highly Diverse Commensals. *Immunity* **47**, 118-134 e118 (2017).
 46. Edwalds-Gilbert, G., et al. Alternative poly(A) site selection in complex transcription units: means to an end? *Nucleic Acids Res* **25**, 2547-2561 (1997).
 47. Aydogan, M. G., et al. An Autonomous Oscillation Times and Executes Centriole Biogenesis. *Cell* **181**, 1566-1581 e1527 (2020).
 48. Stutz, N., et al. The Fas apoptotic pathway in cutaneous T-cell lymphomas: frequent expression of phenotypes associated with resistance to apoptosis. *J Am Acad Dermatol* **67**, 1327 e1321-1310 (2012).
 49. Paiva, C. E., et al. The brazilian version of the edmonton symptom assessment system (ESAS) is a feasible, valid and reliable instrument for the measurement of symptoms in advanced cancer patients. *PLoS One* **10**, e0132073 (2015).
 50. Alexandrov, L. B., et al. Mutational signatures associated with tobacco smoking in human cancer. *Science* **354**, 618-622 (2016).
 51. Wang, Z., et al. Syngeneic animal models of tobacco-associated oral cancer reveal the activity of in situ anti-CTLA-4. *Nat Commun* **10**, 5546 (2019).
 52. Luoma, A. M., et al. Tissue-resident memory and circulating T cells are early responders to pre-surgical cancer immunotherapy. *Cell* **185**, 2918-2935 e2929 (2022).

Point-by-Point Response to Reviewers' Comments

Reviewers Comments:

=====

Reviewer #1 (Remarks to the Author):

Comment:

I reiterate the fact that the manuscript by Liu et al. provides insights into B and T cell responses associated with tertiary lymphoid structures in HNSCC patients. However, I also reiterate that the conclusions drawn in the study are largely based on correlative analyses, without validation through functional experiments. While correlation can suggest interesting associations, it does not establish causation and, in this case, it is unclear whether the immune cell subpopulations observed in TLShi tissue samples actually originate from the mature TLS, or whether they are simply present due to the surrounding inflammatory environment. Additionally, the data provided are not sufficiently robust to prove that B and T cells induce antitumor responses.

At present, the conclusions appear overstated, as they are based on data that do not fully support such broad claims. To ensure accuracy and precision, I recommend that the authors either conduct functional experiments or, at the very least, adjust their conclusions to reflect the limitations of their data, including the title of the manuscript, as there is no solid evidence showing that “mature tertiary lymphoid structures evoke intra-tumoral T and B cell responses via progenitor exhausted CD4⁺ T cells...”.

Response:

Thank you for your thorough review and valuable feedback on our manuscript. We greatly appreciate your insights regarding the limitations of correlation analyses and the absence of functional experiment validation. We understand that correlation does not imply causation, and recognize how this may impact our conclusions.

In next revisions, we have revised the manuscript to acknowledge the limitations of our data within the manuscript. Additionally, we have also adjusted our conclusions to more accurately reflect these limitations.

We are grateful for your suggestions and believe that these revisions have significantly improved the quality and accuracy of the manuscript.

Revision made:

In the revised manuscripts, we added the limitation of the manuscript.

“Our study has several limitations. Firstly, while our trajectory and scTCR/BCR-seq sharing analyses support the intra-tumoral differentiation of CD8⁺ and CD4⁺ T cells as well as B cells in mTLS, the origin of the T and B cells is already in the site of tumor or from the organ such as tLNs or blood is unclear. Secondly, due to the limitations of technology, we were unable to investigate the TCR/BCR repertoire in a spatial context within the TME. Future studies could benefit from spatial transcriptomics of B and T cell receptors, which may reveal lymphocyte clonal dynamics as previously reported⁵⁴. Thirdly, the localization of different immune cell subtypes was determined using Visium v1 technology with relatively low resolution (55 μm in diameter) and may encompassing 5 to 30 cells per spot; thus, we employed bioinformatic tools to deconvolute the spatial transcriptome without directly observing cell-cell interactions. To enhance this research further, it is recommended to adopt newer methodologies featuring higher resolutions, such as Visium HD technology, multiplexed ion beam imaging by time-of-flight (MIBI-TOF) or imaging mass cytometry (CyCIF), to more accurately delineate the precise structure of mTLS to accurately delineate ligand-receptor interactions⁵⁵. In our study utilizing TMA with HNSCC patients, considering the finite size of each tissue core and the inherent complexity of the three-dimensional (3D) organization of TLS⁵⁶, there exists a possibility of underestimation or uneven representation of TLS maturation stages across different regions, which may introduce a degree of bias into our assessment of TLS status in the samples. Fourthly, our study does not include direct functional assays to confirm the results obtained from single-cell and spatial multi-omics technology data. Our research aimed to study the different statuses of TLS in HNSCC tumors employing an integrative approach that combines single-cell and spatial multi-omics technologies.”

.....

*“54.Engblom, C., et al. Spatial transcriptomics of B cell and T cell receptors reveals lymphocyte clonal dynamics. Science **382**, eadf8486 (2023).*

*55.Keren, L., et al. MIBI-TOF: A multiplexed imaging platform relates cellular phenotypes and tissue structure. Sci Adv **5**, eaax5851 (2019).*

*56.Lin, J. R., et al. Multiplexed 3D atlas of state transitions and immune interaction in colorectal cancer. Cell **186**, 363-381 (2023).”*

Reviewer #2 (Remarks to the Author):

Comment:

All the comments and concerns have been addressed.

Response:

We sincerely thank the reviewers for their detailed review and their efforts to improve the quality of our manuscript. We also appreciate their positive comments

Reviewer #3 (Remarks to the Author):

Comment:

The authors have addressed all of my comments in a satisfying manner. I believe the manuscript is now much stronger through more appropriate bioinformatics analyses and quantitative analyses.

Response:

We sincerely thank the reviewers for their detailed review and their efforts to improve the quality of our manuscript. We also appreciate their positive comments

Reviewer #4 (Remarks to the Author):

Comment 1:

Most of my queries have been satisfactorily responded to, although I feel the the authors has the opportunity to analyse the TLS maturity as a continuum rather than dividing it binarily according to Vanhersecke et al, and given themselves an advantage in doing so. I still feel that Figure 6 would have benefited from coming on earlier in the presentation as the take off point and make this manuscript more readable. I am glad to see that that all the missing figures and figure order have been rectified, as otherwise it is very difficult to wade through such a dense paper.

Response 1:

Thank you for your invaluable suggestions, which are crucial for enhancing the quality of our manuscript. In the previous revision, you provided numerous insights regarding the clarity and organization of the manuscript, particularly concerning the classification of tumors into TLS groups and the interpretation of immune cell data. Your feedback on various details within the article has significantly contributed to improving its quality and increasing the value of our research, for which we are deeply grateful.

In both the previous and current revisions, you recommended moving Figure 6 to an earlier section to enhance the article's readability. In response to this suggestion, we engaged in thorough discussions and careful consideration. However, we have decided to maintain the current order of presentation in the manuscript.

Firstly, our study analyzes the cellular composition of different TLS states at the single-cell level, followed by spatial transcriptomics to examine intercellular interactions, thereby further validating with TMAs that mTLS activate B and T cell immune responses. Crucially, the spatial mapping presented in Figure 6 is derived from deconvolution analysis based on the cell types annotated in the scRNA-seq datasets of Figures 1-5. Without the data from Figures 1-5, it would be impossible to obtain the results of Figure 6. Moreover, the high-resolution advantage of single-cell analysis enable us to more accurately depict the spatial distribution of cells. Therefore, we considered that the current order of Figures may maintain greater scientific rigor and logical coherence.

We fully understand your aim to optimize the article's structure for improved reader comprehension. Finally, we sincerely thank you once again for your constructive feedback and ongoing efforts to refining this manuscript.

Comment 2:

Not qualified to review code.

Response 2:

Thanks for the reviewer's careful observation and we apologize for the description of incorrect code storage locations in our manuscript. According to request, we correct the link web of the code.

Revision made:

In the revised manuscripts, we revised the accession code in the “Code availability” Section.

*“Codes used in this study are available at the GitHub website
(https://github.com/lihaowhusos/TLS_in_HNSCC).”*